# γδ T cell-derived IL-4 initiates CD8+ T cell immunity

Shirley Le [1], Nick Dooley [2], Declan Murphy[1], Shangyi Liu[1], Luke C. Gandolfo [3], Zhengyu Ge [1], Rose May[1], Anton Cozijnsen[4], Thomas N. Burn [1], Charlie Jennison[4], Annabell Bachem[1], Calvin Xu [1], Hui-Fern Koay [1], Jan Schröder[1,5], Damian Oyong [2,6], Mayimuna Nalubega[2], Enny Kenangalem[7,8], Stephanie Gras [9,10,11], Ian A. Cockburn [12], Sammy Bedoui [1], Laura K. Mackay [1], Geoffrey I. McFadden [4], Daniel Fernandez-Ruiz [1,13], Michelle Boyle [1,2,7], William R. Heath [1] & Lynette Beattie [1] ✉

Dendritic cells (DCs) are pivotal for initiating adaptive immunity, a process triggered by the activation of DCs via pathogen products or damage. Immunization with sporozoites from *Plasmodium* leads to CD8+ T cell priming in a response initiated by collaboration between conventional type 1 DCs (cDC1s) and γδ T cells. Here we show that Vγ1+ γδ T cells have an initiating role by directly supplying interleukin-4 (IL-4). IL-4 and interferon-γ (IFNγ) synergize with CD4+ T cell-derived CD40L to induce IL-12 production by cDC1. Both IL-12 and IL-4 then directly signal responding CD8+ T cells and drive enhanced IL-12 receptor expression and expansion. This study shows how Vγ1+ γδ T cells can initiate CD8+ T cell immunity to *Plasmodium* and that responses to some pathogens require help from innate-like T cells to pass an initiation threshold and further amplify the response in a process underscored by IL-4 production.

Priming of CD8+ T cells is a complex process that relies on antigen presentation by dendritic cells (DCs) that are sufficiently activated to provide all the necessary costimulatory signals. Classically, DC activation involves pattern recognition receptor (PRR) based detection of the pathogen itself and inflammatory cues associated with the encounter. These signals increase costimulatory molecule expression on the DC and enhance MHC-II expression for antigen presentation to CD4+ T cells[1]. CD4+ T cells then provide CD40L, thereby licensing the DC[2–4] and stimulating production of cytokines that contribute to T cell differentiation and expansion[5]. DCs that are mature but not completely activated (or immunogenic) fail to produce pro-inflammatory cytokines[1], resulting in impaired T cell-mediated immunity[6]. Immune responses to pathogens that induce strong PRR signaling in DCs are less dependent on CD40L-mediated help from CD4+ T cells[7]. Conversely, the responses to some pathogens[7] and other poor immunogens are more dependent upon CD4+ T cell help for full DC activation and licensing[8].

[1]Department of Microbiology and Immunology and The Peter Doherty Institute for Infection and Immunity, University of Melbourne, Parkville, Victoria, Australia. [2]Burnet Institute, Life Science Division, Melbourne, Victoria, Australia. [3]Melbourne Bioinformatics, University of Melbourne, Parkville, Victoria, Australia. [4]School of Biosciences, University of Melbourne, Parkville, Victoria, Australia. [5]Computational Sciences Initiative, Department of Microbiology and Immunology, University of Melbourne, Parkville, Victoria, Australia. [6]Global and Tropical Health Division, Menzies School of Health Research, Darwin, Northern Territory, Australia. [7]Menzies School of Health Research-National Institute of Health Research and Development Malaria Research Program, Timika, Indonesia. [8]District Health Authority, Timika, Indonesia. [9]Infection and Immunity Program, La Trobe Institute for Molecular Science (LIMS), La Trobe University, Bundoora, Victoria, Australia. [10]Department of Biochemistry and Chemistry, School of Agriculture, Biomedicine and Environment, La Trobe University, Bundoora, Victoria, Australia. [11]Department of Biochemistry and Molecular Biology, Monash University, Clayton, Victoria, Australia. [12]Division of Immunology and Infectious Disease, John Curtin School of Medical Research, Australian National University, Canberra, Australian Capital Territory, Australia. [13]Present address: School of Biomedical Sciences, Faculty of Medicine & Health, and the UNSW RNA Institute, The University of New South Wales, Kensington, New South Wales, Australia. ✉e-mail: lynette.beattie@unimelb.edu.au

CD8[+] T cell responses to radiation-attenuated *Plasmodium* sporozoites (RAS) are completely dependent upon help from CD4[+] T cells[9,10], placing *Plasmodium* sporozoites in the category of poor immunogens.

The cytokines produced in response to DC activation and classically associated with differentiation and expansion of CD8[+] T cells include type I interferons, interleukin-2 (IL-2), IL-12 and interferon-γ (IFNγ)[11], but not commonly IL-4. One model known to be dependent on IL-4 is intravenous immunization with RAS[12,13]. RAS injection induces protective immunity due to activation of CD8[+] T cells in the spleen by conventional type 1 DCs (cDC1s)[14]. These activated CD8[+] T cells then expand and a proportion of them differentiate into memory phenotype cells, including liver resident memory T cells (T$_{RM}$) that can protect against reinfection[15]. Strikingly, in both animal models and human vaccination, γδ T cells are associated with the success of this response[16–20], though just how these cells contribute is unclear.

γδ T cells are innate sensors[21] that respond to infection, stress or damage[21,22]. Once activated, γδ T cells exert diverse effector functions including the rapid release of cytokines[23–25]. In mice, Vγ1[+] and Vγ4[+] populations[26] are found in lymphoid tissues, including the spleen[22]. In humans, the predominant γδ T cell subsets express Vδ1[+], Vδ2[+] or Vδ3[+], with the majority of Vδ2[+] cells preferentially pairing with Vγ9 (ref. 27). Vγ9[+]Vδ2[+] T cells are the most abundant γδ T cell population in human peripheral blood and are also found in the spleen[28].

Here, we show that the Vγ1[+] subset of γδ T cells in mice initiates the CD8[+] T cell response to liver-stage malaria parasites via IL-4 production. Human Vγ9[+]Vδ2[+] T cells also produce IL-4 in response to natural infection. These data have implications for our understanding of how CD8[+] T cell responses to weak immunogens can be strengthened early in the priming phase and identify a function for Vγ1[+] γδ T cells and IL-4 in enhancing CD8[+] T cell responses.

## Results

### γδ T cells affect early expansion of CD8[+] and CD4[+] T cells

Intravenous RAS injection initiates priming and expansion of T cells in the spleen, followed by recirculation and resultant accumulation of activated T cells in the liver[14], a proportion of which will then differentiate into memory subsets, including protective liver T$_{RM}$ cells[15]. To understand if the previously described role for γδ T cells in memory T cell responses in RAS-vaccinated mice[16] was due to a role for γδ T cells in T cell priming or the generation of memory T cells, we used PbT-I TCR transgenic T cells to study the response at an antigen-specific level. These T cells recognize an H-2K[b]-restricted epitope (PbRPL6$_{120-127}$) from the *P. berghei*-derived RPL6 protein[14,29]. PbT-I cells were adoptively transferred into B6 (wild-type (WT)) or TCRδ[−/−] (δ[−/−]) mice and responses assessed 6 days after vaccination with RAS (Fig. 1a–c and Extended Data Fig. 1a). This revealed a defect in the accumulation of PbT-I cells in the spleen (Fig. 1a), the liver-draining lymph node (Fig. 1b) and, consequently, the liver (Fig. 1c) of δ[−/−] mice. As a result of this early failure to respond to RAS, fewer memory PbT-I cells were found in the spleen (Fig. 1d) and liver (Fig. 1e) after 3 weeks, with impaired formation of memory T cells subsets, including liver T$_{RM}$ cells (Fig. 1f,g).

CD8[+] T cells within the endogenous repertoire specific for PbRPL6$_{120-127}$[29] also showed impaired accumulation in the spleen and liver of δ[−/−] mice at 6 days (Extended Data Fig. 1b–d) and did not form memory in the spleen or liver (Extended Data Fig. 1e–h). γδ T cells were thereby required for the initiation phase of the CD8[+] T cell response to RAS.

To assess whether genetically attenuated parasite vaccination was also γδ T cell dependent, B6 or δ[−/−] mice were vaccinated with *P. berghei* gene deletion mutants lacking the *mei2* gene (*PbΔmei2*), which do not differentiate into blood-stage parasites but persist late into the liver stage, like the Δ*mei2 P. falciparum* parasites[30]. A single dose of *PbΔmei2* generated impaired memory CD8[+] T cells against RPL6 in the spleen (Fig. 1h,j) and the liver (Fig. 1i,k) of δ[−/−] mice. Thus, vaccination with genetically attenuated parasites also required γδ

T cells. Of note, the poor initial response in δ[−/−] mice did not extended to vaccination with blood-stage parasites (irradiated infected red blood cells) (Extended Data Fig. 1i), despite this response being categorized as relatively weak based on its CD4[+] T cell help dependence[31].

We next assessed whether the initiation of the CD4[+] T cell response to RAS vaccination was also γδ T cell dependent. Splenic accumulation of MHC-II restricted *P. berghei*-specific PbT-II cells was dependent upon γδ T cell-mediated help in response to RAS vaccination (Fig. 1l,m), also resulting in fewer PbT-II cells in the livers of δ[−/−] mice (Extended Data Fig. 1j). PbT-I cell accumulation in the spleen (Fig. 1n) and liver (Extended Data Fig. 1k) of the δ[−/−] mice was also lower than the controls, demonstrating that the addition of large numbers of naive antigen-specific CD4[+] T cells could not rescue the response of CD8[+] PbT-I cells when γδ T cells were absent.

### Vγ1[+] γδ T cells initiate immunity to RAS

Blocking γδ T cell function before RAS vaccination with the pan-TCRδ blocking antibody (α-γδ, clone GL3) (Fig. 2a) impaired the PbT-I response comparably to that previously observed in δ[−/−] mice (Fig. 2b). In contrast, blocking γδ T cell function from 24 h after RAS injection had a milder effect (Fig. 2b), suggesting that the first 24 h are the crucial window of γδ T cell activation. We thereby examined early γδ T cell activation by splitting them into the two major populations found in the periphery: Vγ1[+] or Vγ1[−] populations (Extended Data Fig. 2a). In the spleen, Vγ1[+] γδ T cells upregulated the canonical T cell activation markers CD69 and CD25 1 day after RAS vaccination (Fig. 2c) with a transient increase in the proportion (Fig. 2d) and number (Fig. 2e) of activated cells. A small increase in the number and proportion of activated Vγ1[−] γδ T cells was also observed in the spleen (Fig. 2f,g and Extended Data Fig. 2a). γδ T cell expansion within both Vγ1[+] and Vγ1[−] populations was observed from day 2 (Fig. 2h,i). In the liver, a small increase in the activation of Vγ1[+], but not Vγ1[−], γδ T cells, was detected at day 1 (Extended Data Fig. 2b–f), and an equivalent small increase in activated Vγ1[+] γδ T cells seen in the liver dLN at day 2 (Extended Data Fig. 2g–k), indicating a slight delay in the initiation of the liver dLN response. Expansion of both Vγ1[+] and Vγ1[−] γδ T cell populations was seen in the liver dLN but not in the liver (Fig. 2j–m). To confirm that downregulation of the TCR due to γδ T cell activation did not result in an inability to detect activated cells, we also examined γδ T cell activation in TCRδ-GDL mice, in which green fluorescent protein (GFP) can be used to detect γδ T cells. These analyses showed a similar level of activation of Vγ1[+] γδ T cells within the spleen (Extended Data Fig. 2l,m). As specific activation of Vγ1[+] γδ T cells was suggested, the functional consequence of impairing Vγ1[+] γδ T cell activity was tested (Fig. 2n). Blockade of Vγ1[+] γδ T cells had a similar effect on PbT-I accumulation as blockade of the entire γδ T cell population (α-γδ) (Fig. 2o). Vγ1[+] γδ T cells therefore initiate the CD8[+] T cell response to RAS in mice.

### Antigen presentation is intact in the absence of γδ T cells

Due to the very early timing of γδ T cell activation, we hypothesized that the Vγ1[+] γδ T cells may impact the ability of cDC1 to present antigen to CD8[+] T cells in response to RAS vaccination. We therefore examined the initial upregulation of CD69 within OT-I cells following injection of SIINFEKL expressing CS5M sporozoites (CS5M-RAS). OT-I cells were used because adoptive transfer leads to spontaneous upregulation of CD69 on PbT-I cells for reasons yet to be determined, but OT-I cells allow for sensitive detection of CD69 upregulation in this system. OT-I T cells recapitulated the phenotype of PbT-I cells in δ[−/−] mice, with impaired responses in the spleen, liver dLN and liver 6 days after vaccination with CS5M-RAS (Extended Data Fig. 3a–c). At early time points post-CS5M-RAS vaccination, equal proportions and numbers of cell trace violet (CTV) labeled OT-I cells were present, upregulated cell surface expression of CD69 and divided in the spleens of B6 and δ[−/−] mice (Fig. 3a–g). We observed little evidence of OT-I cell activation in the liver (Extended Data Fig. 3d–j) and a small level of

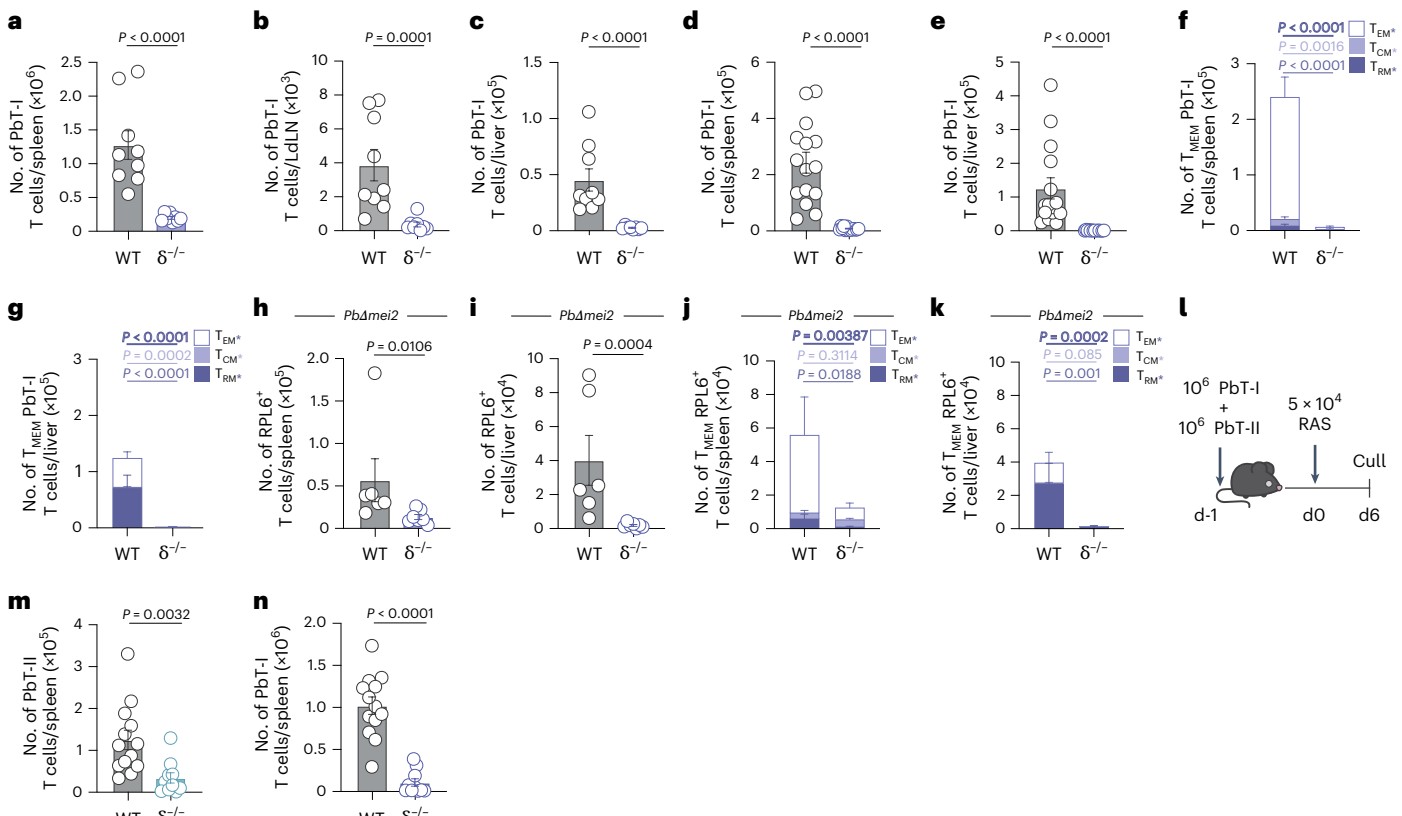

**Fig. 1 | γδ T cells are essential for CD8⁺ T cell response to RAS.** A total of $10^6$ RPL6-specific transgenic CD8⁺ T cells (PbT-I) were transferred into mice 1 day before vaccination with $5 \times 10^4$ RAS. PbT-I cells were analyzed either 6 or 23 days later. **a–c**, Enumeration of PbT-I cells in the spleen (**a**), liver-draining lymph nodes (LdLN) (**b**) or liver (**c**) of WT (B6) ($n = 9$), or $Tcrd^{-/-}$ ($\delta^{-/-}$) ($n = 8$) mice 6 days after immunization. **d,e**, Numbers of PbT-I cells in the spleen (**d**) or liver (**e**) of WT ($n = 15$), or $\delta^{-/-}$ ($n = 14$) mice 23 days after vaccination. **f,g**, Quantified memory T cell ($T_{MEM}$) subsets; central memory ($T_{CM}$; CD62L⁺, CD69⁻), effector memory ($T_{EM}$; CD62L⁻, CD69⁻) and resident memory ($T_{RM}$; CD62L⁻, CD69⁺) within the CD44⁺ PbT-I cell compartment of WT ($n = 15$) or $\delta^{-/-}$ ($n = 14$) spleens (**f**) or livers (**g**) at day 23. Mice were vaccinated with $2 \times 10^4$ *PbΔmei2* sporozoites; 35 days later, mice were culled. **h,i**, Numbers of RPL6 tetramer⁺ cells in the spleen (**h**) and liver (**i**) of WT ($n = 6$) or $\delta^{-/-}$ ($n = 7$) mice 35 days after vaccination. **j,k**, Quantified $T_{MEM}$ subsets; $T_{CM}$ (CD62L⁺, CD69⁻), $T_{EM}$ (CD62L⁻, CD69⁻) and $T_{RM}$ (CD62L⁻, CD69⁺) within the CD44⁺ RPL6⁺ T cell compartment of WT ($n = 6$) or $\delta^{-/-}$ ($n = 7$) spleens (**j**) or livers (**k**) at day 35. **l**, Experimental design. **m,n**, Transgenic CD8⁺ (PbT-I) and CD4⁺ (PbT-II) T cells were transferred into mice 1 day before vaccination with RAS. PbT-II (**m**) or PbT-I (**n**) cell counts in the spleen at day 6 (WT, $n = 13$; $\delta^{-/-}$, $n = 10$). Data show two (**a,c,h–k**) or three (**b,d–g,m,n**) independent experiments where points represent individual mice and bars represent mean. Error bars indicate mean + s.e.m. Data were log-transformed and compared using an unpaired two-tailed Welch's $t$-test.

activation in the liver dLN, but this was equivalent between B6 and $\delta^{-/-}$ mice (Extended Data Fig. 3k–q). These data indicated that responding T cells had access to antigenic signals capable of upregulating CD69 and initiating some proliferation even when γδ T cells were absent. Failure to accumulate at day 6 (Extended Data Fig. 3a–c), however, suggested a lack of signals for extended T cell expansion, differentiation and/or survival. This was suggestive of a lack of co-stimulation or cytokines in the T cell priming process, raising the intriguing possibility that γδ T cells were crucial for triggering upregulation of co-stimulation-like signals on cDC1.

**γδ T cells are not required to supply the CD40L signal to cDC1s**
We next hypothesized that γδ T cells may provide CD40L for signaling CD40 on cDC1s (ref. 32). To investigate this possibility, PbT-I cells were transferred into WT, $\delta^{-/-}$ or $Cd40lg^{-/-}$ ($Cd40l^{-/-}$) mice. CD8⁺ T cell expansion is dependent on help from CD4⁺ T cells in the RAS model[9,10], and these CD4⁺ T cells were previously presumed to provide the CD40L signal. To test this assumption, one group of $Cd40l^{-/-}$ mice was also given CD40L-sufficient PbT-II cells (Fig. 3h). PbT-I cell accumulation in the spleen was severely diminished in $Cd40l^{-/-}$ mice (Fig. 3i), confirming that CD8⁺ T cell accumulation in this model is dependent on CD40L signaling. Addition of CD40L-sufficient PbT-II cells, however, rescued the PbT-I cell response, even though γδ T cells lacked expression of

CD40L in these mice. Therefore, CD4⁺ T cells can provide the CD40L signal required, suggesting that γδ T cell-derived CD40L is not essential to this response. Furthermore, rescue by CD40L-sufficient PbT-II cells showed that this signal is normally provided by CD4⁺ T cells but is insufficient to ensure an appropriate CD8⁺ T cell response if γδ T cells are absent (Fig. 3i,j).

**IL-4 is required for the CD8⁺ T cell response to RAS vaccination**
As CD40L was not the crucial signal provided by γδ T cells, we rationalized that they may provide 'signal 3' that is a cytokine signal, for differentiation and expansion of responding T cells. As IL-4 was identified as a crucial factor in the generation of memory CD8⁺ T cells in the liver following RAS vaccination[12,13] and splenic γδ T cells can secrete IL-4 and IFNγ[24,33], we next investigated the role of these cytokines. Six days after RAS vaccination, there was an increase in the number and proportion of splenic Vγ1⁺ γδ T cells that produced both IFNγ and IL-4 (Extended Data Fig. 4a–c). Vγ1⁻ γδ T cells made IFNγ in response to RAS but no detectable IL-4 (Extended Data Fig. 4d–f). IFNγ blockade (Fig. 4a) impaired the accumulation of PbT-I but did not completely recapitulate the effect of γδ T cell blockade (α-γδ) (Fig. 4b) and had no effect on the accumulation of PbT-II T cells (Fig. 4c). To investigate if γδ T cells were a crucial source of IFNγ for PbT-I cell accumulation, we generated several groups of mixed bone marrow (BM) chimeras

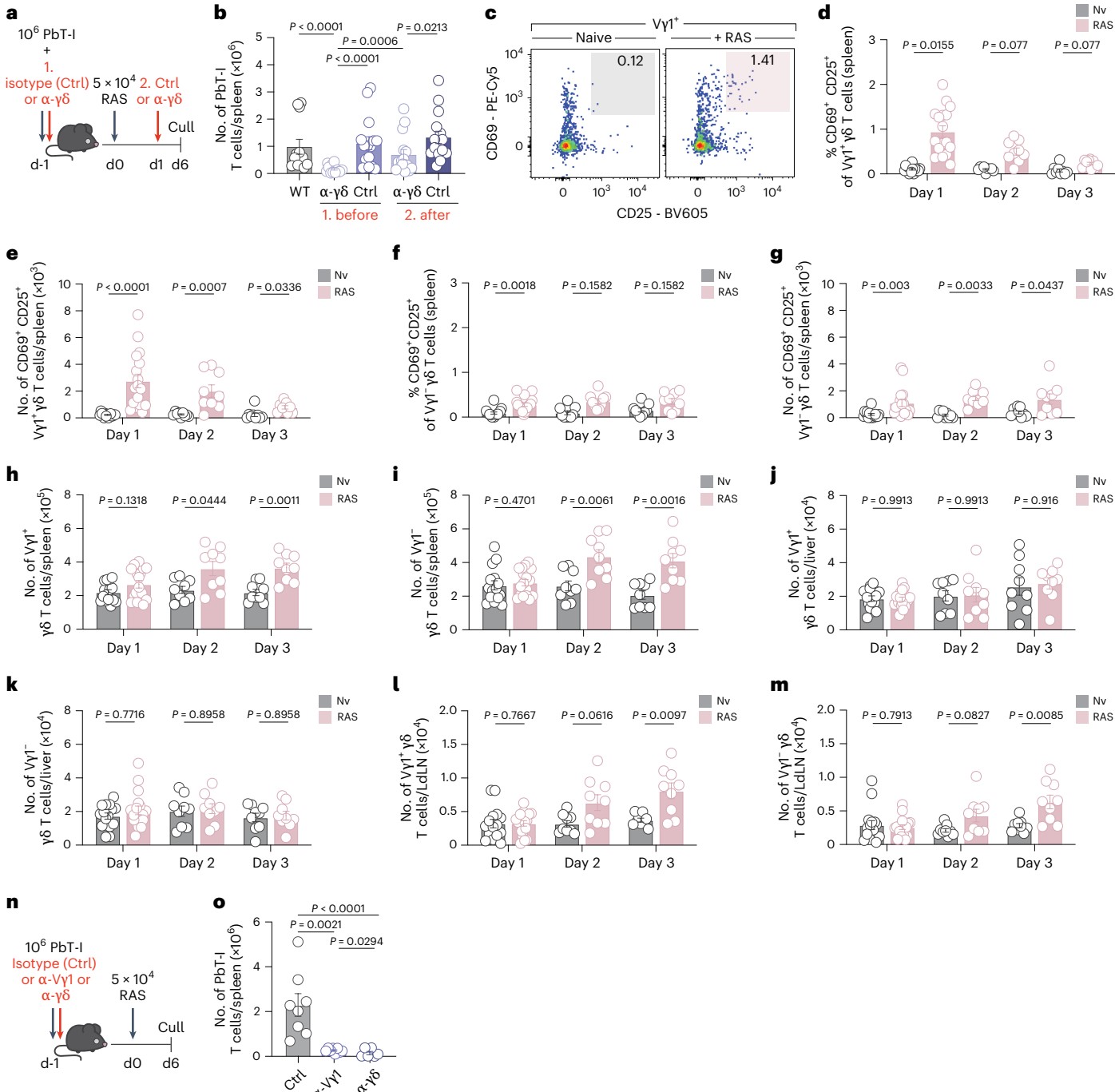

**Fig. 2 | Splenic Vγ1⁺ γδ T cells initiate immunity to RAS. a**, Experimental design. PbT-I cells were transferred into recipient mice 1 day before RAS vaccination (WT, *n* = 12). Mice were treated with an α-pan-γδ-TCR (α-γδ, clone GL3) (i.v.) or an isotype control mAb (Ctrl) (i.v.) either (1) before RAS (α-γδ, *n* = 16; Ctrl, *n* = 15) or (2) 24 h after RAS (α-γδ, *n* = 15; Ctrl, *n* = 15). **b**, Numbers of PbT-I cells in the spleen were assessed at day 6 post-vaccination. **c**, Vγ1⁺ γδ T cells in naive (Nv) (left) or RAS-vaccinated (right) WT (B6) mice 1 day after 5 × 10⁴ RAS vaccination. **d,f**, Frequency of activated (CD69⁺ CD25⁺) Vγ1⁺ (**d**) or Vγ1⁻ (**f**) γδ T cells in naive or RAS-vaccinated WT spleens at days 1 (Nv, *n* = 15; RAS, *n* = 17), 2 (Nv, *n* = 9; RAS, *n* = 9) or 3 (Nv, *n* = 9; RAS, *n* = 9) post-RAS. **e,g**, Number of activated (CD69⁺ CD25⁺)

Vγ1⁺ (**e**) or Vγ1⁻ (**g**) γδ T cells in naive or RAS-vaccinated WT spleens at days 1 to 3 post-injection. **h,i**, Number of splenic Vγ1⁺ (**h**) or Vγ1⁻ (**i**) γδ T cells. **j,k**, Number of liver Vγ1⁺ (**j**) or Vγ1⁻ (**k**) γδ T cells. **l,m**, Number of LdLN Vγ1⁺ (**l**) or Vγ1⁻ (**m**) γδ T cells. **n**, Experimental design. **o**, Numbers of PbT-I cells at day 6 post-vaccination in isotype control mAb-treated (Ctrl) (*n* = 8), α-Vγ1-treated (*n* = 8) (i.v.) or α-pan-γδ-TCR-treated (*n* = 6) (i.v.) mice. Data show three (**a,b**) or four (**c–o**) independent experiments where points represent individual mice and bars represent mean. Error bars indicate mean + s.e.m. Data were log-transformed and compared using an ordinary one-way ANOVA or multiple unpaired two-tailed Welch's *t*-tests (**b,o**) and corrected with Holm–Šidák multiple comparisons test (**d–m**).

(Fig. 4d,e). Eight weeks after reconstitution, the γδ T cell compartment was reconstituted as expected (Fig. 4f). Assessment of the PbT-I and PbT-II cell responses 6 days after RAS vaccination demonstrated equal PbT-I (Fig. 4g) and PbT-II (Fig. 4h) cell accumulation in the spleens of *Ifng*⁻/⁻ + δ⁻/⁻→B6 chimeras (Group 6) when compared to B6 + δ⁻/⁻→B6

chimeras (Group 5) (Fig. 4g,h), suggesting that γδ T cells were not an essential source of IFNγ for PbT-I T cell accumulation. Accumulation of both PbT-I and PbT-II T cells was also similar between B6 → B6 (Group 1) and *Ifng*⁻/⁻→B6 (Group 2) chimeras (Fig. 4g,h), suggesting either that the source of IFNγ was not BM derived, or that the transferred PbT-I and/

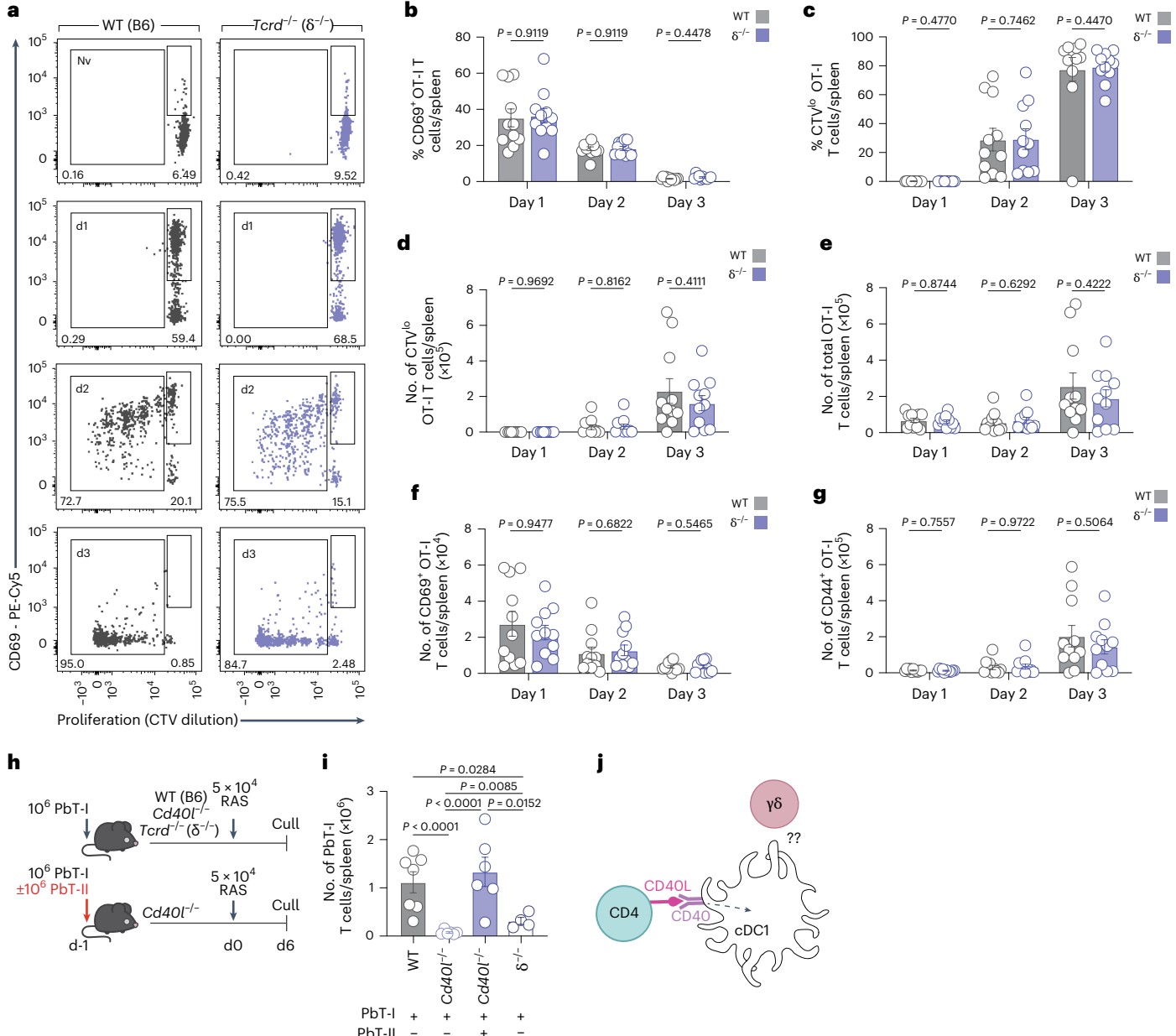

**Fig. 3 | Antigen presentation and CD40L signals are intact in the absence of γδ T cells.** A total of $10^6$ CTV-labeled OVA-specific CD8$^+$ (OT-I) T cells were transferred into WT or $\delta^{-/-}$ recipient mice, which were vaccinated 1 day later with $5 \times 10^4$ CS5M-OVA RAS. OT-I cell numbers were assessed at days 1 to 3 (WT $n = 11$; $Tcrd^{-/-}$ $n = 11$, per day) after transfer. **a**, Flow cytometry plots of total OT-I cells in the spleen. **b**, Frequency of CD69$^+$ OT-I cells in the spleen of either WT or $\delta^{-/-}$ mice. **c,d**, Frequency (**c**) and number (**d**) of CTV$^{lo}$ OT-I cells in the spleen of either WT or $\delta^{-/-}$ mice. **e–g**, Number of total (**e**), CD69$^+$ (**f**) or CD44$^+$ (**g**) splenic OT-I cells. **h**, Experimental design. **i**, WT ($n = 7$), $\delta^{-/-}$ ($n = 4$) and $Cd40l$($Cd154$)$^{-/-}$ mice received PbT-I cells with ($n = 6$) or without co-transferred PbT-II cells ($n = 6$) 1 day before vaccination with RAS. **i**, PbT-I cell counts in the spleen at day 6 post-vaccination. **j**, CD40 signaling to DCs is intact when CD40L-sufficient T cells are provided, showing that CD4$^+$ T cells can provide CD40L in this system. Data show two (**i**) or three (**a–g**) independent experiments where points represent individual mice and bars represent mean. Error bars indicate mean + s.e.m. Data were log-transformed and compared using multiple unpaired two-tailed Welch's *t*-tests and corrected using Holm–Šidák multiple comparisons test (**b–g**) or an ordinary one-way ANOVA (**i**).

or PbT-II T cells were providing IFNγ. Co-transfer of PbT-I and PbT-II T cells into IFNγ$^{-/-}$ hosts (Fig. 4i) further demonstrated no impairment in PbT-I or PbT-II T cell accumulation in the spleen at day 6 (Fig. 4j,k). When combined, these data suggest that IFNγ contributes to PbT-I T cell accumulation in response to RAS vaccination, but γδ T cells are not the essential source of this cytokine, which is likely derived from CD4$^+$ and/or CD8$^+$ T cells.

Strikingly, IL-4 had a profound effect on PbT-I accumulation either when α-IL-4 antibody was used to mediate blockade (Extended Data Fig. 4g,h) or responses were measured after transfer of PbT-I cells

into $Il4^{-/-}$ hosts (Fig. 4l,m). IL-4 was therefore crucial for the response, and the absence of IL-4 mirrored the effect of γδ T cell deficiency.

To further investigate if, as previously reported, CD4$^+$ T cells were the relevant source of IL-4[12], we transferred IL-4-sufficient PbT-I cells with or without IL-4-sufficient PbT-II cells into B6 or $Il4^{-/-}$ mice (Fig. 4l). Accumulation of PbT-I cells was impaired in the $Il4^{-/-}$ mice even in the presence of IL-4-sufficient antigen-specific CD4$^+$ T cells (Fig. 4n). In contrast, antigen-specific CD4$^+$ T cells expanded in the spleen in both B6 and $Il4^{-/-}$ mice (Fig. 4o), demonstrating two phenomena: 1) the CD4$^+$ T cell response was not IL-4 dependent and 2) IL-4-sufficient CD4$^+$ T cells

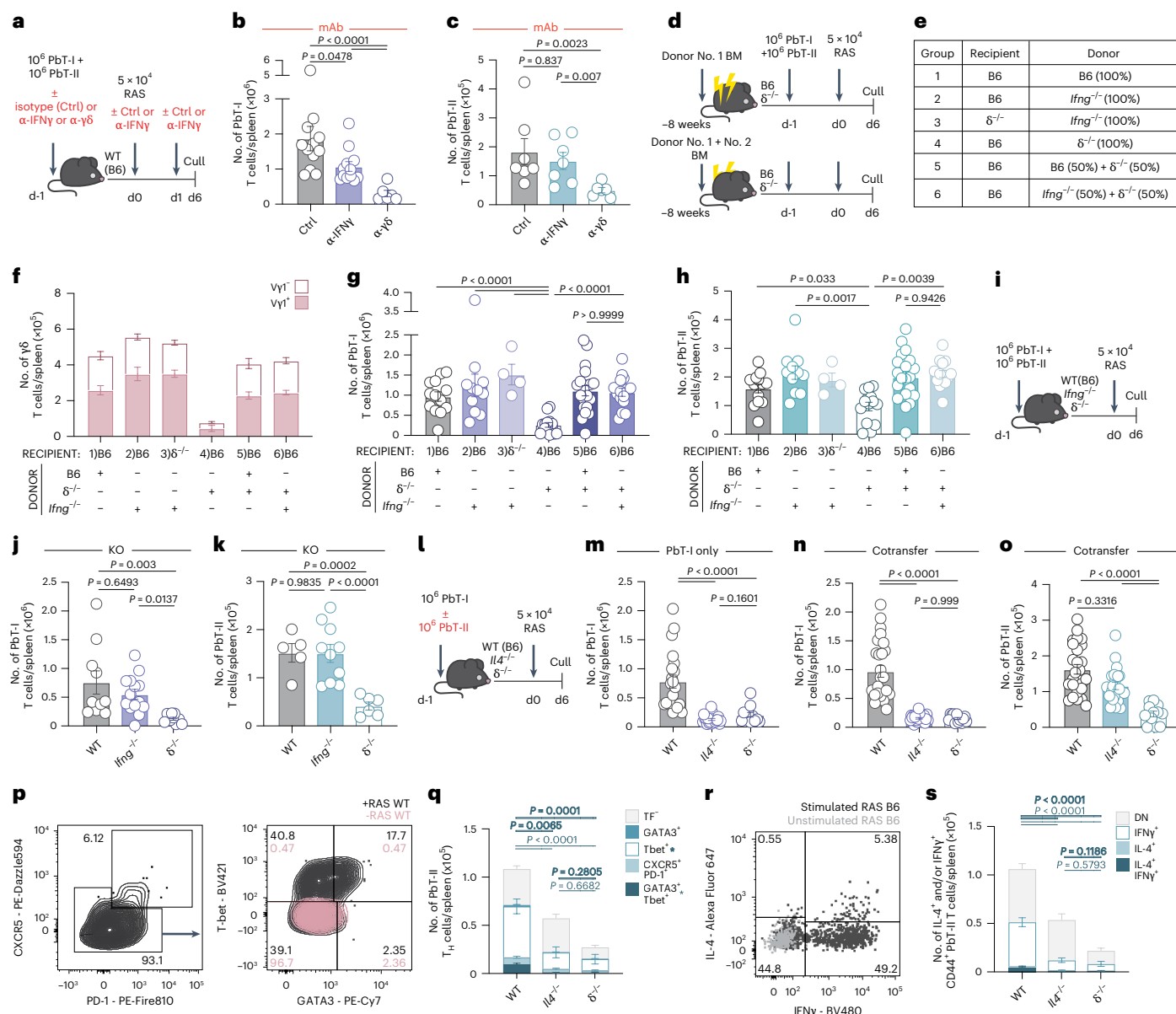

**Fig. 4 | IL-4 is required for the CD8+ T cell response to RAS vaccination.**
**a**, Experimental design. **b,c**, PbT-I and PbT-II cells were transferred into WT (B6) mice treated with an α-IFNγ blocking antibody (i.p.) or an isotype control (Ctrl) (i.p) or α-pan-γδTCR (i.v.) 1 day before immunization with RAS. Shown is the number of PbT-I cells (Ctrl, $n = 12$; α-IFNγ, $n = 12$; α-γδ, $n = 6$) (**b**) and PbT-II cells (Ctrl, $n = 7$; α-IFNγ, $n = 7$; α-γδ, $n = 6$) (**c**) in the spleen on day 6 post-vaccination. **d**, Experimental design. **e–h**, Single and mixed BM chimeras were prepared using BM from WT, δ−/−, and *Ifng*−/− donors as indicated. Eight weeks later, chimeras were given PbT-I and PbT-II cells 1 day before RAS vaccination and then analyzed 6 days later. **e**, Chimera groups relating to **f–h**. **f**, Spleen γδ T cell numbers in chimeras separated by Vγ1 expression (Group 1, $n = 13$; Group 2, $n = 11$; Group 3, $n = 4$; Group 4, $n = 14$; Group 5, $n = 20$; Group 6, $n = 15$). **g,h**, Splenic PbT-I (**g**) and PbT-II (**h**) cell numbers. **i**, Experimental design. **j,k**, WT, *Ifng*−/− and δ−/− mice received PbT-I and PbT-II cells before immunization with RAS. **j,k**, Number of PbT-I cells (**j**) in the spleen of WT ($n = 10$), *Ifng*−/− ($n = 14$) and δ−/− ($n = 9$) mice or PbT-II cells (**k**) in WT ($n = 5$), *Ifng*−/− ($n = 10$) and δ−/− ($n = 6$) mice. **l**, Experimental design. **m–o**, PbT-I

cells were transferred into WT, *Il4*−/− or δ−/− mice 1 day before immunization. An additional cohort of *Il4*−/− mice received both PbT-I and PbT-II cells. **m**, Number of PbT-I cells in the spleen of WT ($n = 21$), *Il4*−/− ($n = 20$) or δ−/− ($n = 11$) mice, 6 days post-vaccination. **n,o**, Numbers of PbT-I cells (**n**) and PbT-II cells (**o**) in the spleen of WT ($n = 26$), *Il4*−/− ($n = 22$) or δ−/− ($n = 12$) mice that received both PbT-I and PbT-II cells. **p**, Transcription factor gating in the spleen. **q**, Number of spleen CD44+ PbT-II cells expressing no transcription factors tested (TF-), GATA3 or Tbet alone, or co-expressing CXCR5 and PD-1, or GATA3 and Tbet from WT ($n = 16$), *Il4*−/− ($n = 9$) or δ−/− ($n = 7$) mice. Splenocytes at day 6 post-vaccination were restimulated ex vivo with PMA/ionomycin for 5 h to assess cytokine production. **r**, IL-4 and IFNγ co-expression in spleen CD44+ PbT-II cells from WT ($n = 16$), *Il4*−/− ($n = 9$) or δ−/− ($n = 7$). **s**, Number of IL-4- and/or IFNγ-expressing CD44+ PbT-II cells. Data were pooled from three independent experiments where points represent individual mice and bars represent mean. Flow cytometry plots are representative. Error bars indicate mean + s.e.m. Data were log-transformed and compared using an ordinary one-way ANOVA.

could not rescue the antigen-specific CD8+ T cell response. CD4+ T cells, therefore, are not the crucial source of IL-4 for CD8+ T cell accumulation following RAS vaccination.

To assess the effect of IL-4 on the T helper phenotype, we examined IFNγ and IL-4 secretion and transcription factor expression in PbT-II

cells. Very few PbT-II cells isolated from vaccinated B6 mice differentiated into Tfh phenotype cells, based on expression of CXCR5 and PD-1, with no difference in the number or proportion of Tfh cells detected in the absence of IL-4 (Fig. 4p,q and Extended Data Fig. 4i). In contrast, larger numbers and proportions of PbT-II cells expressed either Tbet

or both Tbet and GATA3, suggesting a mixed Th1 and Th1/2 phenotype, with the latter dependent on both IL-4 and γδ T cells (Fig. 4p,q and Extended Data Fig. 4i). Cytokine expression analysis showed that PbT-II cells fell into three categories: IFNγ⁺ only, IFNγ⁺/IL-4⁺ or double negative (DN) (Fig. 4r,s). Strikingly, PbT-II cells isolated from δ⁻/⁻ or Il4⁻/⁻ mice did not produce IL-4 and showed a reduction in IFNγ (Fig. 4s and Extended Data Fig. 4j), suggesting that both IL-4 and γδ T cells were required for CD4⁺ T cells to produce IL-4. Collectively, these data show that CD4⁺ T cells proliferate in response to RAS vaccination in the absence of IL-4 but are impaired in their differentiation into IL-4 producing effectors.

### γδ T cells produce IL-4 in experimental and clinical malaria

To determine if γδ T cells provide IL-4 early in the response to RAS vaccination in mice, we took two experimental approaches. First, IL-4 protein measured in the spleen of RAS-vaccinated TCRα⁻/⁻ mice at 6 h was significantly increased only when γδ T cell function was intact (Fig. 5a). Secondly, 4C13R dual IL-4/IL-13 reporter mice that were RAS vaccinated and injected with α-ARTC2 antibody to protect the cytokine-secreting cells[33] (Fig. 5b) showed an increase in the total number of γδ (Fig. 5c) and a specific increase in the number (Fig. 5d,e) and proportion (Fig. 5d,f) of Vγ1⁺ IL-4 producing γδ T cells at 23 h. The γδ T cell activation seen in the 4C13R mirrored that observed in B6 mice in previous experiments (Extended Data Fig. 5a–d and Fig. 2). We did not detect an increase in IL-4 production by Vγ1⁻ γδ T cells (Fig. 5g–i) or an increase in IL-4 production by γδ T cells isolated from the liver at either timepoint (Extended Data Fig. 5e–h). Splenic Vγ1⁺ γδ T cells therefore produce IL-4 in response to RAS vaccination in vivo.

To determine if γδ T cell-derived IL-4 had a functional impact on PbT-I T cell accumulation, we generated mixed BM chimeras (Fig. 5j,k). Eight weeks after reconstitution, the γδ T cell compartment in the spleen (Fig. 5l) and the liver (Extended Data Fig. 5i) reconstituted as expected. Six days after RAS vaccination, impaired PbT-I cell accumulation in the spleens of Il4⁻/⁻+δ⁻/⁻→B6 chimeras (group 6) was observed when compared to B6 + δ⁻/⁻→B6 chimeras (group 5) (Fig. 5m). These data demonstrated that γδ T cells were the important source of IL-4, as B6 + δ⁻/⁻→B6 chimeras contained γδ T cells that could produce IL-4, whereas the Il4⁻/⁻+δ⁻/⁻→B6 chimeras lacked these cells. There was a small but significant increase in PbT-I accumulation when δ⁻/⁻→B6 (Group 4) chimeras were compared with Il4⁻/⁻+δ⁻/⁻→B6 (Group 6) chimeras, suggesting an additional small contribution of γδ T cells that was not IL-4

dependent, but the nature of this contribution is yet to be determined. Nonetheless, these data strongly suggest that γδ T cells are the essential source of IL-4 for the initiation of an effective CD8⁺ T cell response to RAS vaccination.

To further confirm that γδ T cells are the crucial initial source of IL-4, we reconstituted δ⁻/⁻ mice with splenic γδ T cells from either WT or Il4⁻/⁻ or Il4ra⁻/⁻ donors and tested whether these cells could rescue the PbT-I response. TCRδ-deficient mice received either splenic γδ T cells or no γδ T cell transfer (Fig. 5n). Six days after RAS vaccination (8 days after γδ T cell transfer), γδ T cells were recovered from the spleens demonstrating that the transfer was effective (Fig. 5o). As hypothesized, B6-derived but not Il4⁻/⁻-derived γδ T cells supported PbT-I cell accumulation in the spleen in response to RAS vaccination, confirming that these cells provided IL-4 in this vaccination setting (Fig. 5p). Support for PbT-I T cell accumulation by Il4ra⁻/⁻ γδ T cells (Fig. 5p) implied that γδ T cells do not need to sense IL-4 to provide it, suggesting these cells are the crucial initiators of IL-4 production.

In humans, the population of γδ T cells that expand and correlate with protection following RAS vaccination are Vγ9⁺Vδ2⁺ γδ T cells, the most abundant γδ T cell population in the peripheral blood[16,17,19,20] and also found in the spleen[34]. We therefore asked if Vγ9⁺Vδ2⁺ γδ T cells also produce IL-4, or other inflammatory cytokines, in the context of natural infection. Peripheral blood mononuclear cells (PBMCs) from individuals with an uncomplicated P. falciparum malaria infection were PMA-stimulated and cytokine production quantified (Fig. 5q). IL-4 production by Vδ2⁺ γδ T cells was higher in individuals with malaria compared to either healthy, currently uninfected individuals from the same area or healthy, malaria-naive Australians (Fig. 5r and Extended Data Fig. 6a–c). In contrast, IFNγ and TNF production was comparable or reduced between malaria-infected and healthy controls (Fig. 5r). CXCR5 and CCR7 are associated with lymphoid tissue homing in Vγ9⁺Vδ2⁺ γδ T cells[34,35]. Analysis of IL-4 expression showed that this cytokine was largely produced by cells expressing CXCR5 and CCR7, two chemokine receptors that assist with entry to lymphoid tissues including the spleen (Extended Data Fig. 6c). This was different from the pattern seen in IFNγ-producing Vδ2⁺ T cells with a higher proportion of IFNγ producers within the CXCR5⁻ CCR7⁻ subset (Extended Data Fig. 6c). These data suggest that during human malaria initiated by sporozoite infection, Vδ2⁺ T cells that may have spleen homing properties also have the capacity to produce IL-4.

**Fig. 5 | γδ T cell-derived IL-4 directly signals to CD8⁺ T cells and DCs.** TCRα⁻/⁻ mice were injected with α-pan-γδTCR (or isotype, Ctrl) (i.v.) 1 day before RAS vaccination. After 6 h, IL-4 concentration was assessed. **a**, Concentration per gram of spleen in naive (n = 11), vaccinated Ctrl (n = 10) or vaccinated α-pan-γδTCR (n = 11) TCRα⁻/⁻ mice. **b**, Experimental design (**c–i**). B6.4C13R mice were vaccinated with RAS. Naive (23 h, n = 6; 44 h, n = 7) and vaccinated (23 h or 44 h, n = 8) mice were treated with α-ARTC2 (i.v.) and then culled at 23 or 44 h post-RAS. **c**, IL-4⁺ (AmCyan⁺) γδ T cell numbers in enriched spleen. **d**, IL-4 expression in splenic Vγ1⁺ γδ T cells. **e**, IL-4⁺ (AmCyan⁺) Vγ1⁺ γδ T cell numbers in enriched spleen. **f**, IL-4⁺ (AmCyan⁺) cell frequency in splenic Vγ1⁺ γδ T cells. **g**, IL-4 expression in splenic Vγ1⁻ γδ T cells. **h**, IL-4⁺ (AmCyan⁺) Vγ1⁻ γδ T cell numbers in enriched spleen. **i**, IL-4⁺ (AmCyan⁺) cell frequency in splenic Vγ1⁻ γδ T cells. **j**, Experimental design (**k–m**). Chimeras were given PbT-I cells 1 day before RAS vaccination and then analyzed 6 days later. **k**, Chimeras were reconstituted with B6, δ⁻/⁻, Il4⁻/⁻ donor cells (Group 1, n = 20; Group 2 or 3, n = 10; Group 4, n = 6; Group 5 or 6, n = 20). **l**, Spleen γδ T cell numbers at day 6 post-RAS. **m**, PbT-I cell numbers in the spleen. **n**, Experimental design. **o,p**, δ⁻/⁻ mice received γδ T cells from WT, Il4⁻/⁻ or Il4ra⁻/⁻ donors 1 day before transfer of PbT-I cells. Recipient mice were vaccinated with RAS and analyzed 6 days later. **o**, Quantified splenic γδ T cells 6 days after RAS vaccination in δ⁻/⁻ mice receiving γδ T cells from WT (+γδ WT) (n = 17), Il4⁻/⁻ (+γδ Il4⁻/⁻) (n = 12), Il4ra⁻/⁻ (+γδ Il4ra⁻/⁻) (n = 8), or no γδ T cells (no transfer) (n = 13). **p**, PbT-I cell counts in the spleen 6 days after RAS vaccination. **q**, PBMCs were collected from malaria-naive healthy controls

(C, n = 5), malaria-endemic healthy controls (EC, n = 4) and malaria-endemic patients with acute malaria infection (acute, n = 7). **r**, Frequency of IL-4, IFNγ- and TNF-producing Vδ2⁺ γδ T cells after PMA/ionomycin stimulation. **s**, Experimental design. **t,u**, CRISPR-Cas9 ablated sgCd19 (control) or sgIl4ra PbT-I cells were transferred into recipients 1 day before RAS vaccination. **t**, Surface IL-4Rα expression on sgCd19 (control) or sgIl4ra PbT-I cells at day 6. **u**, Numbers of PbT-I cells in the spleen of mice receiving sgCd19 (n = 9) or sgIl4ra (n = 9) PbT-I cells or sgCd19 cells with α-pan-γδTCR (n = 6) at day 6. **v**, Experimental design. **w,x**, Batf3⁻/⁻ recipients received sgCd19 (control) or sgIl4ra gene-edited CD24⁺ cDC1 2 days before T cell transfer. Recipient mice were vaccinated with RAS and analyzed 6 days later. **w**, Spleen PbT-I cell numbers in Batf3⁻/⁻ mice that received sgCd19 (control) (n = 9) or sgIl4ra cDC1s (n = 9) or no cDC1s (- transfer) (n = 10). **x**, Splenic cDC1 counts six days after RAS vaccination. **y**, IL-4 acts directly on CD8⁺ T cells and cDC1s for CD8⁺ T cell accumulation. Data show two (**a–i,s–x**), four (**l–p**) or three (**c,d,j,k**) independent experiments where points represent individual mice and bars represent mean. Flow cytometry plots are representative. Box and whisker plot center line represents the median; box limits indicate the upper and lower quartiles and whiskers extend to 1.5x the interquartile range. Error bars indicate mean + s.e.m. Data were log-transformed and compared using an ordinary one-way ANOVA (**a,l–p,u–x**), multiple unpaired two-tailed Welch's t-tests and Holm–Šidák multiple comparisons correction (**c–i**) or multiple unpaired two-tailed t-test or (TNF) Mann–Whitney U test after Shapiro–Wilk normality testing (**r**). h, hours.

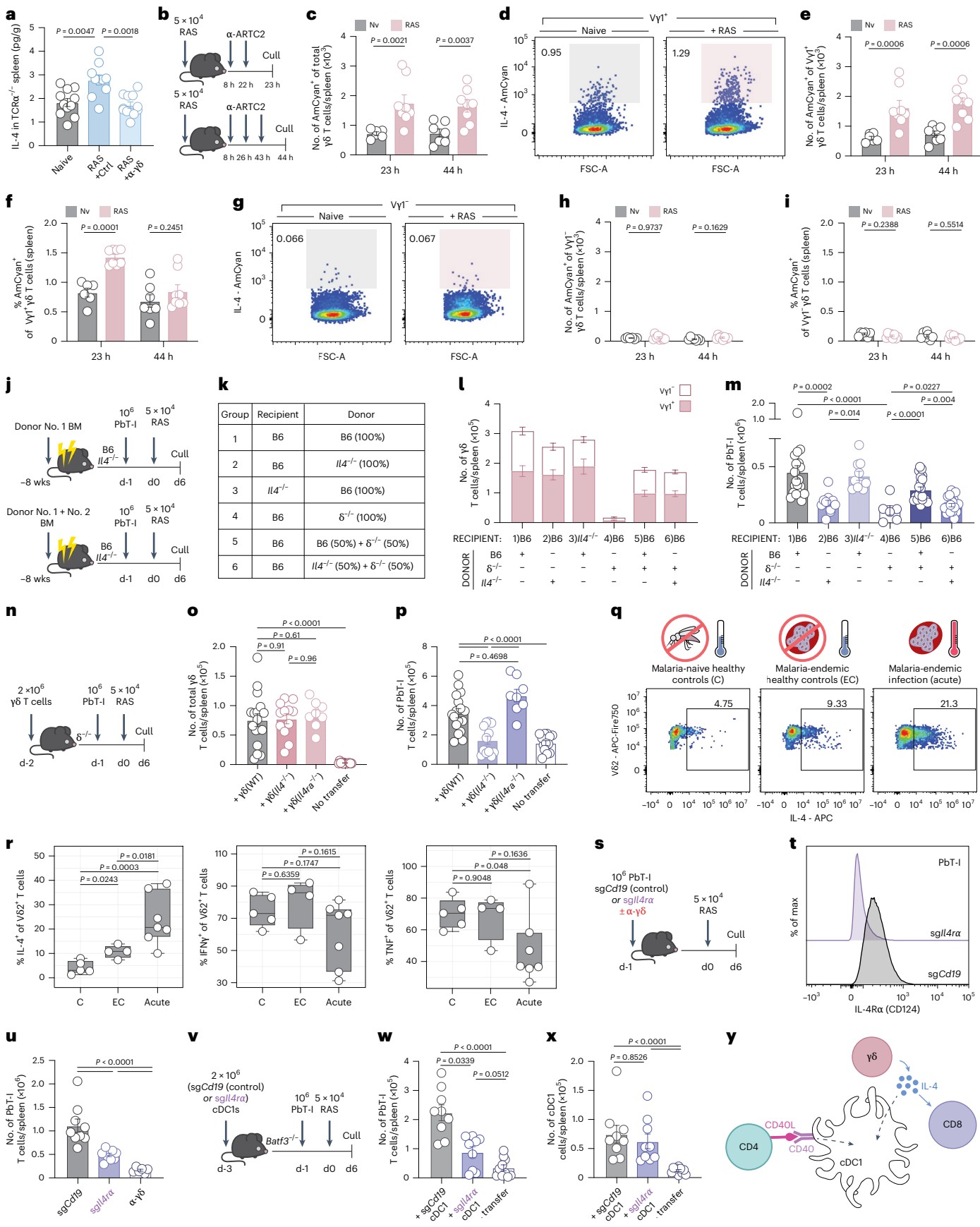

## IL-4 signals both CD8+ T cells and cDC1s to initiate the CD8+ response

IL-4 is a prototypical T helper 2 cytokine that has potent effects on CD4+ T cell differentiation and function[36] but less well-studied impacts on CD8+ T cells. We therefore asked if the RAS-induced γδ T cell-derived IL-4 acted directly on the responding CD8+ T cells. CRISPR-Cas9 deletion of the IL-4Rα gene (*Il4ra*) (or *Cd19* as a control) from naive PbT-I cells before transfer and immunization with RAS (Fig. 5s,t) showed that direct IL-4Rα signaling was important for PbT-I cell accumulation in the spleen (Fig. 5u).

PbT-I cells lacking the IL-4R still retained a modest capability to accumulate when compared to a complete failure in the absence of γδ T cells (Fig. 1a) or IL-4 itself (Fig. 4m), suggesting an additional activity that contributes to CD8+ T cell accumulation. To investigate whether this involved direct signaling of cDC1 by IL-4, we developed an in vivo model for cDC1 manipulation. Donor cDC1 were expanded in vivo in B6 mice[37] before enrichment, CRISPR-Cas9-mediated deletion of either *Cd19* or *Il4ra* and transfer into *Batf3*−/− mice for replenishment of the cDC1 pool (Fig. 5v). Splenic PbT-I cell accumulation was impaired when cDC1 did not express IL-4Rα (sg*Il4ra*) (Fig. 5w), suggesting direct IL-4 signaling in cDC1. Numbers of cDC1 recovered were equivalent between groups (Fig. 5x and Extended Data Fig. 6d). Collectively, these experiments demonstrate that IL-4 acts directly on DC and on CD8+ T cells for optimal CD8+ T cell expansion in the context of RAS vaccination (Fig. 5y).

## IL-4 and IFNγ synergize with CD40 to drive IL-12 production by cDC1s

To identify the molecules induced by IL-4 in cDC1s, gene expression was assessed in vitro following stimulation with media alone, αCD40, IFNγ, IL-4, IFNγ+IL-4, αCD40 + IFNγ, αCD40 + IL-4 or αCD40 + IL-4 + IFNγ. Expanded cDC1s were sort purified and then cultured in each condition before isolation of RNA at 4 h (Fig. 6a).

Analysis of sequenced RNA revealed a marked transcriptomic shift in the presence of IL-4 that was augmented by the addition of αCD40 (Fig. 6b,c), a shift further augmented by the addition of IFNγ (Fig. 6b). Specifically, there were 1,081 upregulated and 1,395 downregulated genes in response to αCD40 + IL-4 when compared to αCD40 alone (Fig. 6c and Supplementary Table 1). This was substantially increased by the addition of IFNγ, with 2,277 genes upregulated and 2,037 genes downregulated in the presence of αCD40 + IL-4 + IFNγ (Fig. 6c and Supplementary Table 1). *Il12a* was the most upregulated gene when comparing αCD40 with αCD40 + IL-4 + IFNγ (LFC 13.14, FDR $9.03 \times 10^{-8}$) and the third most upregulated gene when comparing αCD40 + IL-4 with αCD40 alone (LFC 7.055, FDR $8.97 \times 10^{-6}$) (Fig. 6d,e). This gene encodes the p35 subunit of IL-12, which, when combined with the p40 subunit (which is constitutively expressed by cDC1), makes bioactive IL-12. Efficient upregulation of *Il12a* was dependent on the combination of CD40, IFNγ and IL-4 signals (Fig. 6f).

IL-12 blockade in vivo (Fig. 6g) showed significant impairment in PbT-I accumulation (Fig. 6h). To confirm that cDC1 were the crucial source of IL-12, *Batf3*−/− mice were reconstituted with cDC1 from *Il12b*−/− (*Il12p40*−/−) mice (Fig. 6i,j), showing that PbT-I cell accumulation was impaired when cDC1 were unable to produce bioactive IL-12 (Fig. 6k). This supports a model where CD40 signaling synergizes with IL-4 and IFNγ in cDC1 to induce IL-12 that is crucial for the accumulation of CD8+ T cells in response to RAS vaccination (Fig. 6l).

## IL-4 promotes CD8+ T cell expansion by increasing IL-12R expression

IL-4 and IL-12 were both essential for enhanced accumulation of CD8+ T cells, suggesting synergistic actions in CD8+ T cell expansion. To isolate the effects of the cytokines, we first examined the impact of IL-4 and IL-12 on CD8+ T cell expansion in vitro.

PbT-I cells activated by peptide-coated antigen presenting cells showed limited cell growth in the absence of exogenous cytokine (Fig. 7a). Addition of either IL-4 or IL-12 only modestly impacted the number of cells recovered (Fig. 7b,c), but addition of both IL-4 and IL-12 resulted in significantly higher cell recovery at day 6 (Fig. 7d), suggesting that IL-4 and IL-12 act synergistically to increase the T cell response in vitro.

To explore whether the synergy between IL-4 and IL-12 was mediated by changes in cytokine sensitivity resulting from alterations in receptor expression, we assessed IL-4R and IL-12R expression on T cells at day 4 of culture. Compared to addition of IL-4 alone, a combination of IL-4 and IL-12 induced a small increase in IL-4R (Fig. 7e). More substantially, the addition of IL-4 + IL-12 increased expression of IL-12R over either cytokine alone (Fig. 7f), suggesting that the combination of IL-4 and IL-12 amplifies the sensitivity of CD8+ T cells to IL-12 signaling through increased expression of IL-12R. Further, in vitro activated OT-I T cells stimulated with IL-4 and IL-12 were maintained in greater numbers when transferred into mice (Fig. 7g).

To extend these findings in vivo, we first confirmed that naive PbT-I cells expressed detectable levels of IL-4Rα (Fig. 7h,i). PbT-I cells adoptively transferred into either B6, *Il4*−/− or δ−/− mice and exposed to RAS vaccination only showed full upregulation of IL-4Rα and IL-12R in the presence of IL-4 or γδ T cells (Fig. 7j–l), supporting our in vitro findings and implicating γδ T cells as the mediator.

## γδ T cell-derived IL-4 and cDC1-derived IL-12 promote liver T_RM cells

To confirm that removal of both IL-4 and IL-12 in vivo would recapitulate the lack of γδ T cells and therefore inhibit the formation of protective liver T_RM cells (Fig. 1), we blocked IL-4 or IL-12, or both, in the context of RAS vaccination and then examined T_RM numbers in the liver 29 days later (Fig. 7m). Enumeration of the number of liver T_RM cells showed that IL-4 had a dominant effect on liver T_RM formation, but blockade of both cytokines phenocopied blockade of γδ T cells (with α-γδ Ab) in the spleen (Fig. 7n,o) and liver (Fig. 7p,q). These data show that γδ T cells drive the response to RAS via delivery of IL-4 to DCs, which in turn acts with IFNγ to drive IL-12 production. The γδ T cell-derived IL-4 and DC-derived IL-12 then synergize to enhance the expansion of CD8+ T cells (Fig. 7r), a proportion of which will differentiate into protective liver T_RM cells.

## Discussion

For full immunogenicity, DCs must be effectively activated, either via strong PRR signaling or through CD4+ T cell help via CD40L[1]. Here, we showed that the priming of CD8+ T cells to *Plasmodium* sporozoite antigens depends on an additional layer of DC activation via IL-4, which is produced by γδ T cells. Here, IL-4 has a dual role: 1) when synergized with CD40 and IFNγ signals in DCs, it promotes enhanced production of bioactive IL-12 and 2) by directly signaling CD8+ T cells to increase IL-12R, it increases their sensitivity to IL-12, thereby promoting expansion. In this vaccination setting, IL-4 is produced by Vγ1+ γδ T cells in the spleen, which become activated within the first 24 h of injection of sporozoites.

The absolute requirement for γδ T cells for successful RAS vaccination in mice[16] and the correlations with effective RAS vaccination in humans[16,17,19,20] suggest that *Plasmodium* sporozoites are not a sufficient stimuli to push DCs over a required activation threshold and an additional IL-4 signal in combination with IFNγ and CD40 is required. In help-dependent infections, the CD40L signal from CD4 T cells is usually sufficient for DCs to reach the required activation threshold for effective CD8 T cell priming. The absolute requirement for IL-4-producing γδ T cells detected here likely derives from RAS being a very weak immunogen, thereby requiring additional signals for sufficient DC activation. The small number of activated IL-4-producing γδ T cells observed is consistent with the expected minimal size of the initial response to RAS given that so far, we have been unable to detect DC activation ex vivo, despite these cells clearly being involved.

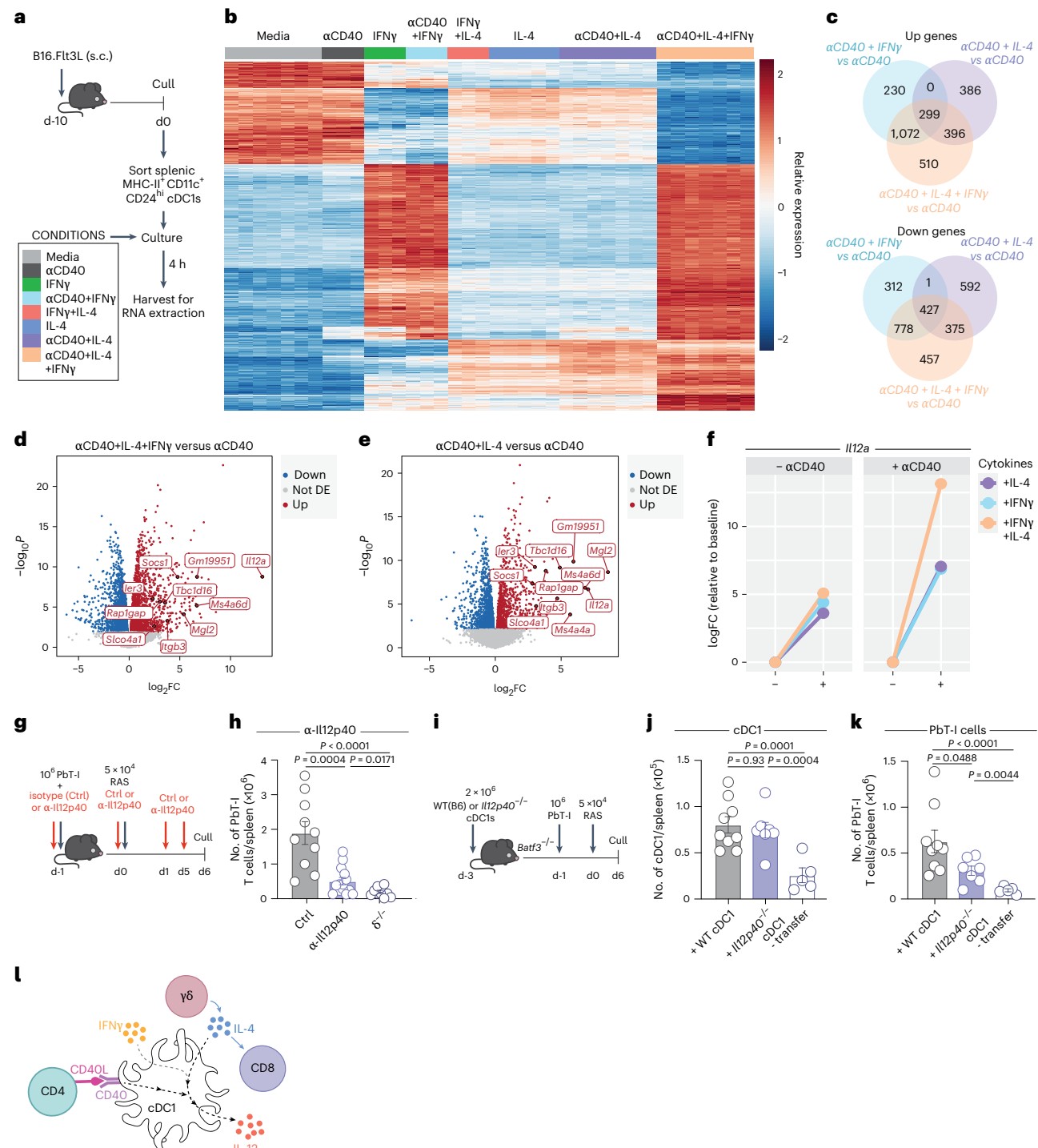

**Fig. 6 | CD40 and IL-4 drive a unique gene expression profile in cDC1s.**
**a**, Experimental design. **b**–**f**, Sorted cDC1s were cultured for 4 h in media
alone ($n = 7$), αCD40 ($n = 3$), αCD40 + IL-4 ($n = 7$), αCD40 + IFNγ ($n = 3$),
αCD40 + IL-4 + IFNγ ($n = 7$), IL-4 ($n = 5$), IFNγ ($n = 3$) or IL-4 + IFNγ ($n = 3$). s.c.,
subcutaneous. **b**, Heatmap of differentially expressed genes (DEGs) between
media alone and αCD40 + IL-4 + IFNγ, across the different stimulation conditions.
**c**, Upregulated (Up) and downregulated (Down) DEGs between cDC1s stimulated
with αCD40 + IL-4, αCD40 + IFNγ or αCD40 + IL-4 + IFNγ relative to αCD40
only. **d**,**e**, Volcano plots of DEGs between αCD40 only and αCD40 + IL-4 + IFNγ
(**d**) or αCD40 + IL-4 (**e**) stimuli. **f**, Log fold change (FC) in *Il12a* expression in
cells subjected to IL-4 or IFNγ, or a combination of both relative to baseline
(no cytokine) when in the absence (left) or presence (right) of αCD40.
**g**, Experimental design. **h**, WT (B6) or δ$^{-/-}$ ($n = 10$) mice received PbT-I cells 1 day
before RAS vaccination and WT mice were treated with either α-IL-12p40 (i.p.)

($n = 14$) or an isotype control mAb (Ctrl) (i.p.) ($n = 10$). **h**, Number of PbT-I cells
in the spleen at day 6 post-RAS vaccination. **i**, Experimental design. **j**,**k**, CD24$^+$
cDC1 from WT (B6) or *Il12p40*$^{-/-}$ mice were transferred into *Batf3*$^{-/-}$ recipients
followed by PbT-I cells 2 days later. Mice were vaccinated with RAS and analyzed
at day 6 post-RAS. **j**, Splenic cDC1 counts in *Batf3*$^{-/-}$ mice that received WT
($n = 9$), *Il12p40*$^{-/-}$ ($n = 7$) cDC1 or no cDC1s (- transfer) ($n = 5$). **k**, Number of PbT-I
cells in spleens of *Batf3*$^{-/-}$ mice that received WT ($n = 9$), *Il12p40*$^{-/-}$ ($n = 7$) cDC1
or no cDC1s (- transfer) ($n = 5$). **l**, γδ T cell-derived IL-4 acts on cDC1s with IFNγ
to promote IL-12 production that, along with IL-4, is required for CD8$^+$ T cell
accumulation. Data show seven (**a**–**f**) or two (**h**,**j**,**k**) independent experiments
where points show pooled biological replicates (**f**) and individual mice (**h**,**j**,**k**).
Error bars indicate mean + s.e.m. Data were log-transformed and compared by an
ordinary one-way ANOVA.

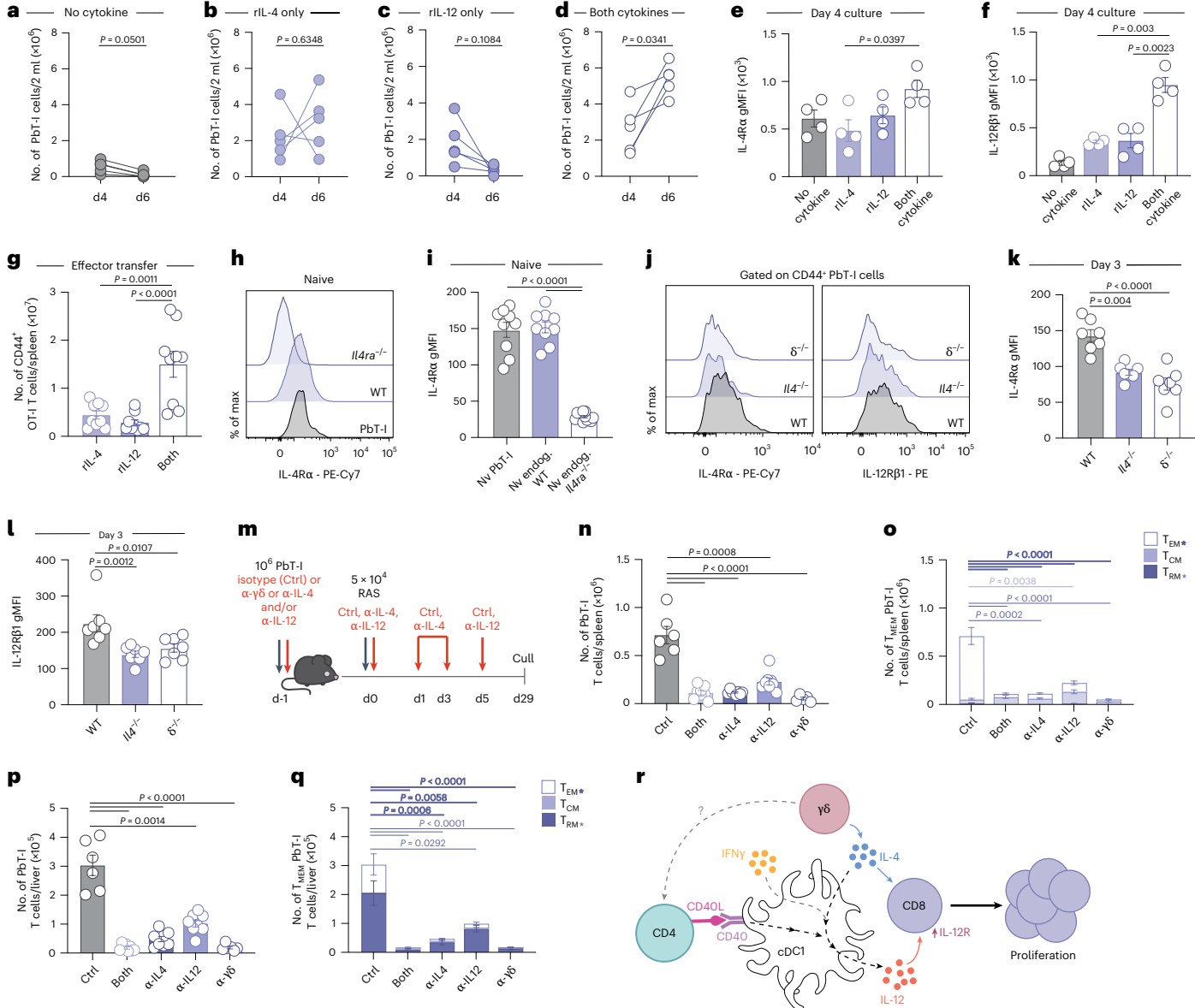

**Fig. 7 | IL-4 and IL-12 synergize to promote expansion of CD8+ T cells. a–d**, PbT-I cells were peptide-activated in vitro in media alone (no cytokine) (*n* = 5) (**a**) or in the presence of 60 ng ml$^{-1}$ rIL-4 (rIL-4 only) (*n* = 5) (**b**), 10 ng ml$^{-1}$ rIL-12 (rIL-12 only) (*n* = 5) (**c**) or both rIL-4 and rIL-12 (both cytokines) (*n* = 5) (**d**). **e,f**, IL-4Rα (**e**) and IL-12Rβ1 (**f**) surface expression on activated PbT-I cells 4 days after culture in the presence of different cytokines (*n* = 4). OT-I cells were activated in vitro in the presence of 60 ng ml$^{-1}$ rIL-4 (*n* = 9), 10 ng ml$^{-1}$ rIL-12 (*n* = 9), or both rIL-4 and rIL-12 (both) (*n* = 9) and then transferred into recipient mice at day 4 of culture. **g**, Number of OT-I cells in the spleen 3 days after effector transfer. **h**, Surface IL-4Rα expression on endogenous naive CD8+ T cells in *Il4ra*$^{-/-}$ (top), WT (B6) mice (middle), or on naive PbT-I cells transferred into naive WT recipients (bottom). **i**, Geometric mean fluorescence intensity (gMFI) of surface IL-4Rα expression on endogenous naive CD8+ T cells in *Il4ra*$^{-/-}$ (*n* = 12), WT mice (*n* = 9) or on naive PbT-I cells transferred into naive WT recipients (*n* = 9). B6, *Il4*$^{-/-}$ and δ$^{-/-}$ mice received 10$^6$ PbT-I cells 1 day before RAS vaccination. PbT-I cells were then assessed 3 days later. **j**, Surface IL-4Rα (left) or IL-12Rβ1 (right) expression on PbT-I cells at day

3. **k,l**, gMFI of surface IL-4Rα (**k**) or IL-12Rβ1 (**l**) expression on PbT-I cells at day 3 post-vaccination (*n* = 7). **m**, Experimental design. **n–q**, Mice received PbT-I cells 1 day before RAS vaccination and were treated with an isotype control (Ctrl) (i.p.) (*n* = 6), α-pan-γδTCR (i.v.) (*n* = 5), α-IL-12p40 (i.p.) (*n* = 8), α-IL-4 (i.p.) (*n* = 8) or both α-IL-4 and α-IL-12p40 (*n* = 8). **n,p**, Number of PbT-I cells in the spleen (**n**) and liver (**p**), 29 days after vaccination. **o,q**, Frequency of T$_{CM}$, T$_{EM}$ and T$_{RM}$ within the CD44+ PbT-I cell compartment in the spleen (**o**) and liver (**q**) 29 days after RAS vaccination. **r**, γδ T cell-derived IL-4 acts on cDC1s, along with CD40 signals and IFNγ, to promote IL-12 production, which, together with IL-4, signals directly on CD8+ T cells to drive proliferation and therefore enhance liver T$_{RM}$ formation. Data show five (**a–d**), four (**e,f**), three (**g**) or two (**i–k,h–q**) independent experiments where points show biological replicates (**a–f**) and individual mice (**e–q**). Histogram plots are representative. Error bars indicate mean + s.e.m. Data were log-transformed then compared using a paired two-tailed *t*-test (**a–d**) or an ordinary one-way ANOVA (**e–q**).

Our findings likely extend beyond the context of RAS vaccination to other scenarios involving poor immunogens. In such cases, the sensing of infection or cellular damage by innate or innate-like cells may trigger IL-4 production, thereby enabling DCs to effectively prime protective immune responses. One such innate-like T cell population are invariant natural killer T cells, which are known to produce IL-4 in response to lipid ligands[38]. These ligands can be used as adjuvants to boost the response to RAS[39] and heat-killed sporozoites[40], likely via enhanced IL-4 production. However, the need for natural killer T cell-derived cytokines on top of CD4+ T cell help in a four-cell paradigm, as we have shown here for γδ T cells, has not been described.

Correlations between Vγ9+Vδ2+ γδ T cells expansion and activation with protection following RAS vaccination in humans[16,17,19,20] led us to examine this population for cytokine production following natural infection showing an increase in the proportion of Vγ9+Vδ2+ γδ T cells that produced IL-4 in people from an endemic setting, with a further increase in people experiencing acute infection. Of note, this pattern was not mirrored for IFNγ or TNF following restimulation, suggesting that Vγ9+Vδ2+ γδ T cells may perform similar biological functions as the Vγ1+ population in mice, but further analysis is needed.

Here, we identified that in mice, the Vγ1+ subset of γδ T cells provide a crucial function by supplying IL-4 for initiation of the CD8+ T cell response. As we used a model of intravenous vaccination with RAS, activation of γδ T cells was observed in the spleen, liver and liver dLN, consistent with the antigen distribution in this model. Despite identifying the specific subset involved and resolving the timing of activation, we have not yet successfully identified the trigger for Vγ1+ γδ T cell activation. Identification of the activation triggers for γδ T cells remains a challenge for the field[41,42], but we hypothesize that the cells are responding to a parasite or parasite induced self-ligand exposed by the infectious process. In mice, γδ T cell-derived IL-17 (ref. [43]) and IFNγ[44], induced by liver-stage parasites, have been shown to play important roles in the pathology of *Plasmodium* blood-stage infection. Here, we have shown that the initiation of the CD4+ T cell response to liver-stage parasites is also dependent on a factor provided by γδ T cells but have not yet identified the nature of this factor. These concepts highlight that γδ T cells play a varied and substantial role in malaria disease.

Previous studies linked the presence of γδ T cells with CD8+ DC (now cDC1) accumulation in the liver of RAS-vaccinated mice and the generation of hepatic CD8+ T cell memory[16]. Although we have not directly investigated these hepatic DCs, our data show that splenic cDC1 collaborate with γδ T cells to drive efficient priming of splenic CD8+ T cells. It is possible that γδ T cell-induced IL-4 and IL-12 leads to inflammation in the liver and the accumulation of CD8+ cDCs, commensurate with the accumulation of CD8+ T cells in the liver. However, as liver DC accumulation was observed late in the response (>60 days post-immunization), it is unlikely to contribute to the γδ T cell-dependent priming observed in the current study.

Before the discovery of T_RM cells, a role for IL-4 in RAS vaccination (for *P. yoelii* in BALB/c mice[12,13]) was identified. CD4+ T cells, rather than γδ T cells, were implicated as the source of IL-4 (refs. [12,13]). We speculate that a lack of CD40L signaling in the absence of CD4+ T cells, rather than a lack of CD4+ T cell-derived IL-4, was likely responsible for the phenotype observed. However, our data suggest that γδ T cell-derived IL-4 drives production of IL-4 by CD4+ T cells. Nevertheless, we have shown that the initial and most important source of IL-4 is γδ T cell derived.

Notably, we have shown IL-4 and IL-12 synergistically enhance CD8+ T cell expansion by increasing expression of IL-12R. This runs counter to CD4+ T cell responses, where IL-4-induced GATA3 activation is reported to suppress IL-12 signaling[45]. Synergistic effects of IL-4 and IL-12 as reported for CD4+ T cells[46] and human T cells[47] require further investigation. Here, we show that in addition to enhancing CD8+ T cell sensitivity to IL-12, IL-4 amplified αCD40- and IFNγ-driven IL-12 production in DCs. The combined effects of IL-4 and IFNγ on IL-12p70 production by DCs have been described previously[48], but the specific DC subset that responds to these cytokines in vivo was not investigated. Here, we have demonstrated that the responding population are cDC1, but whether other cDC populations such as cDC2 can also respond to IL-4 or combined IL-4 and IFNγ signals when CD40L is provided remains unknown. Regardless, in the absence of IL-4, the entire CD8+ T cell response to RAS fails, highlighting its significance for weakly stimulating pathogens (or tumors) that cannot intrinsically drive strong CD8+ responses.

Importantly, a newly appreciated role for IL-4 in the context of immunotherapy has recently been highlighted. IL-4 production by B cells was shown to be one mechanism for successful PD-1 immunotherapy[49], and two recent studies demonstrated the importance of an IL-4 driven type 2 response for effective and long-lived CAR T cell or anti-tumor responses[50,51]. These studies demonstrate that IL-4 has therapeutic implications beyond infectious diseases and indicate that γδ T cells should be investigated as a source of IL-4 in these models.

In conclusion, our study reveals that by providing IL-4, Vγ1+ γδ T cells drive the initiation of the response to a poorly immunogenic pathogen, transforming it into a robust trigger for CD8+ T cell immunity and tissue-tropic memory.

## Online content

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

## Methods

### Mice

C57BL/6, B6.SJL-PtprcaPep3b/BoyJ (CD45.1), PbT-I[14] GFP or tdTomato, PbT-II[31] GFP, $Tcrd^{-/-}$[52], Trdc$^{tm1(EGFP/HBEGF/luc)Impr}$ (TCRδ-GDL)[53], Il4$^{tm2Nnt/J}$ (Jax strain # 002496) backcrossed to B6 ($Il4^{-/-}$), $Batf3^{-/-}$[54], $Cd40lg(Cd154)^{-/-}$[55], $Ifng^{-/-}$[56], $Tcra^{-/-}$[57], Il4ra$^{tm1Fbb}$ backcrossed to B6 ($Il4ra^{-/-}$, F. Brombacher)[58], $Il12b(p40)^{-/-}$, and 4C13R[59] experimental mice were sex matched and used at 6-12 weeks of age, or up to 20 weeks for chimeric mice. Mice were allocated to experimental groups without randomization or blinding. Experimental mice were bred in specific pathogen-free conditions and held at the Department of Microbiology and Immunology, The University of Melbourne, Australia, at 20–26 °C, 45–65% humidity on a 12-h day–night light cycle. For sporozoite generation, 4- to 5-week-old Swiss Webster mice were purchased from the Monash Animal Service and held at the School of Botany, University of Melbourne, Australia. All mice were maintained on standard chow ad libitum and experimental work and animal handling conducted in strict accordance with the standards approved by the Animal Ethics Committee at the University of Melbourne (ethics project IDs 27552, 2015168, 20088). TCRδ-GDL[53] mice were provided by I. Prinz (University Medical Center Hamburg-Eppendorf).

### Mosquitoes and parasites

*Plasmodium berghei* ANKA (PbA) WT Cl15cy1 (BEI Resources, NIAID, NIH: MRA-871) or *P. berghei* CS5M, which express the cognate antigen for OT-I T cells[60], and *PbΔmei2* parasites were used. *Anopheles stephensi* mosquitoes (strain STE2/MRA-128 from BEI Resources, The Malaria Research and Reference Reagent Resource Centre) were reared and maintained at 27 °C, 75–80% humidity on a 12-h day/night light cycle in an insectary approved by the Australian Department of Agriculture, Fisheries and Forestry[61]. Larvae were bred in filtered drinking water, refreshed every 3 days and fed with Sera vipan baby fish food (Sera). Eclosed adult mosquitoes were then maintained on 10% sucrose in aluminum cages within an insectary incubator. Naive Swiss Webster mice were injected intraperitoneally (i.p.) with infected RBC from infected donor mice. Three days post-infection, parasitemia was confirmed by Giemsa smear and exflagellation quantified. Adult mosquitoes were then allowed to feed on infected, anaesthetized mice. After 20 to 22 days, sporozoites were hand-dissected from mosquito salivary glands, resuspended in PBS then, if relevant, irradiated with 20 K rads from a gamma $^{60}$Co source[15] (WEHI irradiation facility, Melbourne, Australia). Irradiated, or fresh, sporozoites were injected via the tail vein.

In blood-stage malaria experiments, donor mice were injected i.p. with frozen stabilates of blood-stage *P. berghei* ANKA (PbA) WT Cl15cy1 (BEI Resources, NIAID, NIH: MRA-871). Three to seven days later, mice were bled for parasitemia, the red-blood cells (RBCs) irradiated with 20 K rads[62] from a gamma $^{60}$Co source (WEHI irradiation facility), then injected via the tail vein.

### Generation of *PbΔmei2* parasites

To generate *PbΔmei2* parasites, the mei2 gene region was reversed so that the coding sequence reads left to right. 5′ and 3′ homology flanks were amplified from P*b* genomic DNA with the primers CJ472 + CJ473, and CJ474 + CJ475 (Extended Data Table 2), producing 798 bp and 832 bp amplicons, respectively. CJ473 and CJ474, the 5′ flank reverse and 3′ flank forward primer featured a 23-base multi-cloning-site complementary tail (including a possible guide sequence) which was utilized in a further PCR reaction to generate the 5′, 3′ flank fusion fragment. This was cloned into the CRISPR-Cas9 plasmid pYC_L2 (a gift from A. Vaughan) with the restriction enzymes KpnI and EcoRI. Guide sequences were identified with CHOPCHOP[63] then cloned into separate KO flank containing plasmids with the enzyme Esp3I to generate the mei2KO plasmid pCJ133 and 134 respectively (Guide 3 and 28, respectively). Next, 15 μg of each plasmid was co-transfected into magnet-purified P*b* schizont-stage parasites[64,65] and injected intravenously (i.v.) into Swiss

Webster mice. After 24 h, mice were put on water containing 70 μg ml$^{-1}$ pyrimethamine (Sigma). When mice reached >1% parasitemia, they were euthanized and cryostocks prepared. Genotyping was performed by PCR (primers in Extended Data Table 2) and clonal parasites generated by limiting dilution[66].

### Conventional T cell isolation and adoptive transfer

Naive CD8$^+$ PbT-I, OT-I or CD4$^+$ PbT-II cells were negatively enriched from spleens and lymph nodes. Tissues were passed through a 70 μM filter then red blood cells lysed to prepare a single cell suspension. Cells were labeled with rat monoclonal antibodies specific for MHC Class II (M5-114), Mac-1 (M1/70), Gr-1 (RB6-8C5) and either CD4 (GK1.5) (for PbT-I and OT-I CD8$^+$ T cells) or CD8α (53-6.7) (for PbT-II CD4$^+$ T cells) before incubation with BioMag goat anti-rat IgG coupled magnetic beads (Qiagen) and magnetic separation. Enriched T cells were 80% to 95% pure. Purified T cells were injected via the tail vein into recipient mice. For co-transfers, CD8$^+$ and CD4$^+$ T cells were transferred at a ratio of 1:1.

### γδ T cell enrichment and adoptive transfer

Naive γδ T cells were negatively enriched from spleens. Spleens were finely minced and digested for 20 min in 1 mg ml$^{-1}$ Collagenase III and 20 μg ml$^{-1}$ DNase I supplemented with a competitive antagonist of P2X$_7$R (A-438079 hydrochloride; Santa Cruz Biotechnology). Digested spleens were filtered through a 70 μM mesh followed by red blood cell lysis. Cells were labeled with rat monoclonal antibodies specific for MHC-II (M5-114), Mac-1 (M1/70), CD4 (GK1.5) and B220 (RA3-6B2) before incubation with BioMag goat anti-rat IgG coupled magnetic beads (Qiagen) and magnetic separation. Enriched γδ T cells were injected via the tail vein into recipient mice.

### DC isolation

DCs were expanded in vivo via subcutaneous injection of B16.Flt3L tumor cells[67] and then isolated from the spleens. Spleens were finely minced in 1 mg ml$^{-1}$ Collagenase III, 20 μg ml$^{-1}$ DNase I and incubated at 37 °C for 20 min. DC-T cell complexes were disrupted with the addition of 0.1 M EDTA (pH 7.2). Digested spleens were then filtered through a 100 μM mesh before light-density separation with 1.077 g cm$^{-3}$ Nycodenz medium. cDC1s/pre-cDC1s were negatively enriched for through incubation with rat mAb against CD3ε (KT3), Thy1.1 (T24/31.7), Gr-1 (RB6-8C5), B220 (RA3-6B2), Ter-119 and Mac-1 (M1/70) before incubation with BioMag goat anti-rat IgG coupled magnetic beads (Qiagen) and magnetic separation. Target cells were characterized as CD11c$^+$, MHC-II$^{int/+}$, CD11b$^-$, CD24$^{hi}$, CD8α$^+$ with an average purity of 60-80%. Purified cDC1s were injected via the tail vein into recipient mice.

### Gene deletion in T cells by CRISPR-Cas9 editing

Purified transgenic T cells were gene-edited via electroporation with single guide (sg) RNA ribonucleoproteins (RNPs) (sgRNA/Cas9 RNPs). sgRNA/Cas9 RNPs were formed in Nuclease-free H$_2$O using 0.6 μl Alt-R S. *pyogenes* Cas9 Nuclease V3 (10 mg ml$^{-1}$, Integrated DNA Technologies) incubated with 0.3 nmol of sgRNA targeting either *Il4rα* (5′-AGUGGAGUCCUAGCAUCACG-3′, 3′-AUCCAG GAACCACUCACACG-5′) or *Cd19* (5′-AAUGUCUCAGACCAUAUGGG-3′) purchased from Synthego (CRISPRevolution sgRNA EZ Kit) for 10 min at room temperature. sgRNA sequences were designed using Benchling and CRISPOR to determine suitable targets based on target efficiency and off-target predictions, and biological relevance. For each reaction, $10 \times 10^6$ target cells were resuspended in reconstituted P3 buffer, sgRNA/Cas9 RNP complex and electroporated using a Lonza 4D-Nucleofector (program code DN100). Electroporated cells were then rested in complete RPMI medium (RPMI1640, 10% FCS, 2 mM L-glutamine, 100 U ml$^{-1}$ penicillin, 100 mg ml$^{-1}$ streptomycin and 50 mM 2-mercaptoethanol) in a 96-well plate for at least 10 min at 37 °C. Cells were injected into mice via the tail vein.

## Gene deletion in primary DCs by CRISPR-Cas9 editing

Purified CD24+ cDC1s were gene-edited via electroporation with sgRNA/Cas9 RNPs. sgRNA/Cas9 RNPs were formed as described above for T cell gene editing. In each reaction, $10 \times 10^6$ target cells were resuspended in 20 μl reconstituted P3 buffer, followed by the sgRNA/Cas9 RNP complex, and then electroporated using a Lonza 4D-Nucleofector (program code CM137). Electroporated cells were then rested in complete RPMI in a 96-well plate for at least 20 min at 37 °C then injected i.v. into mice.

## In vitro cDC1 gene expression assay

DCs were isolated through the protocol outlined above. CD11c+, MHC-II$^{int/+}$, CD11b−, CD24$^{hi}$, F4/80−, B220−, CD172a− cells were sort purified using a BD FACSAria III Cell Sorter into filtered RPMI1640 supplemented with 50% heat-inactivated FCS. $5 \times 10^5$ cells were plated in a 24-well plate in complete Kenneth D Shortman RPMI (KDS-RPMI1640, 10% FCS, 2 mM L-glutamine, 100 U ml$^{-1}$ penicillin, 100 mg ml$^{-1}$ streptomycin and 50 mM 2-mercaptoethanol). Cells were incubated in the presence of αCD40 (10 μg ml$^{-1}$), recombinant mouse (rm) IL-4 (60 ng ml$^{-1}$), rmIFNγ (20 ng ml$^{-1}$), or LPS (1 μg ml$^{-1}$). Plates were then incubated at 37 °C and 5% $CO_2$ for 4 h[68]. For RNA isolation, cells were washed and resuspended in TRIzol (Life Technologies), snap frozen and stored at −80 °C.

## RNA isolation and sequencing

RNA was extracted using a Direct-zol RNA MicroPrep kit (Zymo Research) as per manufacturer's protocol. Libraries were prepared using Illumina stranded mRNA library kits and the outputs sequenced to a depth of 20 M reads per sample on a 150PE on Illumina NovaSeq X Plus 10B flow cell. Library preparation and sequencing was conducted at the Australian Genome Research Facility.

## Bioinformatic analyses

Sequencing reads were aligned to the GRCm39 reference genome and transcriptome (v105) using the STAR aligner (v2.7.8a)[69] and gene counts established using featureCounts from the subread package (v2.0.0)[70] before analysis with R (v4.5.0). Technical replicate counts were summed, gene-wise, to produce single biological replicates for subsequent analysis. Genes were filtered if they failed to achieve a count above 10 in all samples in at least one experimental group. Batch integration and normalization was performed as follows. Counts tables were combined for genes seen in both batches, counts-per-million values were calculated, using scaling factors derived from the TMM method[71] (v4.6.2), then $\log_2$ transformed with a prior count of 1. The RUV-III[72] (v0.9.7.1) methodology was applied with biological replicates nominated as replicates, mouse housekeeping genes[73] nominated as 'negative control' genes, and $k = 15$ factors of unwanted variation. Integration and normalization success was assessed with relative log expression plots[74], PCA plots[75] and $P$ value histograms. The limma package[76] (v3.64.1) was then used to fit gene-wise linear models for the experimental design with the output from RUV-III as additional model covariates, incorporating the TMM scaling factors and the prior count of 1. Empirical Bayes moderated $t$-statistics were used to test for differential expression using the limma function eBayes with arguments trend=TRUE and robust=TRUE. Genes were judged to be differentially expressed if their Benjamini and Hochberg[77] adjusted $P$ value was less than 0.05. Heatmaps were produced using pheatmap (v1.0.13) from RUV-III adjusted data using gene-wise standardization, producing a $Z$-score, with genes clustered by Pearson correlation. Venn diagrams were made with ggvenn (v0.1.10), and volcano and logFC plots were made with ggplot2 (v3.5.2) and ggrepel (v0.9.6).

## Generation of mixed and single BM chimeras

Mice were irradiated (2 x 550 rad, 3 h apart) then reconstituted with donor BM cells. For mixed BM chimeras, mice were reconstituted with a 1:1 ratio of BM cells from two donors. BM cells were prepared by removing both the tibia and femur of donor mice. Bones were flushed with complete RPMI then passed through a 70 μM mesh filter. T cells were removed by resuspending pellets with Abs against CD4 (RL172), CD8α (3.168) and Thy1 (Jlj) followed by incubation with rabbit complement to remove bound cells. Irradiated mice were injected via the tail vein with $3-5 \times 10^6$ bone marrow cells. Chimeras were injected i.p. with 0.1 ml T24 (α-Thy1.1) the following day. Mice were left to recover for 8 weeks and received antibiotic water (2.5 g liter$^{-1}$ neomycin sulfate, 0.94 g liter$^{-1}$ Polymyxin B sulfate) for the first 4 weeks, commencing 1 day before irradiation, followed by normal water for at least another 4 weeks before experimental work.

## In vivo antibody depletion/blockade

Mice were treated i.p. with purified α-IL-4 (clone 11B11, BioXCell) or α-horseradish peroxidase (clone HRPN, InVivoMab, BioXCell) isotype control (500 μg per mouse, days −1 and 0; 200 μg per mouse, days 1-3) for 5 days starting 1 day before RAS vaccination[12]. For IL-12p40 or IFNγ neutralization, mice were treated i.p. with purified α-IL-12p40 (clone C17.8, WEHI Hybridoma Facility; BioXCell) or the α-trinitrophenol isotype control (clone 2A3, BioXCell) (250 μg per mouse days −1, 0, 1 and 5)[78] or α-IFNγ (clone R4-6A2, InVivoMab, BioXCell) or the α-horseradish peroxidase (clone HRPN, InVivoMab, BioXCell) isotype control (200 μg per mouse, days −1, 0 and 1). To block γδ T cell function in vivo, mice received a single i.v. dose of α-pan-γδTCR (clone GL3[79], WEHI Hybridoma Facility) (200 μg per mouse, day −1), α-Vγ1 (clone 2.11, BioXCell) (200 μg per mouse$^{-1}$, day −1)[80] or Purified Armenian Hamster IgG (clone HTK888, BioLegend)[16] isotype control (200 μg per mouse, day −1). For B6.4C13R experiments, mice were injected i.v. with the α-ARTC2 nanobody (s + 16a; Treg Protector, BioLegend)[81] (23-h timepoint; 50 μg per mouse at 8- and 23-h post-vaccination, 44-h timepoint; 50 μg per mouse at 8, 26 and 43 h post-vaccination).

## Organ processing and flow cytometry

Single cell suspensions were prepared from spleen or livers by passing the tissues through 70 μM mesh filters. For experiments in which splenic DCs were enumerated or in B6.4C13R experiments, spleens were finely minced and incubated in 1 mg ml$^{-1}$ Collagenase III and 20 μg ml$^{-1}$ DNase I at 37 °C for 20 min then filtered through a 100 μM mesh. Liver preparation required resuspension in ~35% isotonic Percoll and centrifugation at 21 to 24 °C, 500 x $g$, for 20 min. Red blood cells were lysed and removed from spleen and liver. In B6.4C13R experiments, MHC-II (M5-114), Mac-1 (M1/70), CD4 (GK1.5) and B220 (RA3-6B2) were removed with rat mAb and BioMag goat anti-rat IgG coupled magnetic beads (Qiagen) followed by magnetic separation before surface staining. Surface staining was performed at 4 °C in PBS containing a fixable viability dye and fluorochrome-conjugated mAb. H2-K$^b$-PbRPL6$_{120-127}$, tetramers were provided by S. Gras. Sphero Blank Calibration beads were used to generate cell counts. Samples were acquired on a Cytek Aurora using the SpectroFlo v3.0 system (Cytek).

## T cell stimulation for cytokine and transcription factor staining

Cells were stimulated in complete RPMI supplemented with phorbol myristate acetate (PMA; 50 ng ml$^{-1}$, Sigma-Aldrich) and Ionomycin (1 μg ml$^{-1}$, Sigma-Aldrich) in the presence of Brefeldin A (10 μg ml$^{-1}$, Sigma-Aldrich), GolgiStop (1:1500, BD Biosciences), and A-438079 hydrochloride (25 μM, Santa Cruz Biotechnology) for 5 h at 37 °C. After stimulation, cells were stained for surface markers then pre-fixed in 1% paraformaldehyde for 15 min followed by fixation and permeabilization with the eBioscience Foxp3 Transcription Factor Staining Buffer Set (Invitrogen), as per manufacturer's instructions. Cells were then stained for intracellular proteins overnight at 4 °C in Perm Buffer supplemented with 2% rat and mouse serum.

## Multiplex bead array for cytokine detection

Spleens were homogenized in cold PBS supplemented with 1uM EDTA. Homogenates were snap frozen and maintained at −80 °C. When

required, homogenates were centrifuged at 2,000 x *g* for 10 min, and cytokine concentrations were measured from the supernatant using a LegendPlex Mouse B cell Kit 12-plex (BioLegend) following manufacturer's instructions.

### In vitro activation and culture of T cells

PbT-I cells were activated in complete RPMI for up to 6 days with PbRPL6$_{120-127}$ (NVFDFNNL) peptide-pulsed splenocytes, in the presence of LPS (1 μg ml$^{-1}$; Sigma-Aldrich) and no added cytokine, rmIL-12 (10 ng ml$^{-1}$ (ref. 82)) or rmIL-4 (60 ng ml$^{-1}$ (ref. 83); BioLegend) or both. Activated cells were fed with complete RPMI at day 3 and split at day 4. To assess phenotype, cells were washed and labeled with a viability dye and fluorochrome-conjugated monoclonal antibodies.

### Natural malaria infection in Timika cohort

PBMCs and plasma samples were collected during previously conducted trials in Timika, Papua, between 2004 and 2005[84]. All patients gave informed consent before participation. No compensation was provided. Timika is a lowland town located in the South-Central Papuan province of Indonesia[85]. Malaria transmission is perennial in lowland Papua, with a prevalence of 28.3% in children under five years, 46.3% in children aged 5 to 15 years, and 36.8% in adults older than 15 years[85]. For the parent trials, adults with slide-confirmed malaria and fever, or fever within the last 48 h, were enrolled in randomized controlled trials of artemisinin combined therapy[84]. Malaria parasite infection was categorized as *P. falciparum* or *P. vivax* monoinfection via microscopy. Exclusion criteria included pregnant or lactating women and children with a body weight of 10 kg and under. In a subset of trial participants, blood samples were collected at enrollment for PBMC isolation (<20 ml). For the current study, PBMCs were selected from patients with parasite infection categorized as *P. falciparum* monoinfection via microscopy.

### Stimulation of PBMCs

PBMCs were isolated by Ficoll Paque Plus density centrifugation from trials conducted in Timika, Papua, or from malaria-naive Australian donors (Extended Data Table 1). Isolated PBMCs were cryopreserved in 10% DMSO/FCS. Cryopreserved PBMCs were thawed at 37 °C, washed twice in RPMI containing 10% FCS and then once in RPMI alone. Cells were aliquoted to concentrations of 10$^7$ cells ml$^{-1}$ in 10% FCS/RPMI. PBMCs were rested overnight in 10% FCS/RPMI after thawing. After rest, PBMCs were stimulated with PMA (25 ng ml$^{-1}$) and Ionomycin (1 μg ml$^{-1}$) for 6 h at 37 °C. Brefeldin A (10 mg ml$^{-1}$) and monensin (10 mg ml$^{-1}$) were added after 2 h of stimulation. To maximize surface labeling CCR7 and CXCR5 due to downregulation after exposure to PMA and Ionomycin, fluorescent-tagged anti-CCR7 and anti-CXCR5 antibodies were included during stimulation in the presence of human Fc block. Stimulated PBMCs were washed with 2% FCS/PBS, and then viability and surface staining were performed. Cells were fixed and permeabilized with BD Cytofix/Cytoperm Fixation/ Permeabilization Kit, and then intracellular staining was performed with fluorescent-tagged antibodies. Cells were resuspended in 2% FCS/PBS, and events were collected on a Cytek Aurora using the SpectroFlo v3.0 system (Cytek).

### Statistical analyses

Statistical analyses of graphed data were performed using GraphPad Prism software. *P* values were deemed significant if less than 0.05. Data distribution was assumed to be normal, but this was not formally tested, and as such, all statistical tests were performed on log-transformed data unless otherwise specified. No statistical methods were used to predetermine sample sizes, but our sample sizes are similar to those reported in previous publications[40,83,86]. Data collection and analysis were not performed blind to the conditions of the experiments. No data points were excluded from the statistical analyses.

### Reporting summary

Further information on research design is available in the Nature Portfolio Reporting Summary linked to this article.

### Data availability

RNA-sequencing data have been deposited in the EMBL-EBI gene expression database under the accession code E-MTAB-16291. Source data are provided with this paper.

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

## Acknowledgements

The work was supported by the Australian National Health and Medical Research Council (grants 1105817 and 2002682; L.B.) and the Australian Research Council (grant DP220103545; D.F.-R., W.R.H. and L.B.). This research was funded in whole or part by the National Health and Medical Research Council (1105817 and 2002682; L.B.). For the purpose of open access, the author has applied a Creative Commons Attribution (CC BY) licence to any Author Accepted Manuscript version arising. S.G. is supported by an NHMRC SRF fellowship (1159272). We wish to acknowledge the WEHI Irradiation Facility for resources and the University of Melbourne Bioresources platform and the Melbourne Cytometry platform for support and expertise. We also wish to thank M. Damtsis and G. Davey for all the indirect contributions that make great research possible and Y. Alexandre for critical reading of the paper. We thank the staff of the local hospital, Indonesian Ministry of Health, and everyone involved in the Timika studies.

## Author contributions

S. Le performed the experiments with help from D.M., S. Liu, Z.G., R.M. and L.B. and wrote the first draft of this paper. N.D. and M.N. performed and analyzed the human cohort samples, with the support of D.O., supervised by M.B. Human samples were collected by E.K. A.C. provided RAS, and C.J. made ΔMei2 parasites, both under supervision by G.I.M. C.X. and T.N.B. contributed to the methodology, supervised by H.-F.K. or L.K.M. S.G. provided resources. J.S. and L.C.G. conducted the bioinformatic analysis of the RNA-sequencing data. A.B., S.B., I.A.C. and D.F.-R. helped with the conceptualization and methodology. W.R.H. and L.B. conceptualized, funded and supervised the study and wrote the paper.

## Competing interests

The authors declare no competing interests.

## Additional information

**Extended data** is available for this paper at https://doi.org/10.1038/s41590-025-02397-z.

**Correspondence and requests for materials** should be addressed to Lynette Beattie.

**Extended Data Table 1 | Human cohort clinical characteristics**

| Group | n | Age (years) Median [IQR] | Male, number (%) | Parasites (x10$^5$/mL) Median [IQR] |
|---|---|---|---|---|
| Naive | 5 | 39 [32 - 46] | 2 (40%) | - |
| Endemic Healthy Control | 4 | 25 [20 - 30] | 3 (75%) | 0 |
| *Pf* | 7 | 26 [23 - 27] | 3 (43%) | 186,648 [7,253 – 275.528] |

**Extended Data Table 2 | Primers used for the generation of ΔMei2KO parasites**

| Name | Gene | F/R | Sequence | Purpose |
|---|---|---|---|---|
| CJ472 | mei2 | F | TATAGGTACCGTATGGGAACATGCATATTATGTGG | Forward to amplify the 5' flank of mei2 ko primer KpnI |
| CJ473 | mei2 | R | CCAACCTAGGTATAGGCGCGCCTACAAAAGGAATATGGGGAATACACC | Reverse for 5' Flank with linker including PAM site |
| CJ474 | mei2 | F | AGGCGCGCCTATACCTAGGTTGGGGATGTTTATAAATAAAATAGTGTAAAATTCG | 3' Flank fwd with PAM linker |
| CJ475 | mei2 | R | atatGAATTCCCTGTTTAGGTTTATTTTGTCATTTCAC | 3' flank rev with EcoRI |
| CJ562 | mei2 | F | GGCATGATGCCGAATGCC | Outside 5' flank for genotyping |
| CJ563 | mei2 | R | CATGTGTACACACGATCATATGG | Inside excised region downstream of 5' flank. Absent in KO |
| CJ564 | mei2 | F | GATGACCCGAGTTGTGAAGG | Inside excised region upstream of 3' flank. Absent in KO |
| CJ565 | mei2 | R | GAATTTTGAATATATGATCATATCGCACG | Outside 3' flank for genotyping |
| CJ566 | mei2 | R | CTAGGTATAGGCGCGCC | Inside linker to detect KO |
| CJ567 | mei2 | F | GCGCCTATACCTAGGTTGG | Inside linker to detect KO |
| CJ595 | mei2 | F | GCCTATACCTAGGTTGGGG | Alternative fwd in linker region of mei2KO for KO genotyping |
| Mei2KO guide 28 | mei2 | F | tattCATGTGTACACACGATCATA | Guide for mei2 |
| Mei2KO guide 28 | mei2 | R | aaacTATGATCGTGTGTACACATG | Guide for mei2 |
| Mei2KO Guide 3 | mei2 | F | tattAATGATGACCCGAGTTGTGA | Guide for mei2 |
| Mei2KO Guide 3 | mei2 | R | aaacTCACAACTCGGGTCATCATT | Guide for mei2 |

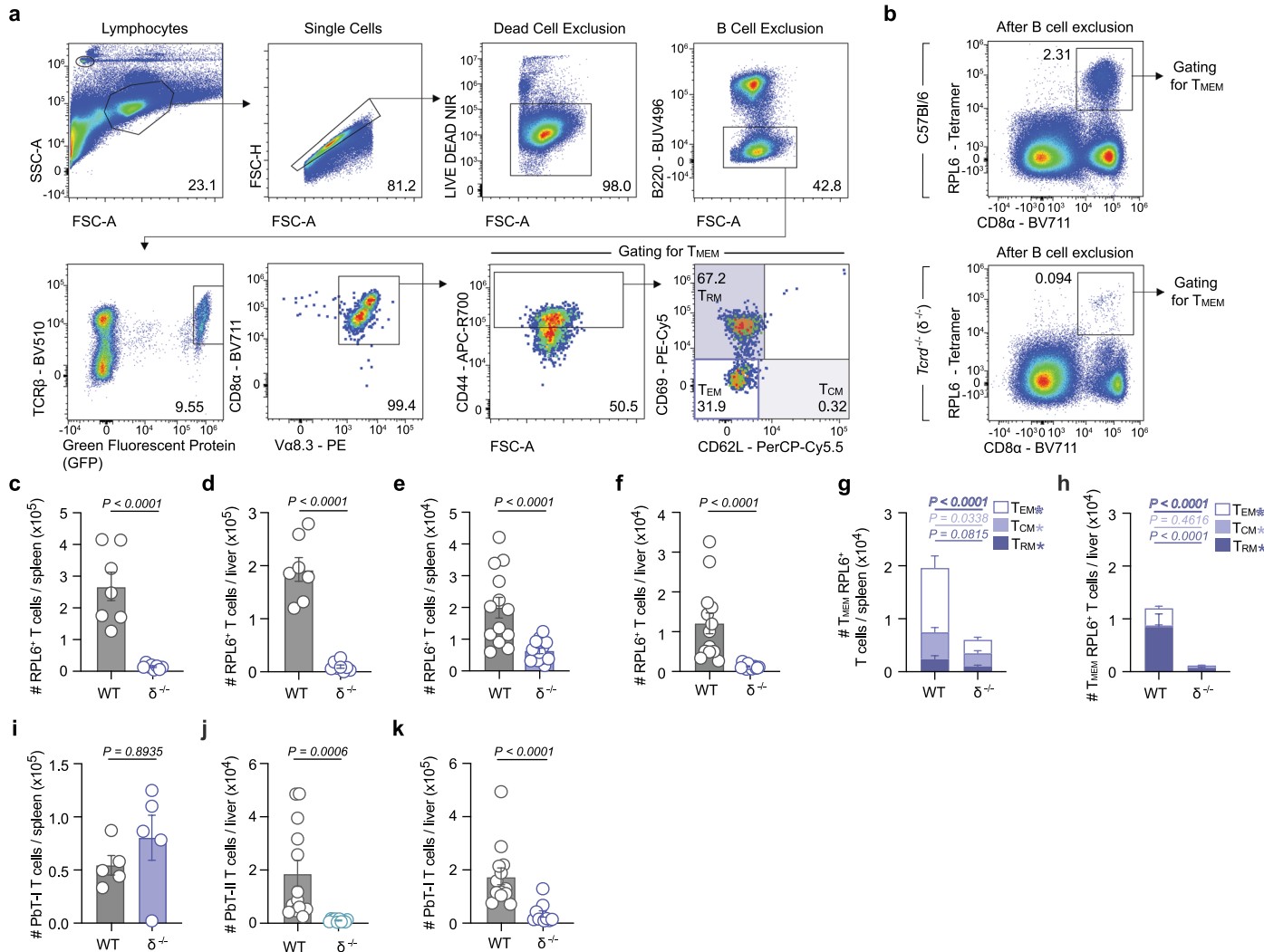

**Extended Data Fig. 1 | γδ T cells are essential for the CD8+ T cell response to RAS.** Mice were vaccinated with $5 \times 10^4$ RAS then assessed (**a, e–h**) 35 days or, (**b–d**) 6 days later. (**a**) Gating of transgenic GFP+TCRβ+Vα8.3+CD8α+ PbT-I cells in the liver of B6 mice and subsequent $T_{MEM}$ stratification; central memory $T_{CM}$ (CD62L+, CD69−), effector memory $T_{EM}$ (CD62L−, CD69−) and resident memory $T_{RM}$ (CD62L−, CD69+). (**b**) Gating of RPL6+ tetramer staining in the liver of WT (B6) or δ−/− mice 6 days after RAS. For memory timepoint experiments, gating for $T_{MEM}$ was applied. (**c and d**) Numbers of RPL6 tetramer positive CD8+ T cells in the (**c**) spleen and (**d**) liver of WT (n = 7) or δ−/− (n = 8) mice, 6 days after immunization with RAS. (**e and f**) Numbers of RPL6 tetramer positive CD8+ T cells in the (**e**) spleen or (**f**) liver of WT (n = 13) or δ−/− (n = 14) mice, 35 days after RAS vaccination. (**g and h**) Frequency of memory T cell ($T_{MEM}$) subsets; $T_{CM}$ $T_{EM}$ and $T_{RM}$ within the

RPL6 tetramer positive CD44hi CD8+ T cell compartment in the (**g**) spleen or (**h**) liver of WT (n = 13) or δ−/− (n = 14) mice. (**i**) Numbers of PbT-I cells in the spleen of WT (n = 5) or δ−/− (n = 5) mice, 6 days after injection of irradiated *P. berghei* infected red blood cells. Relating to Fig. 1 (**l, m**), mice received $10^6$ PbT-I and II cells 1 day before $5 \times 10^4$ RAS and were analyzed at day 6 post-immunization. Numbers of (**j**) PbT-II or (**k**) PbT-I cells recovered in the liver of WT (n = 6), or δ−/− (n = 7) mice. Data show (**a, d–g, i, j, e–h, j, k**) 3 or (**c, d, i, h**) 2 independent experiments where points show individual mice and stacked bar plots show mean of each memory cell subset. Individual data are supplied in Extended Data Source Data File 1. Flow cytometry plots are representative. Error bars indicate mean + S.E.M. Data were log-transformed and compared using an unpaired two-tailed t test.

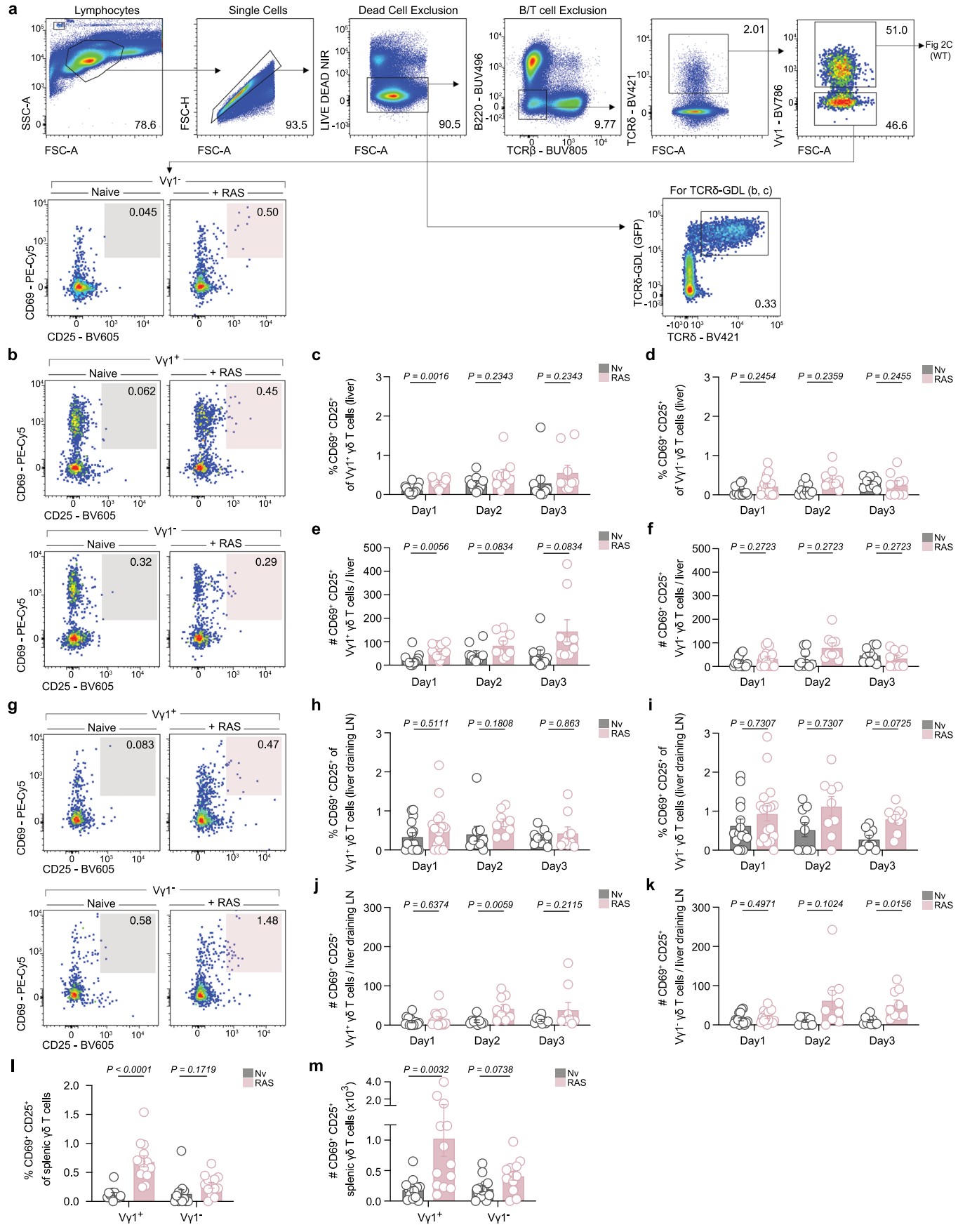

**Extended Data Fig. 2 | See next page for caption.**

**Extended Data Fig. 2 | Splenic Vγ1⁺ γδ T cells initiate immunity to RAS.**
(**a**) Gating used to generate cell numbers and frequencies in Extended Data Fig. 2 and Fig. 2d–m. For TCRδ-GDL experiments, γδ T cells were gated for using both GFP and fluorochrome-conjugated mAb. (**b**) Activation at day 1 in naive (Nv) or RAS-vaccinated Vγ1⁺ or Vγ1⁻ γδ cells in the liver (**c** and **d**) Frequency of activated (CD69⁺ CD25⁺) (**c**) Vγ1⁺ or (**d**) Vγ1⁻ γδ T cells in naïve or RAS-vaccinated WT (B6) spleens at day 1 (Nv, n = 15; RAS, n = 17), 2 (Nv, n = 9; RAS, n = 9), or 3 (Nv, n = 9; RAS, n = 9) post-injection (**e** and **f**) Number of activated (CD69⁺ CD25⁺) (**e**) Vγ1⁺ or (**f**) Vγ1⁻ γδ T cells in naïve or RAS-vaccinated WT spleens at day 1 (Nv, n = 15; RAS, n = 17), 2 (Nv, n = 9; RAS, n = 9), or 3 (Nv, n = 9; RAS, n = 9) post-injection.
(**g**) Activation at day 1 in naive or RAS-vaccinated Vγ1⁺ or Vγ1⁻ γδ T cells in the liver-draining lymph nodes (LdLN). (**h** and **i**) Frequency of activated (CD69⁺ CD25⁺) (**h**) Vγ1⁺ or (**i**) Vγ1⁻ γδ T cells in naïve or RAS-vaccinated liver-draining lymph nodes (LdLN). (**j** and **k**) Number of activated (CD69⁺ CD25⁺) (**j**) Vγ1⁺ or (**k**) Vγ1⁻ γδ T cells in naïve or RAS-vaccinated liver-draining lymph nodes (LdLN). TCRδ-GDL mice were vaccinated with $10^5$ RAS then analyzed 24 h later. (**l**) Frequency of activated (CD69⁺, CD25⁺) Vγ1⁺ or Vγ1⁻ γδ T cells in naive (Nv) (n = 10) or RAS-vaccinated WT mice (n = 13). (**m**) Number of activated (CD69⁺, CD25⁺) Vγ1⁺ or Vγ1⁻ γδ T cells. Data shows 4 independent experiments where points indicate individual mice. Error bars indicate mean + S.E.M. Data were log-transformed and compared using multiple unpaired two-tailed Welch's t tests and corrected with Holm-Sidák multiple comparisons testing.

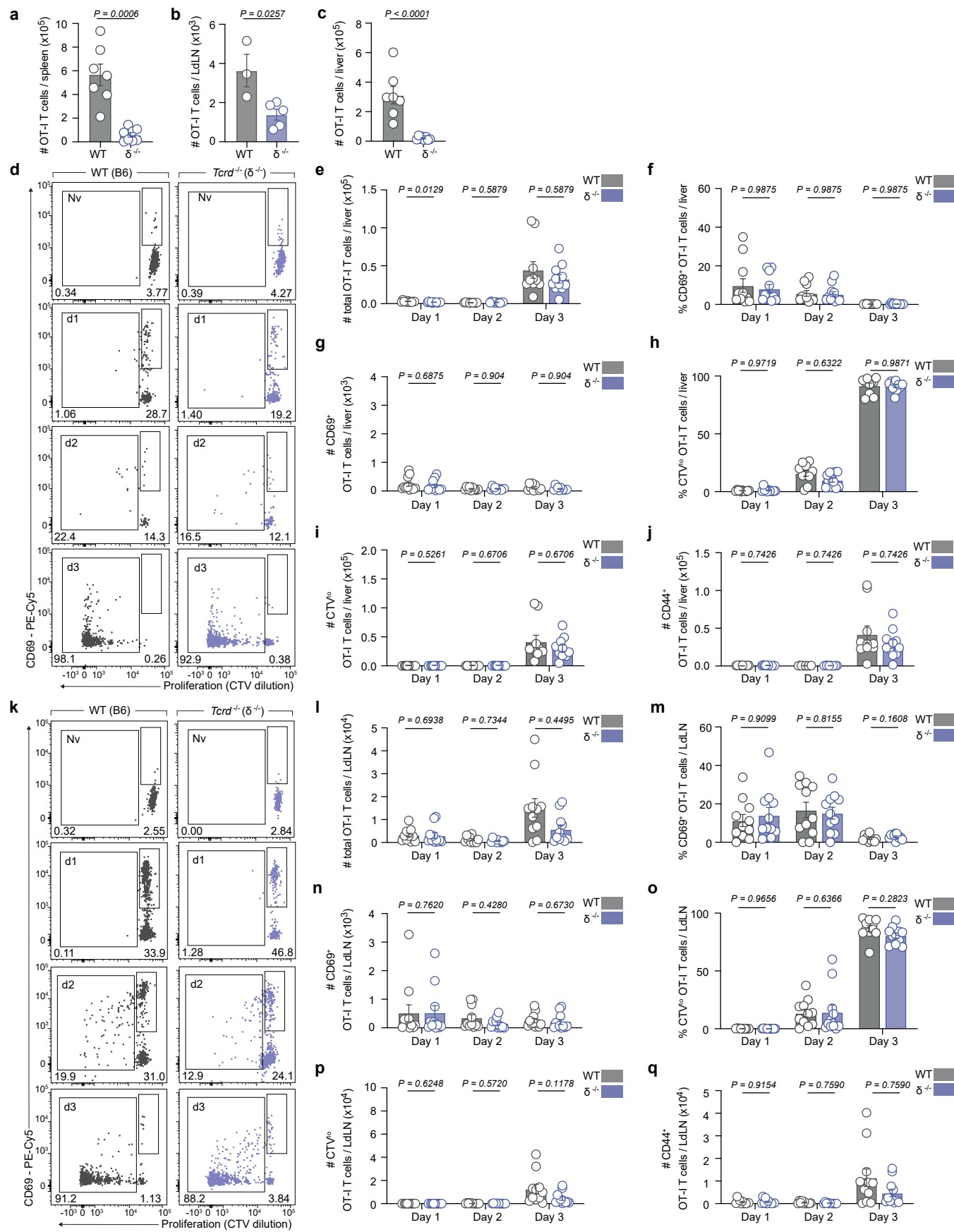

**Extended Data Fig. 3 | See next page for caption.**

**Extended Data Fig. 3 | Antigen presentation and CD40L signals are intact in the absence of γδ T cells.** $10^6$ CTV-labeled OVA-specific CD8$^+$ T (OT-I) cells were transferred into WT (B6) or $\delta^{-/-}$ recipient mice one day before vaccination with $5 \times 10^4$ RAS. Mice were analyzed at (**a–c**) day 6 or (**d–q**) day 1 (WT or $\delta^{-/-}$ n = 11), day 2 (WT or $\delta^{-/-}$ n = 11), and day 3 (WT or $\delta^{-/-}$ n = 11) after transfer. (**a–c**) Number of OT-I cells in the (**a**) spleen (WT, n = 7; $\delta^{-/-}$ n = 8), or (**b**) pooled liver-draining lymph nodes (WT, n = 3; $\delta^{-/-}$ n = 6), or (**c**) liver (WT, n = 7; $\delta^{-/-}$ n = 8), 6 days after immunization. (**d and k**) Flow cytometry plots of total OT-I cells in the (**d**) liver or (**k**) liver-draining lymph nodes (LdLN). (**e and l**) Number of total OT-I cells in the (**e**) liver or (**l**) LdLN (**f and m**) Frequency of CD69$^+$ OT-I cells in the (**f**) liver or (**m**) LdLN. (**g and n**) Number of CD69$^+$ OT-I cells in the (**g**) liver or (**n**) LdLN. (**h and o**) Frequency of CTV$^{lo}$ OT-I cells in the (**h**) liver or (**o**) LdLN. (**i and p**) Number of CTV$^{lo}$ OT-I cells in the (**i**) liver or (**p**) LdLN. (**j and q**) Number of CD44$^+$ OT-I cells in the (**j**) liver or (**q**) LdLN. Data show (**a–c**) 2 or (**d–q**) 3 independent experiments where points represent individual mice or (**b**) pooled tissue from 2 mice per independent experiment and bars represent mean. Error bars indicate mean + S.E.M. Data were log-transformed data and compared using (**a–g**) multiple unpaired two-tailed Welch's t tests and corrected using the Holm-Sidák multiple comparisons test.

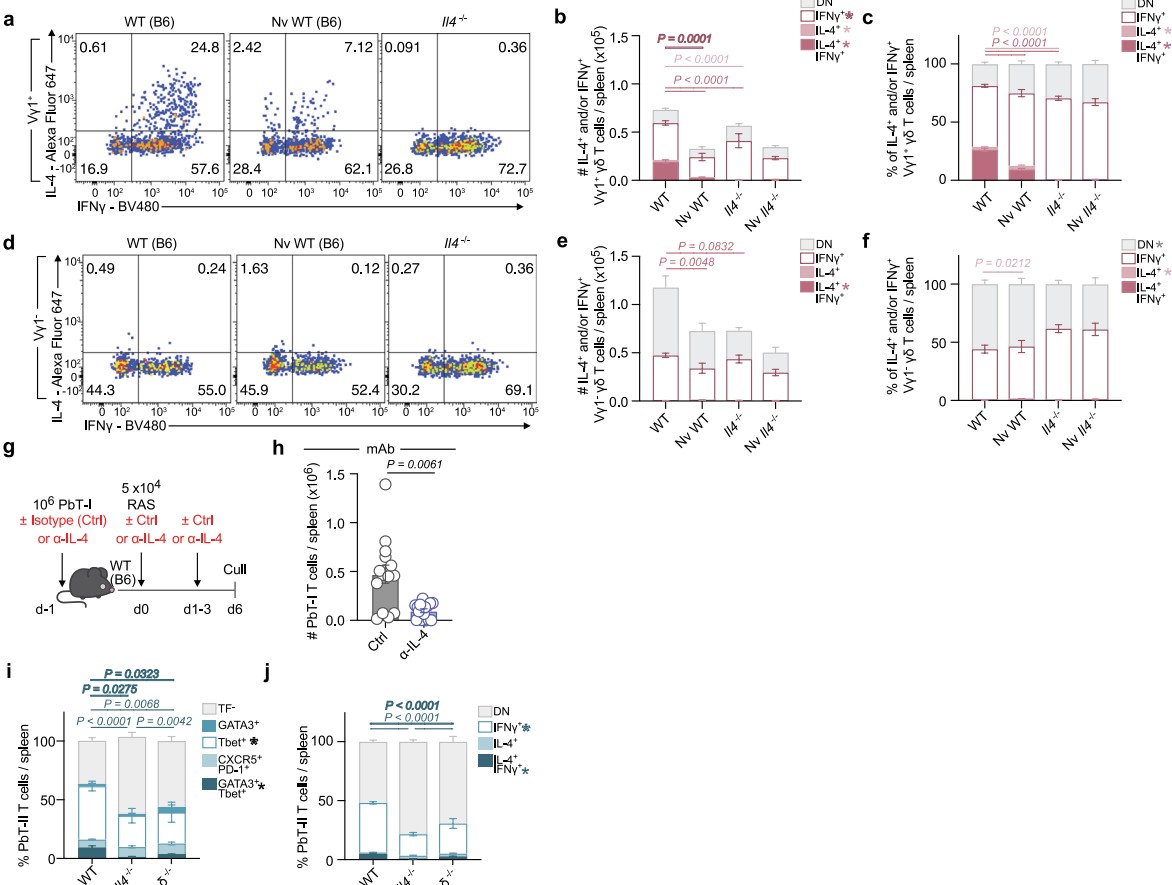

**Extended Data Fig. 4 | IL-4 is required for the CD8⁺ T cell response to RAS vaccination.** WT (B6), *Il4⁻/⁻*, and δ⁻/⁻ mice received 10⁶ PbT-I and 10⁶ PbT-II cells. Mice were vaccinated with 5 × 10⁴ RAS or left unvaccinated (Nv) the following day. Tissues were assessed at day 6 post-vaccination. (**a** and **d**) Intracellular IL-4 and/or IFNγ production by (**a**) Vγ1⁺ or (**d**) Vγ1⁻ cells in the spleen of only WT, naive WT and *Il4⁻/⁻* mice, 5 h after PMA/ionomycin restimulation. (**b**) Number and (**c**) frequency of IFNγ and IL-4 single and co-producers within the splenic Vγ1⁺ cells of vaccinated WT (n = 16) and *Il4⁻/⁻* (n = 7), or naive WT (n = 9) and *Il4⁻/⁻* (n = 5). (**e**) Number and (**f**) frequency of IFNγ and IL-4 single and co-producers within the splenic Vγ1⁻ cells of vaccinated WT (n = 16) and *Il4⁻/⁻* (n = 7), or naive WT (n = 9) and *Il4⁻/⁻* (n = 5). (**g**) Experimental design (**h**). Wild-type mice received PbT-I cells then were vaccinated one day later with RAS. To determine the effect of blocking IL-4, mice were immunized with α-IL-4 (or isotype control) (i.p.) (n = 15) across 5 days, beginning one day before RAS immunization. Mice were culled 6 days after RAS vaccination. (**h**) Numbers of PbT-I cells in the spleen of isotype control and α-IL-4 treated mice 6 days after RAS vaccination. (**i**) Frequency of spleen CD44⁺ PbT-II cells expressing no transcription factors tested (TF-), GATA3 or Tbet alone, or co-expressing CXCR5 and PD-1, or GATA3 and Tbet from WT (n = 16), *Il4⁻/⁻* (n = 9) or δ⁻/⁻ (n = 7) mice. (**j**) Frequency of intracellular IL-4 and/or IFNγ production by the CD44⁺ PbT-II cell population at day 6 post-vaccination. Data shows (**a–e**, **i**, **j**) 3 or (**g**, **h**) 4 independent experiments where points show individual mice. Individual data are supplied in Extended Data Source Data File 4. Flow cytometry plots are representative. Error bars indicate mean + S.E.M. Data were log-transformed and compared using (**a–e**, **i**, **j**) an ordinary one-way ANOVA or (**g**, **h**) an unpaired two-tailed t test.

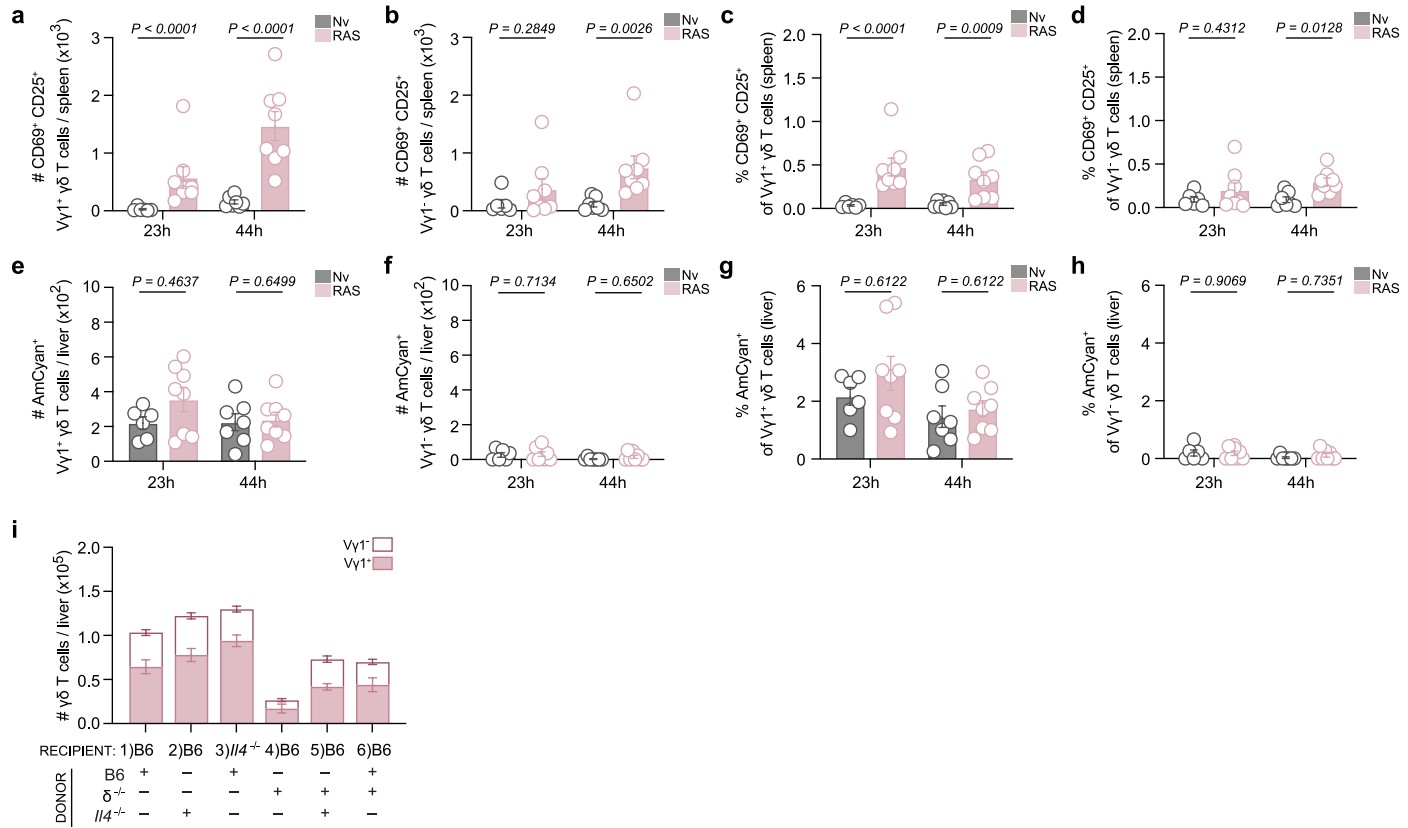

**Extended Data Fig. 5 | IL-4 from γδ T cells.** Relating to Fig. 5b–i. B6.4C13R mice were vaccinated with 5 × 10⁴ RAS and treated with α-ARTC2 nanobody (i.v.) at 8 and 22 or 8, 26 and 44 h after vaccination. (**a** and **b**) Total number of activated (CD69⁺CD25⁺) (**a**) Vγ1⁺ or (**b**) Vγ1⁻ γδ T cells in enriched spleens of naive (23 h, n = 6; 44 h, n = 7) and vaccinated (23 h or 44 h, n = 8) mice. (**c** and **d**) Frequency of activated (CD69⁺CD25⁺) (**c**) Vγ1⁺ and (**d**) Vγ1⁻ γδ T cells in enriched spleens. (**e** and **f**) Total number of IL-4⁺ (AmCyan⁺) (**e**) Vγ1⁺ or (**f**) Vγ1⁻ γδ T cells in the liver.

(**g** and **h**) Frequency of IL-4⁺ (AmCyan⁺) (**g**) Vγ1⁺ or (**h**) Vγ1⁻ γδ T cells in the liver. Relating to Fig. 5j–m. (**i**) Total number of γδ T cells in the liver of chimeras. Data shows (**a–h**) 2 or (**i**) 4 independent experiments where points show individual mice. Error bars indicate mean + S.E.M. Data were log-transformed and compared using (**i**) an ordinary one-way ANOVA, or (**c–i**) multiple unpaired two-tailed Welch's t tests with Holm-Sidák multiple comparisons correction.

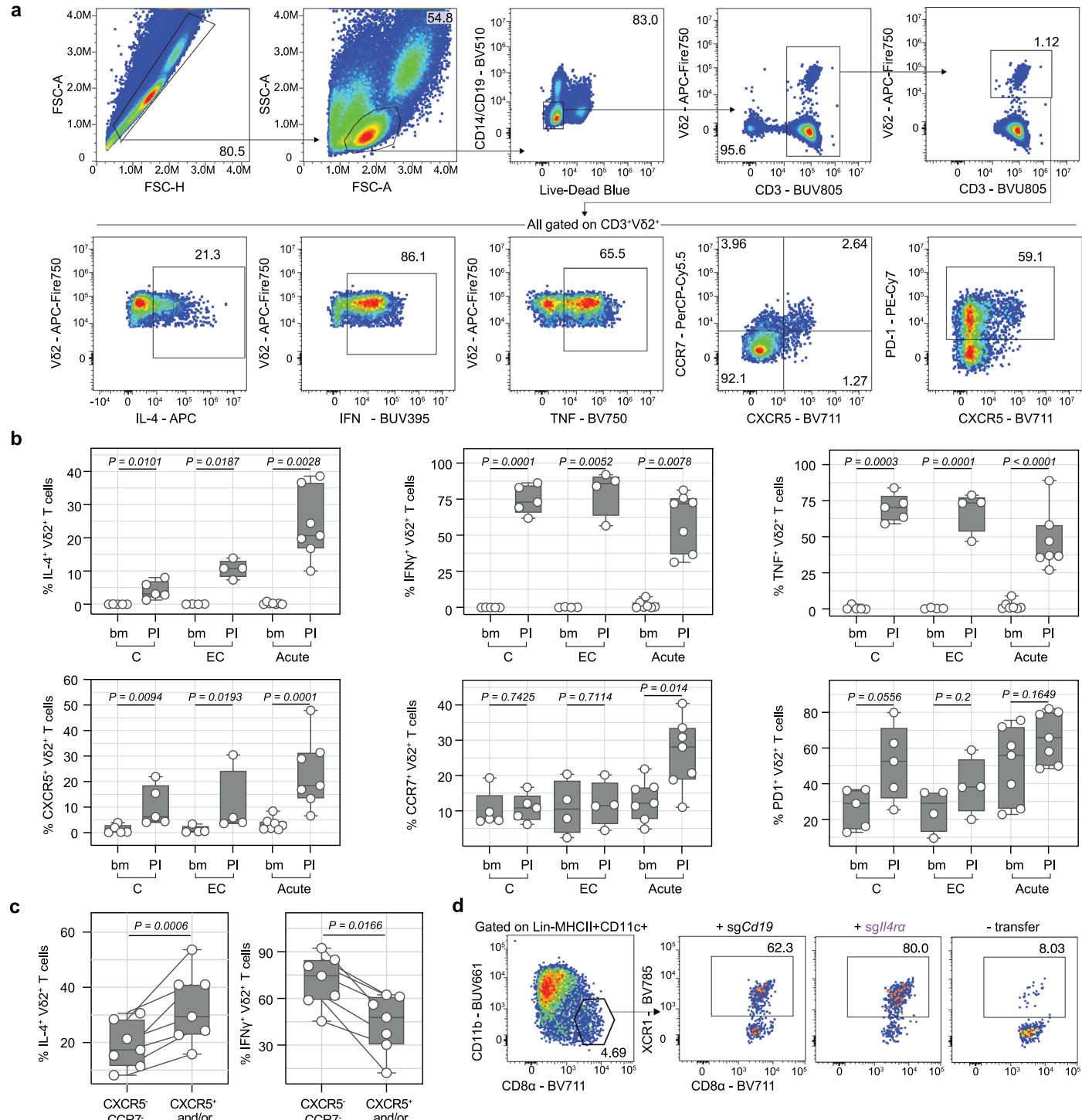

**Extended Data Fig. 6 | CD8⁺ T cells and dendritic cells respond directly to IL-4.**
Peripheral blood mononuclear cells (PBMCs) were collected from malaria-naïve healthy controls (C, *n* = 5), malaria-endemic healthy controls (EC, *n* = 4), and malaria-endemic patients with acute malaria infection (Acute, *n* = 7). (**a**) Vδ2⁺ T cells events were analyzed by flow cytometry as CD14⁻CD19⁻, Live CD3⁺ Vδ2⁺ single lymphocytes. (**b**) Cytokine production by PBMCs and surface marker upregulation was captured during stimulation with PMA/Ionomycin (PI) or blank media (bm). (**c**) CXCR5⁻CCR7⁻ and CXCR5⁺ and/or CCR7⁺ Vδ2⁺ γδ T cells production of IL-4 and IFNγ and/or TNF during acute malaria. (**d**) Flow cytometry plots showing cDC1 gating on live, MHC-II⁺ CD11c⁺ in the spleen of *Batf3*⁻/⁻ mice that received gene-edited cDC1s (+sg*Cd19*, +sg*Il4ra*), or no cDC1s (- transfer) Data points represent individual replicates. Box and whisker plot center line represents the median, box limits indicate the upper and lower quartiles, whiskers extending to 1.5x the interquartile range. Flow cytometry plots are representative. Error bars indicate mean + SD. Data were log-transformed and boxplot comparisons were made by (**b**) multiple unpaired two-tailed Student's t tests or Mann-Whitney *U* test (TNF) based on normality defined by Shapiro-Wilk normality testing or (**c**) a paired two-tailed t test.

# Reporting Summary

## Statistics

For all statistical analyses, confirm that the following items are present in the figure legend, table legend, main text, or Methods section.

| n/a | Confirmed | |
|---|---|---|
| ☐ | ☒ | The exact sample size (*n*) for each experimental group/condition, given as a discrete number and unit of measurement |
| ☐ | ☒ | A statement on whether measurements were taken from distinct samples or whether the same sample was measured repeatedly |
| ☐ | ☒ | The statistical test(s) used AND whether they are one- or two-sided<br>*Only common tests should be described solely by name; describe more complex techniques in the Methods section.* |
| ☒ | ☐ | A description of all covariates tested |
| ☐ | ☒ | A description of any assumptions or corrections, such as tests of normality and adjustment for multiple comparisons |
| ☐ | ☒ | A full description of the statistical parameters including central tendency (e.g. means) or other basic estimates (e.g. regression coefficient) AND variation (e.g. standard deviation) or associated estimates of uncertainty (e.g. confidence intervals) |
| ☐ | ☒ | For null hypothesis testing, the test statistic (e.g. *F*, *t*, *r*) with confidence intervals, effect sizes, degrees of freedom and *P* value noted<br>*Give P values as exact values whenever suitable.* |
| ☒ | ☐ | For Bayesian analysis, information on the choice of priors and Markov chain Monte Carlo settings |
| ☒ | ☐ | For hierarchical and complex designs, identification of the appropriate level for tests and full reporting of outcomes |
| ☐ | ☒ | Estimates of effect sizes (e.g. Cohen's *d*, Pearson's *r*), indicating how they were calculated |

*Our web collection on statistics for biologists contains articles on many of the points above.*

## Software and code

Policy information about availability of computer code

| | |
|---|---|
| Data collection | For flow cytometry experiments, cells were acquired on a Cytek Aurora using the SpectroFlo v3.0 system (Cytek) as stated in the relevant Methods sections. Cells were sorted using a BD FACSAriaTM III Cell Sorter and the BD FACSDiva program v9. |
| Data analysis | For RNA sequencing analysis: STAR (version 2.7.8a), featureCounts (version 2.0.0), R (version 4.5.0), EDASeq (version 2.42.0), ruv (version 0.9.7.1), edgeR (version 4.6.2), limma (version 3.64.1), ggplot2 (version 3.5.2), ggrepel (version 0.9.6), ggvenn (version 0.1.10), pheatmap (version 1.0.13).<br>Analyses of flow cytometry data were performed on FlowJo v10.10.0. Statistical analyses of graphed data were performed using GraphPad Prism software V10.<br>Cells were sorted using a BD FACSAriaTM III Cell Sorter and the BD FACSDiva program v9.<br>These packages are stated in the relevant Methods sections. |

For manuscripts utilizing custom algorithms or software that are central to the research but not yet described in published literature, software must be made available to editors and reviewers. We strongly encourage code deposition in a community repository (e.g. GitHub). See the Nature Portfolio guidelines for submitting code & software for further information.

# Data

Policy information about availability of data

All manuscripts must include a data availability statement. This statement should provide the following information, where applicable:

- Accession codes, unique identifiers, or web links for publicly available datasets
- A description of any restrictions on data availability
- For clinical datasets or third party data, please ensure that the statement adheres to our policy

Data Availability
Mouse RNA sequencing data are deposited with EMBL-EBI gene expression database accession number E-MTAB-16291 (request currently pending).
All other raw data is provided as Source Data Files and/or are available from the corresponding author upon reasonable request.

# Research involving human participants, their data, or biological material

Policy information about studies with human participants or human data. See also policy information about sex, gender (identity/presentation), and sexual orientation and race, ethnicity and racism.

| | |
|---|---|
| Reporting on sex and gender | Reported in 'Supplementary Table 1: Human cohort clinical characteristics' |
| Reporting on race, ethnicity, or other socially relevant groupings | Numbers of males and females used in the study are described in Supplementary Table 1. The data presented are not disaggregated by gender.<br>The samples used in this study were recruited as part of a different study as published in https://doi.org/10.1016/S0140-6736(07)60160-3 |
| Population characteristics | Reported in: Supplementary Table 1: Human cohort clinical characteristics |
| Recruitment | Patients were recruited as part of the original study described in:<br>https://doi.org/10.1016/S0140-6736(07)60160-3<br>Patients presenting to study hospitals with microscopy-diagnosed malaria of any species were enrolled as a part of their treatment following written informed consent. Healthy and Endemic controls enrolled following written informed consent. |
| Ethics oversight | Studies in natural infection settings of Timika in Papua were approved by the ethics committees of the Northern Territory Department of Health and Menzies School of Health Research (HREC 05-16, HREC 03-64, HREC 10-13970), the QIMR Berghofer Medical Research Institute Human Ethics Committee (HREC P3444) and the Indonesian National Institute of Health Research and Development (NIHRD KS.02.01.2.1.4042). |

Note that full information on the approval of the study protocol must also be provided in the manuscript.

# Field-specific reporting

Please select the one below that is the best fit for your research. If you are not sure, read the appropriate sections before making your selection.

☒ Life sciences          ☐ Behavioural & social sciences          ☐ Ecological, evolutionary & environmental sciences

For a reference copy of the document with all sections, see nature.com/documents/nr-reporting-summary-flat.pdf

# Life sciences study design

All studies must disclose on these points even when the disclosure is negative.

| | |
|---|---|
| Sample size | Required sample sizes were estimated based on the average magnitude of variation in previous or pilot experiments, constrained by the availability of biological samples, particularly radiation attenuated sporozoites, which are a limiting factor for numbers of animals used in most experiments contained in this study. Generally, 3-8 mice were used per group and then experiments independently repeated at least 2-3 times and results pooled. |
| Data exclusions | Data were analysed without exclusion unless explicitly stated. |
| Replication | All data are representative of multiple independent experiments repeated at least 2-3 times. |
| Randomization | Age/Sex matched mice were randomly allocated into groups. |
| Blinding | Investigators were not blinded. |

# Behavioural & social sciences study design

All studies must disclose on these points even when the disclosure is negative.

| | |
|---|---|
| Study description | *Briefly describe the study type including whether data are quantitative, qualitative, or mixed-methods (e.g. qualitative cross-sectional, quantitative experimental, mixed-methods case study).* |
| Research sample | *State the research sample (e.g. Harvard university undergraduates, villagers in rural India) and provide relevant demographic information (e.g. age, sex) and indicate whether the sample is representative. Provide a rationale for the study sample chosen. For studies involving existing datasets, please describe the dataset and source.* |
| Sampling strategy | *Describe the sampling procedure (e.g. random, snowball, stratified, convenience). Describe the statistical methods that were used to predetermine sample size OR if no sample-size calculation was performed, describe how sample sizes were chosen and provide a rationale for why these sample sizes are sufficient. For qualitative data, please indicate whether data saturation was considered, and what criteria were used to decide that no further sampling was needed.* |
| Data collection | *Provide details about the data collection procedure, including the instruments or devices used to record the data (e.g. pen and paper, computer, eye tracker, video or audio equipment) whether anyone was present besides the participant(s) and the researcher, and whether the researcher was blind to experimental condition and/or the study hypothesis during data collection.* |
| Timing | *Indicate the start and stop dates of data collection. If there is a gap between collection periods, state the dates for each sample cohort.* |
| Data exclusions | *If no data were excluded from the analyses, state so OR if data were excluded, provide the exact number of exclusions and the rationale behind them, indicating whether exclusion criteria were pre-established.* |
| Non-participation | *State how many participants dropped out/declined participation and the reason(s) given OR provide response rate OR state that no participants dropped out/declined participation.* |
| Randomization | *If participants were not allocated into experimental groups, state so OR describe how participants were allocated to groups, and if allocation was not random, describe how covariates were controlled.* |

# Ecological, evolutionary & environmental sciences study design

All studies must disclose on these points even when the disclosure is negative.

| | |
|---|---|
| Study description | *Briefly describe the study. For quantitative data include treatment factors and interactions, design structure (e.g. factorial, nested, hierarchical), nature and number of experimental units and replicates.* |
| Research sample | *Describe the research sample (e.g. a group of tagged Passer domesticus, all Stenocereus thurberi within Organ Pipe Cactus National Monument), and provide a rationale for the sample choice. When relevant, describe the organism taxa, source, sex, age range and any manipulations. State what population the sample is meant to represent when applicable. For studies involving existing datasets, describe the data and its source.* |
| Sampling strategy | *Note the sampling procedure. Describe the statistical methods that were used to predetermine sample size OR if no sample-size calculation was performed, describe how sample sizes were chosen and provide a rationale for why these sample sizes are sufficient.* |
| Data collection | *Describe the data collection procedure, including who recorded the data and how.* |
| Timing and spatial scale | *Indicate the start and stop dates of data collection, noting the frequency and periodicity of sampling and providing a rationale for these choices. If there is a gap between collection periods, state the dates for each sample cohort. Specify the spatial scale from which the data are taken* |
| Data exclusions | *If no data were excluded from the analyses, state so OR if data were excluded, describe the exclusions and the rationale behind them, indicating whether exclusion criteria were pre-established.* |
| Reproducibility | *Describe the measures taken to verify the reproducibility of experimental findings. For each experiment, note whether any attempts to repeat the experiment failed OR state that all attempts to repeat the experiment were successful.* |
| Randomization | *Describe how samples/organisms/participants were allocated into groups. If allocation was not random, describe how covariates were controlled. If this is not relevant to your study, explain why.* |
| Blinding | *Describe the extent of blinding used during data acquisition and analysis. If blinding was not possible, describe why OR explain why blinding was not relevant to your study.* |

Did the study involve field work?    ☐ Yes    ☐ No

## Field work, collection and transport

| | |
|---|---|
| Field conditions | *Describe the study conditions for field work, providing relevant parameters (e.g. temperature, rainfall).* |
| Location | *State the location of the sampling or experiment, providing relevant parameters (e.g. latitude and longitude, elevation, water depth).* |
| Access & import/export | *Describe the efforts you have made to access habitats and to collect and import/export your samples in a responsible manner and in compliance with local, national and international laws, noting any permits that were obtained (give the name of the issuing authority, the date of issue, and any identifying information).* |
| Disturbance | *Describe any disturbance caused by the study and how it was minimized.* |

# Reporting for specific materials, systems and methods

We require information from authors about some types of materials, experimental systems and methods used in many studies. Here, indicate whether each material, system or method listed is relevant to your study. If you are not sure if a list item applies to your research, read the appropriate section before selecting a response.

### Materials & experimental systems

| n/a | Involved in the study |
|---|---|
| ☐ | ☒ Antibodies |
| ☒ | ☐ Eukaryotic cell lines |
| ☒ | ☐ Palaeontology and archaeology |
| ☐ | ☒ Animals and other organisms |
| ☐ | ☒ Clinical data |
| ☒ | ☐ Dual use research of concern |
| ☒ | ☐ Plants |

### Methods

| n/a | Involved in the study |
|---|---|
| ☒ | ☐ ChIP-seq |
| ☐ | ☒ Flow cytometry |
| ☒ | ☐ MRI-based neuroimaging |

## Antibodies

| | |
|---|---|
| Antibodies used | Name, Fluorochrome, Supplier/Manufacturer, Catalog #, dilution<br>For human studies:<br>LiveDead Blue - Invitrogen L34962 (1:5000)<br>Vδ2 APC/Fire 750 B6 Biolegend 331420 (1:100)<br>CD3 BUV805 SK7 BD Biosciences 612893 (1:200)<br>CCR7 PerCP-Cy5.5 150503 BD Biosciences 561144 (2:25)<br>CXCR5 (CD185) BV711 J252D4 Biolegend 356934 (1:50)<br>PD-1 (CD279) PE-Cy7 EH12.1 BD Biosciences 561272 (1:50)<br>IFNγ BUV395 B27 BD Biosciences 563563 (1:50)<br>TNF BV750 Mab11 BD Biosciences 566359 (1:100)<br>IL4 APC MP4-25D2 Biolegend 500812 (1:10)<br>CD14 BV510 M5E2 Biolegend 301842 (1:50)<br>CD19 BV510 SJ25D1 Biolegend 363020 (1:50)<br>For Mouse studies:<br>LIVE/DEAD Fixable Near IR Dead Cell Stain Kit Invitrogen L10119 (1:1000)<br>B220 BUV496 RA3-6B2 BD Biosciences 356938 (1:500)<br>CD11a PE-Cy7 2D7 BD Biosciences 558191 (1:500)<br>CD11b BV711 M1/70 Biolegend 558191 (1:300)<br>CD11b BUV661 M1/70 BD Biosciences 612977 (1:300)<br>CD11c BV605 N418 Biolegend 117333 (1:200)<br>CD19 PerCP-Cy5.5 1D3 Biolegend 152406 (1:400)<br>CD124 (IL-4Ra) PE mIL4R-M1 BD Biosciences 552509 (1:200)<br>CD124 (IL-4Ra) PE-Cy7 11B11 Biolegend 504117 (1:200)<br>CD172a (SIRPa) FITC P84 Biolegend 144006 (1:200)<br>CD212 (IL-12Rb1) PE 114 BD Biosciences 551974 (1:100)<br>CD24 PE M1/69 BD Biosciences 553262 (1:200)<br>CD25 BV605 PC61 BD Biosciences 563061 (1:200)<br>CD27 BUV737 LG.3A10 BD Biosciences 612831 (1:200)<br>CD4 BUV395 RM4.5 BD Biosciences 565974 (1:400)<br>CD44 APC-R700 IM7 Biolegend 565480 (1:300)<br>CD62L BV605 Mel-14 BD Biosciences 563252 (1:200)<br>CD62L PerCP-Cy5.5 Mel-14 Biolegend 104432 (1:200)<br>CD69 PE-Cy5 H1.2D3 Biolegend 104432 (1:200)<br>CD8a BV711 53-6.7 Biolegend 100748 (1:400)<br>CD8a FITC 53-6.7 BD Biosciences 553031 (1:400) |

CD8a RB744 53-6.7 BD Biosciences 570486 (1:400)
MHC-II (I-A/E) Alexa Fluor 700 M5/114.15.2 Invitrogen 56-5321-82 (1:400)
NK1.1 BUV563 PK136 BD Biosciences 741233 (1:400)
NK1.1 PerCP-Cy5.5 PK136 BD Biosciences 551114 (1:400)
Streptavidin PE - Invitrogen S866 (1.5uL/ug of monomer)
TCRb BV510 H57-597 BD Biosciences 563221 (1:400)
TCRb BV786 H57-597 BD Biosciences 568222 (1:400)
TCRb BUV805 H57-597 BD Biosciences 748405 (1:400)
TCRb PerCP-Cy5.5 H57-597 Biolegend 109227 (1:400)
TCRd BV421 GL3 BD Biosciences 562892 (1:200)
TCRd BV605 GL3 Biolegend 118129 (1:200)
Va8.3 PE T50 Biolegend 125707 (1:300)
Vb12 PE Mr11-1 Biolegend 139704 (1:300)
TCR Vg1 PE 2.11 Biolegend 141105 (1:300)
XCR1 BV785 ZET Biolegend 148225 (1:200)
IFNg BV480 XMG1.2 BD Biosciences 566097 (1:200)
IL-4 Alexa Fluor 647 11B11 BD Biosciences 557739 (1:75)
Tbet BV421 4B10 Biolegend 644832 (1:200)
Tbet RB613 4B10 BD Biosciences 571286 (1:200)
GATA3 PE-Cy7 L50-823 BD Biosciences 560405 (1:50)
CXCR5 PE-Dazzle594 L138D7 Biolegend 145522 (1:200)
PD-1 PE-Fire810 29F.1A12 Biolegend 135253 (1:200)
CD25 APC PC61.5 Invitrogen 17-0251-81 (1:200)
B220 n/a RA3-6B2 Invitrogen 14-0452-85 (1:500)

Validation

All antibodies are from commercial sources and have been validated by the vendors. Validation data are available on the manufacture's websites.
Antibodies were titrated by the authors to find the optimal dilution (amount) for staining.

# Eukaryotic cell lines

Policy information about cell lines and Sex and Gender in Research

Cell line source(s)

*State the source of each cell line used and the sex of all primary cell lines and cells derived from human participants or vertebrate models.*

Authentication

*Describe the authentication procedures for each cell line used OR declare that none of the cell lines used were authenticated.*

Mycoplasma contamination

*Confirm that all cell lines tested negative for mycoplasma contamination OR describe the results of the testing for mycoplasma contamination OR declare that the cell lines were not tested for mycoplasma contamination.*

Commonly misidentified lines
(See ICLAC register)

*Name any commonly misidentified cell lines used in the study and provide a rationale for their use.*

# Palaeontology and Archaeology

Specimen provenance

*Provide provenance information for specimens and describe permits that were obtained for the work (including the name of the issuing authority, the date of issue, and any identifying information). Permits should encompass collection and, where applicable, export.*

Specimen deposition

*Indicate where the specimens have been deposited to permit free access by other researchers.*

Dating methods

*If new dates are provided, describe how they were obtained (e.g. collection, storage, sample pretreatment and measurement), where they were obtained (i.e. lab name), the calibration program and the protocol for quality assurance OR state that no new dates are provided.*

☐ Tick this box to confirm that the raw and calibrated dates are available in the paper or in Supplementary Information.

Ethics oversight

*Identify the organization(s) that approved or provided guidance on the study protocol, OR state that no ethical approval or guidance was required and explain why not.*

Note that full information on the approval of the study protocol must also be provided in the manuscript.

# Animals and other research organisms

Policy information about studies involving animals; ARRIVE guidelines recommended for reporting animal research, and Sex and Gender in Research

Laboratory animals

C57BL/6, B6.SJL-PtprcaPep3b/BoyJ (CD45.1), PbT-I GFP, PbT-II GFP, Tcrd-/-, Trdctm1(EGFP/HBEGF/luc)Impr (TCRd-GDL), Il4-/-, Batf3-/-, Cd40l (Cd154)-/- , Il4ra-/-, 4C13R, Ifng-/-, Tcra-/-, Il12p40-/- mice were used as detailed in Methods under 'Mice'. All experimental mice were used at 6-12 weeks of age, or up to 20 weeks for chimeras. Mice were bred and maintained at the Peter

Doherty Institute for Infection and Immunity mouse facility and housed at 20–26°C, 45–65% humidity on a 12-h day–night light cycle. Mice used for parasite generation were were purchased from the Monash Animal Service and held at the School of Botany, The University of Melbourne. All mice were maintained on a standard chow diet.

| | |
|---|---|
| Wild animals | No wild animals were used. |
| Reporting on sex | Both male and female mice were used as Detailed in Methods under 'Mice'. All mice were age and sex matched within experiments. |
| Field-collected samples | No field-collected samples were used in this study. |
| Ethics oversight | Stated in the methods section under 'Mice.' All experimental work and animal handling was conducted in strict accordance with the standards approved by the Animal Ethics Committee at the University of Melbourne (ethic project ID: 2015168, 20088, 27552). |

Note that full information on the approval of the study protocol must also be provided in the manuscript.

# Clinical data

Policy information about clinical studies

All manuscripts should comply with the ICMJE guidelines for publication of clinical research and a completed CONSORT checklist must be included with all submissions.

| | |
|---|---|
| Clinical trial registration | This was not a clinical trial |
| Study protocol | PBMCs and plasma samples were collected during previously conducted trials in Timika, Papua, between 2004 and 2005. Timika is a lowland town located in the South-Central Papuan province of Indonesia. |
| Data collection | Malaria transmission is perennial in lowland Papua, with a prevalence of 28.3% in children under five years, 46.3% in children aged 5-15 years, and 36.8% in adults over 15 years. For the parent trials, adults with slide-confirmed malaria and fever, or a history of fever within the last 48 hours, were enrolled in randomised controlled trials of artemisinin combined therapy. Malaria parasite infection was categorised as P. falciparum or P. vivax mono-infection via microscopy. Exclusion criteria included pregnant or lactating women and children with a body weight of 10kg and under. In a subset of trial participants, blood samples were collected at enrolment for PBMC isolation (<20 mL). For the current study, PBMCs were selected from patients with parasite infection categorised as P. falciparum mono-infection via microscopy. |
| Outcomes | The primary outcome data was described in parent collection study (https://www.sciencedirect.com/science/article/pii/S0140673607601603?via%3Dihub). Here, the secondary outcome is to investigate the immune cell responses involved in the pathogenesis of Plasmodium falciparum infection as potential targets for immunotherapeutic interventions |

# Dual use research of concern

Policy information about dual use research of concern

## Hazards

Could the accidental, deliberate or reckless misuse of agents or technologies generated in the work, or the application of information presented in the manuscript, pose a threat to:

| No | Yes | |
|---|---|---|
| ☐ | ☐ | Public health |
| ☐ | ☐ | National security |
| ☐ | ☐ | Crops and/or livestock |
| ☐ | ☐ | Ecosystems |
| ☐ | ☐ | Any other significant area |

## Experiments of concern

Does the work involve any of these experiments of concern:

No | Yes

☐ ☐ Demonstrate how to render a vaccine ineffective

☐ ☐ Confer resistance to therapeutically useful antibiotics or antiviral agents

☐ ☐ Enhance the virulence of a pathogen or render a nonpathogen virulent

☐ ☐ Increase transmissibility of a pathogen

☐ ☐ Alter the host range of a pathogen

☐ ☐ Enable evasion of diagnostic/detection modalities

☐ ☐ Enable the weaponization of a biological agent or toxin

☐ ☐ Any other potentially harmful combination of experiments and agents

# Plants

**Seed stocks**

*Report on the source of all seed stocks or other plant material used. If applicable, state the seed stock centre and catalogue number. If plant specimens were collected from the field, describe the collection location, date and sampling procedures.*

**Novel plant genotypes**

*Describe the methods by which all novel plant genotypes were produced. This includes those generated by transgenic approaches, gene editing, chemical/radiation-based mutagenesis and hybridization. For transgenic lines, describe the transformation method, the number of independent lines analyzed and the generation upon which experiments were performed. For gene-edited lines, describe the editor used, the endogenous sequence targeted for editing, the targeting guide RNA sequence (if applicable) and how the editor was applied.*

**Authentication**

*Describe any authentication procedures for each seed stock used or novel genotype generated. Describe any experiments used to assess the effect of a mutation and, where applicable, how potential secondary effects (e.g. second site T-DNA insertions, mosiacism, off-target gene editing) were examined.*

# ChIP-seq

## Data deposition

☐ Confirm that both raw and final processed data have been deposited in a public database such as GEO.

☐ Confirm that you have deposited or provided access to graph files (e.g. BED files) for the called peaks.

**Data access links**
*May remain private before publication.*

*For "Initial submission" or "Revised version" documents, provide reviewer access links. For your "Final submission" document, provide a link to the deposited data.*

**Files in database submission**

*Provide a list of all files available in the database submission.*

**Genome browser session**
(e.g. UCSC)

*Provide a link to an anonymized genome browser session for "Initial submission" and "Revised version" documents only, to enable peer review. Write "no longer applicable" for "Final submission" documents.*

## Methodology

**Replicates**

*Describe the experimental replicates, specifying number, type and replicate agreement.*

**Sequencing depth**

*Describe the sequencing depth for each experiment, providing the total number of reads, uniquely mapped reads, length of reads and whether they were paired- or single-end.*

**Antibodies**

*Describe the antibodies used for the ChIP-seq experiments; as applicable, provide supplier name, catalog number, clone name, and lot number.*

**Peak calling parameters**

*Specify the command line program and parameters used for read mapping and peak calling, including the ChIP, control and index files used.*

**Data quality**

*Describe the methods used to ensure data quality in full detail, including how many peaks are at FDR 5% and above 5-fold enrichment.*

**Software**

*Describe the software used to collect and analyze the ChIP-seq data. For custom code that has been deposited into a community repository, provide accession details.*

# Flow Cytometry

## Plots

Confirm that:

☒ The axis labels state the marker and fluorochrome used (e.g. CD4-FITC).

☒ The axis scales are clearly visible. Include numbers along axes only for bottom left plot of group (a 'group' is an analysis of identical markers).

☒ All plots are contour plots with outliers or pseudocolor plots.

☒ A numerical value for number of cells or percentage (with statistics) is provided.

## Methodology

| | |
|---|---|
| Sample preparation | Described in the Methods section |
| Instrument | Cells were acquired on a Cytek Aurora using the SpectroFlo v3.0 system (Cytek) and further analysed on FlowJo (TreeStar, Inc) |
| Software | SpectroFlo v3.0 system (Cytek) and further analysed on FlowJo (TreeStar, Inc) |
| Cell population abundance | Absolute numbers of cells per organ are shown in most instances, with gating strategies described or shown. Cell sort purities for Figure 6 were above 95% pure (of live cells). |
| Gating strategy | Gating strategies are described in the text or the figure legends or gating strategies are shown. Typically, lymphocytes are first gated using FSC-A vs SSC-A, followed by doublet exclusion using FSC-A/H. Dead cells were then excluded by Live-Dead Blue or NIR staining followed by specific gating per experiment to detect target cells. |

☒ Tick this box to confirm that a figure exemplifying the gating strategy is provided in the Supplementary Information.

# Magnetic resonance imaging

## Experimental design

| | |
|---|---|
| Design type | *Indicate task or resting state; event-related or block design.* |
| Design specifications | *Specify the number of blocks, trials or experimental units per session and/or subject, and specify the length of each trial or block (if trials are blocked) and interval between trials.* |
| Behavioral performance measures | *State number and/or type of variables recorded (e.g. correct button press, response time) and what statistics were used to establish that the subjects were performing the task as expected (e.g. mean, range, and/or standard deviation across subjects).* |

## Acquisition

| | |
|---|---|
| Imaging type(s) | *Specify: functional, structural, diffusion, perfusion.* |
| Field strength | *Specify in Tesla* |
| Sequence & imaging parameters | *Specify the pulse sequence type (gradient echo, spin echo, etc.), imaging type (EPI, spiral, etc.), field of view, matrix size, slice thickness, orientation and TE/TR/flip angle.* |
| Area of acquisition | *State whether a whole brain scan was used OR define the area of acquisition, describing how the region was determined.* |

Diffusion MRI   ☐ Used   ☐ Not used

## Preprocessing

| | |
|---|---|
| Preprocessing software | *Provide detail on software version and revision number and on specific parameters (model/functions, brain extraction, segmentation, smoothing kernel size, etc.).* |
| Normalization | *If data were normalized/standardized, describe the approach(es): specify linear or non-linear and define image types used for transformation OR indicate that data were not normalized and explain rationale for lack of normalization.* |
| Normalization template | *Describe the template used for normalization/transformation, specifying subject space or group standardized space (e.g. original Talairach, MNI305, ICBM152) OR indicate that the data were not normalized.* |
| Noise and artifact removal | *Describe your procedure(s) for artifact and structured noise removal, specifying motion parameters, tissue signals and physiological signals (heart rate, respiration).* |

| Volume censoring | *Define your software and/or method and criteria for volume censoring, and state the extent of such censoring.* |
|---|---|

## Statistical modeling & inference

| Model type and settings | *Specify type (mass univariate, multivariate, RSA, predictive, etc.) and describe essential details of the model at the first and second levels (e.g. fixed, random or mixed effects; drift or auto-correlation).* |
|---|---|
| Effect(s) tested | *Define precise effect in terms of the task or stimulus conditions instead of psychological concepts and indicate whether ANOVA or factorial designs were used.* |

Specify type of analysis: ☐ Whole brain ☐ ROI-based ☐ Both

| Statistic type for inference<br><br>(See Eklund et al. 2016) | *Specify voxel-wise or cluster-wise and report all relevant parameters for cluster-wise methods.* |
|---|---|
| Correction | *Describe the type of correction and how it is obtained for multiple comparisons (e.g. FWE, FDR, permutation or Monte Carlo).* |

## Models & analysis

n/a | Involved in the study
☐ | ☐ Functional and/or effective connectivity
☐ | ☐ Graph analysis
☐ | ☐ Multivariate modeling or predictive analysis

| Functional and/or effective connectivity | *Report the measures of dependence used and the model details (e.g. Pearson correlation, partial correlation, mutual information).* |
|---|---|
| Graph analysis | *Report the dependent variable and connectivity measure, specifying weighted graph or binarized graph, subject- or group-level, and the global and/or node summaries used (e.g. clustering coefficient, efficiency, etc.).* |
| Multivariate modeling and predictive analysis | *Specify independent variables, features extraction and dimension reduction, model, training and evaluation metrics.* |

