## [Peer Review File · Nature Immunology]

$\gamma\delta$ T cell-derived IL-4 initiates CD8 T cell immunity

Corresponding Author: Dr Lynette Beattie

Version 0:

Reviewer comments:

Reviewer #1

(Remarks to the Author)

This is a relevant paper describing a new contribution of gamma-delta ($\gamma\delta$) T cells to the immune response against Plasmodium in mice. The experiments are in general novel and interesting, although data analysis, presentation and description could be significantly improved, as detailed below. The major drawback of the study is the lack of evidence for (substantial) production of IL-4 by $\gamma\delta$ T cells, which would be essential to support the proposed model linking to Plasmodium antigen-specific CD8⁺ T cells. This is also critical to rule out that IL-4 is produced by other cells which could act upstream of the $\gamma\delta$ T cell response.

General issues:

- It seems that no littermate controls were used throughout the study – can the authors comment on this, given the importance of normalizing the microbiome between groups to study immune responses to specific microorganisms?
- A more detailed reasoning should be provided for the models employed; examples: the use of TCRd-GDL mice in Fig 2; or the use of OT-I cells in Fig 3 – namely, what was the advantage compared to the previous PbT-I model?
- The authors should also discuss the physiological relevance and the limitations of their system, especially regarding the dynamics of the observed response in the spleen in the first 24h (versus the natural course of the disease that includes a 48h liver stage). In fact, what is happening at early time points in the liver with the resident $\gamma\delta$ T cells?
- The observation that IL-4 signaling in PbT-1 cells results in IL-12R induction in vivo should be strengthened experimentally to support the working model. Ideally, in Fig 7, authors should assess the expression of IL-12R in PbT-1 cells transferred into TCRd KO recipients reconstituted with either WT or IL-4 KO $\gamma\delta$ T cells (as in Figure 4, panel k).

SPECIFIC COMMENTS ON THE TEXT:

Lines 99-101: The T cell response to intravenously injected RAS is initiated largely in the spleen with priming and expansion occurring in this site, followed by recirculation and resultant rapid accumulation of activated T cells in the liver¹⁸. In the mentioned referenced, the following is stated: “the spleen is the main site for priming sporozoite specific T cells after intravenous administration of parasites, but they do not formally exclude the liver draining lymph nodes or the liver as important sites of activation for protective immunity”. Indeed, since a role for $\gamma\delta$ T cells in the liver stage has been proposed following attenuated PbA sporozoite infection (Chora et al. Immunity 2023), it would be interesting to address the contribution of liver versus spleen $\gamma\delta$ T cells to the accumulation of PbT-I cells in the liver/spleen. Given that the PbT-1 accumulation phenotype does not happen upon infection with iRBCs, and $\gamma\delta$ T cells are essential in the first 24h for later PbT-I cell accumulation in the spleen, the role of $\gamma\delta$ T cells in liver stage should be addressed. It would be interesting to assess the accumulation of PbT-I cells in the liver of WT or TCRd KO splenectomized mice, for example. This becomes even more apparent at Fig 2, where the authors conclude that splenic Vg1⁺ $\gamma\delta$ T cells initiate immunity to RAS (given their increased frequencies of CD69, CD25 and CD11a after 24h of RAS infection), without analysing the liver compartment. If all the phenotype is due to $\gamma\delta$ cells in the spleen in the first 24h, this may question the physiological relevance of the model, given that “efficient priming via this route [spleen] most likely derives from the large load of irradiated sporozoites deposited in the spleen after intravenous administration and the high frequency of T cells found in this organ” [from Lei Shong Lau et al. PLOS Pathogens 2014, where PbT-I transgenic line generation was described]?

Lines 216-219 – there is a $\gamma\delta$ contribution to PbT-I cell accumulation (Fig 4h) that is not IL-4 dependent (comparison between groups 5 and 6), which should be discussed; additionally, IL-4 has also been shown to drive the expansion of some

gd T cell subsets; how does this data exclude that IL-4 is acting upstream of gd T cell activation? This could be tested (related to Fig 4k) by reconstituting TCRd KO mice with gd T cells from IL4Ra KO.

Line 241 – it is mentioned that IFN-g, unlike IL-4, is not largely produced by gd T cells expressing CXCR5 and/or CCR7, but the data shows that those subsets produce both cytokines at similar frequencies. It is true that CXCR5- and/or CCR7- have increased frequencies of IFN-g production compared to their positive counterparts, but the way it is written is not accurate. Additionally, in lines 366-367 of discussion, it is stated that Vg9+Vd2+ gd T cells produced IL-4, but not IFN-g or TNF following restimulation, which is not at all supported by the data (Fig 4m).

Lines 291-295 – IL-12 blockade is not specific to cDC1; also, it would be important to assess IL-12 production by cDC1 in TCRd KO mice to support the proposed model.

SPECIFIC COMMENTS ON THE FIGURES:

Figure 1, panels e and f – Y scales should be adjusted (potentially in Log scale?) to facilitate reading. Same should be done for Supp. Figure 1, panels g and h. Also in Supp. Figure 1, panel g it is not clear if the statistics presented refers to all CD8 T cell memory subsets – Tcm differences look they might be significant?

Figure 1, panel f – for consistency, the authors should show the absolute number of PbT-1 cells in the spleens of anti-Vg1 and anti-GD antibodies, and not the values normalized to control spleens.

Fig 1e,f and Supp. Fig 1g,h – A more detailed gating strategy for memory T cells should be provided.

Supp Fig 1b is not referenced in the manuscript.

Supp. Figure 2, panel a - for consistency, the frequency of CD69+ CD25+ cells within Vg1- gd T cells should be added to the plots.

Fig 2c – FACS plots are not representative of frequencies in Fig2d; if data in Fig2d are normalized, it should be stated in the plot or caption.

Figure 3, panel a, the PbT-1 proliferation fails to distinguish discrete populations undergoing 1, 2 etc divisions. Why is this? Also, the frequency of proliferating OT-1 cells transferred to WT or TCRd KO mice should be presented and potentially have a separate graph with the mean values for the two different conditions, to assess if the cells transferred into WT recipients have increased proliferation compared to those transferred into TCRd KO mice.

Fig 3a-c – are these CD44+ OT-I cells? If so, is the overall frequency of OT-I or CD44 expression similar between B6 and TCRd KO?

Fig 4d – the fact that PbT-II CD4 T cells are completely dependent on gd T cells but not IL-4 is not discussed, and it is a missing link in the final working model of Fig 7m – an extra arrow with a '?' should be added?

Fig 4f-h – Group 4 and 6 of the table of fig 4f are switched in figs 4g,h; additionally, it would be interesting to see the reconstitution of the gd liver compartment as shown for spleen in fig 4g.

Figure 4, panel h - in the BM Chimera experiments, to claim gd T cells are the source of IL-4 based on such comparisons, the recipient mice should be IL-4 KO mice instead of WT B6 mice.

Figure 4, panel j and k, the legend on the third bar should be "no transfer" as is in the legend?

Fig 4k – it would be good to show absolute numbers, given that the transfer of gd from IL-4 KO mice still supports half the accumulation of PbT-I cells of transfer with gd cells from B6 mice. Is that half comparable to PbT-I cell numbers in TCRd KO mice?

Fig S4d – all p-values presented have the same value; is this so or a mistake?

Fig S5b – these data on XCR1 expression are not mentioned in the manuscript?

Fig 6h is referenced in the manuscript, but it is not shown?

Reviewer #2

(Remarks to the Author)

<Reviewer comments to Authors>

Previous studies have suggested that the establishment of malaria immunity induced by sporozoite vaccines is positively correlated with the degree of $\gamma\delta$ T cell expansion (Rer40: Seder et al., 2013 Science). However, how $\gamma\delta$ T cells contribute to the establishment of malaria immunity induced by sporozoite vaccines has remained unclear.

This manuscript showed the importance of $\gamma\delta$ T cells in the expansion of Plasmodium-specific CD8+ T cells in response to RAS immunization and the following liver accumulation to develop into resident memory T cells. Authors tried to prove that $\gamma\delta$ T cell-derived IL-4 was required both to affect IL-12 production in cDC1 and directly affect priming of CD8+ T cells alongside IL-12. Moreover, they showed that $\gamma\delta$ T cell-derived human malaria patients produce IL-4. The experiment is well-conducted, and I was intrigued by this manuscript. However, Zavala groups has already reported that IL-4 is required for protective immunity against liver-stage Plasmodium infection (Ref 16: Carvalho et al., 2002 Nat Med). They also showed that IL-4 signaling in CD8+ T cells is required for the protective immunity (Ref 38: Morrot et al., 2005 JEM). Furthermore, it is known that $\gamma\delta$ T cells can produce IL-4 in both mice and humans (Ref 24: Gerber et al., 1999 J Immunol). At present, it seems that this paper lacks sufficient novelty and impact to be published in Nature Immunology. However, this paper is valuable as it elucidates an aspect of the role of $\gamma\delta$ T cells in RAS vaccination-induced immunity against liver stage malaria, which has been ambiguous until now. I believe that this manuscript will be improved by additional experiments and more detailed analyses as pointed out below.

The authors demonstrate that the absence of an IL-4 signal to cDC1 reduces PbTI (CD8+ T) cell priming efficiency (Fig. 5). This will be attributed to a lack of IL-12 production from cDC1 without IL-4 signaling (Fig. 6), although cDC1 cell numbers

remain unaffected (Fig. 5e). Furthermore, Figure 4d shows no impact on PbTII (CD4+ T) cells even in an IL-4-deficient state. While Figure 7m provides a summarized image, the actual results suggest a more complex scenario. For instance, despite reduced cDC1-derived IL-12 levels in the absence of IL-4, PbTII cell priming remains unaffected. This implies that these PbTII cells differentiate into Tfh rather than Th1 cells. The Tfh PbTII cells might provide CD40L stimulation to cDC1.

Regarding with CD4+ T Cells (PbTII Cells):

What polarization of CD4+ T cells (Th1, Th2, Tfh) is responsible for producing CD40L in this situation? How does the presence or absence of IL-4 affect this polarization? IL-4 promotes Th2 and Tfh polarization, while IL-12 drives Th1 polarization. Figure 4d shows that $\gamma\delta$ T cells do not affect PbTII (CD4+ T) cells even in an IL-4^{-/-} state, but was there any change in polarization?

Regarding with cDC1:

Figure 6 presents in vitro culture data and bulk RNA sequencing data for cDC1. While it is surprising that IL-4 and CD40 stimulation increase dendritic cell IL-12 production, the actual in vivo situation is more complex. As shown in Figure S4c, $\gamma\delta$ T cells produce not only IL-4 but predominantly IFN γ . This suggests that IFN γ stimulation should also be considered. For example, during blood-stage infection, $\gamma\delta$ T cells are known to activate cDC1 via IFN γ (Yarob et al., Front Immunol 2024). Regarding CD4+ T cells, it can be inferred that IFN γ -producing $\gamma\delta$ T cells enable cDC1 to produce IL-12. In my opinion, authors should do comparative analysis of IFN- γ -producing and IL-4-producing $\gamma\delta$ T cell fractions (single producers or double producers?).

Moreover, single-cell RNA sequencing of cDC1 during IL-4 inhibition combined with RAS vaccination. If IL-4+ $\gamma\delta$ T cells and IFN- γ + $\gamma\delta$ T cells are distinct populations, cDC1 may split into groups receiving different signals (IFN- γ receptor-positive vs. IL-4 receptor-positive populations). Even if most IL-4+ $\gamma\delta$ T cells also produce IFN- γ , there should still be IFN γ + IL-4- population of $\gamma\delta$ T cells, suggesting multiple cDC1 subsets. Single-cell RNA sequencing is necessary to identify these complex populations. Additionally, it is possible that the cDC1 activating CD4+ T cells differs from the one activating CD8+ T cells.

<the other comments>

All; CD4 T cells should be CD4+ T cells. CD8 T cells should be CD8+ T cells.

Line 78; Not only Ref 20 but also the other precise reference that showed "in human vaccination, $\gamma\delta$ T cells are associated with the success of this response" should be added.

Line 107; the first " δ -/" should be "TCR δ -/" in somewhere.

Lines 122-125; Is this effect limited to irradiated parasites? How about genetically modified attenuated sporozoites (lisp2-, uis4-), or wild type parasites?

Line 133/Figure1 and FigureS1; I wonder why authors showed PbTII and PbTI cells/spleen in main Figure1 and PbTII and PbTI cells/liver in Supplementary Figure. Figure 1 title is "~liver TRM cell generation."

Line 147/Figure 2; Even in Naïve mice, most splenic $\gamma\delta$ T cells expressed CD11a. CD11a is actually upregulated after RAS vaccination?? I understand that CD25 and CD69 are useful for activation marker for $\gamma\delta$ T cells. But Authors did not show the data of upregulation of CD11a in $\gamma\delta$ T cells.

Lines 155-171; Why not use the same PbT-I cells and RAS for this experiment?

Line 174-175; Previous paper (Inoue et al., 2012 PNAS) showed that $\gamma\delta$ T cells provide CD40L for signalling CD40 on cDC1, a signal known to be essential in the response to RAS. Authors should mention or just refer the paper in the sentence. Related with this, I'm curious about CD40L expression in $\gamma\delta$ T cells and CD4+ T cells were occurred after RAS vaccination or not. I'm sure that Figure 3e implied that $\gamma\delta$ T cells might not express CD40L. But that is not direct evidence.

Lines 221-228, 367-369; Authors should show IL-4 expression from splenic $\gamma\delta$ T cells after vaccination clearly in mouse model.

Lines 240-241; Around 30% of CXCR5+CCR7+ cells were IFN γ +, similar to IL-4+. Authors should rephrase more accurately

Lines 280-289; Authors should show IL-12 on protein level and also show whether that is reflected on cDC1 in vivo in different conditions (TCRdKO, IL-4 blockade)

Line 347; I could not find the data "In this vaccination setting, IL-4 is produced by V γ 1+ $\gamma\delta$ T cells in the spleen"

Lines 449 and 455; sporozoites and infected RBCs were irradiated with 20K rads. Is this information described in Ref 61? If not, authors should add Ref.

Lines 477-480, Figure 4i; Negative enrichment was used to isolate $\gamma\delta$ T cells. Possibility of contamination with another subset. Is there any contamination with the other cell subsets. E.g. NK cells. Authors should show how match enriched by this method. Is there any problem if you use positive selection?? Did you forget to add Gr1 or is there any meaning without Gr1? (Gr1 was used in T cell enrichment)

Lines 465 and 479; F4/80 is not Mac1/CD11b. It should be separated from there.

Lines 561 and 563; "TCRd -/-" should be "TCR δ -/-".

Line 575; In vivo antibody depletion/blockade. Authors injected antibody via two ways (ip and iv). If the route of injection is important, provide the reasons for it.

Related with this, authors should add the information (ip or iv) in Figures/Figure Legends.

Figure1 and FigureS1; Authors showed two time-points only. Thus, the time-points should be unified. "day23" in Figure1 and "day35" in FigureS1. Those are different.

Figure1 and FigureS1; Authors should clarify what the statistical test was used.

Figure S1; Authors should explain what is "GFP" in Fig S1b.

Figure S1, S2, 3, S5; proportions of cell populations should be added even in representative flow plots. The information is helpful to understand the data.

Figure 2 and S2; Author should explain what is TCR δ -GDL and why you used the mice.

Figure 2 and S2; I'm curious about impact on numbers of whole $\gamma\delta$ T cells and V γ 1+ $\gamma\delta$ T cells after RAS vaccination (not only CD11a+CD25+CD69+ gated cells).

Figure 2c ,d; in Fig 2d, %CD11a+CD69+CD25+ $\gamma\delta$ T cells is 0.75% (V γ 1+) in average. This is not match with Fig2 c (9.89% in RAS). Why?

Figure 2f; Why is this the only figure "normalized to control" and not absolute numbers?

Figure 4f, g; Mouse group numbers are mixed up (4 and 6)

Figure 6; I wonder why B16 Flt2L(sc) was used in this experiment. There is no explanation.

Figure6 a-e; CD40 should be α CD40

Figure 6b; Heat map should start with control (media) Same order as in 6a.

Figure 6d; Should mention both conditions under x axis (CD40 vs CD40+IL-4)

Figure 7a-f; How was (STAT3, 4, 6) signaling impacted?

Figure 7g; No details, Number of cells, CD44+?

Reviewer #3

(Remarks to the Author)

The study by Le et al is very nice and should be published in Nature Immunology. I think the case is made very strongly that IL-4 made by gamma/delta cells reactive to RAS contributes to CD8 responses both directly on T cells and through actions on cDC1. The figures are well-structured and easy to follow, have large enough sample sizes for and cover the various steps in the model. The study both confirms some previous facts, such as the role of gdT cells and IL-4 in the response as well as extending the work to propose a new model for mechanism. The topic of how the immune system responds to plasmodia is very important since there are really no effective vaccines against plasmodia (yet), so that work like this could help in that regard eventually.

The authors use an immunization model with radiation-attenuated Plasmodium sporozoites (RAS) and transfer of antigen-specific transgenic T-cells. They show that in this model, the V γ 1+ gd T-cells and, in particular, their IL-4 production are crucial for the expansion of the transferred CD8+ T-cells. Evidence is given using CRISPR'd transferred cells that IL-4 is acts both directly CD8 T-cells and on cDC1 and stimulates DCs to induce IL-12 receptor and IL-12 production, with IL-4 synergizing with IL-12 to expand the T-cells. I particularly like the analysis in Fig. 5 and 6 showing the impact at a molecular level for the CD40 + IL-4 signaling in DCs.

My vote would be to publish this immediately without any more experiments. The case is made for the new model, and additional experiment could strengthen the conclusions (maybe) but I don't think its necessary. For example, a reviewer might ask for more high-tech specific evidence, such as CD8-specific Cre crossed to IL-4Ra floxed allele in mice. But I don't think that's necessary. A reviewer asking for that might just want to prevent this result from getting published any time soon. Or you could ask the authors to delve deeper into how IL-4 plus CD40 induces the changes (is it STAT6 or IRS pathway), asking if it is unique or whether other cytokines can or can't do it, but that's a different issue. Some people might criticize the model saying that it is artificial or that the authors don't only look at the CD8 T-cells as physiologic readout, and not the survival or something. I would challenge that point of view. Every model is artificial and CD8 expansion is likely to be highly

relevant to the outcome.

In short, I support the publication of this study without new experiments. It's solid, high quality and important, and provides a new twist in DC biology. It's very interesting to think of this model in the evolutionary context. My colleague Osami Kanagawa always referred to IL-4 as the 'happiness factor', and this is pertinent here. The specificity is in the gdT cell reacting to RAS. That's the danger signal. Then the happiness factor is secreted to enable augmented responses both in the priming APC (inducing the IL-12) and the CD8 T cell directly. My guess is that the IL-4 levels are in the happiness range and not in the TH2-inducing high dose range of GATA-3 autoactivation. So it's interesting that the old innate style gdT cells side reserves that mechanism even now. Just musing.

Minor Comments

Figure 2d: what is the reference population for the % value (Total T-cells? A particular population of T-cells?)

Figure 3: The figure title is a bit misleading, as the figure doesn't really examine the CD40-CD40L signaling. The figure shows a negative result for a measurement (of T cells) in CD40L knockouts. This is an interpretation that CD40 signaling is intact. I would alter the words for more accuracy.

Figure 4: It is too bad that IL-4 is so hard to measure in the mouse V γ 1+ cells. In 4 f/g/h, groups 4 and 6 seem to be mixed between the table and graphics. I would check this for accuracy.

Figure 5+6: The cells transferred at day -3 in Figure 5 are only strongly enriched in DC1 (60-80%, according to methods). The cells in Figure 6 were sorted, but even with sorting, there is no 100% purity. Is it possible that e.g. DC2 and not DC1 are the population responding to IL-4 treatment? I doubt it, but maybe the authors could add some text to discuss this. The BATF3 knockout result is the key here.

Regarding Fig 6 b and d, could the authors make a supplementary data item, for example a table, containing data on the most up/downregulated genes.

Figure S5: Do the Pb-T-1 cells also express the IL-4Ra on day 0 when, according to Fig 2, the γ dT cells are required

Decision Letter:

13th Mar 2025

Dear Dr Beattie,

As you are already aware, your Article, " γ dT cell-derived IL-4 initiates CD8 T cell immunity" has now been seen by 3 referees (reports below).

While they find your work of considerable potential interest, they have raised substantial concerns that must be addressed. In light of these comments, and looking over your Author Response document (thanks for sending that to me), we are interested in considering a revised version in line with your revision plan. We appreciate the difficulty here, but please note that we do expect that the IL-4 reporter mouse will need to generate some positive data, as the issue raised by the reviewer with regards to production of that cytokine by the gamma delta T cells seems to us to be a critical issue.

If you choose to revise your manuscript taking into account all reviewer and editor comments, please highlight all changes in the manuscript text file in Microsoft Word format.

* If you have not done so already please begin to revise your manuscript so that it conforms to our Article format instructions at <http://www.nature.com/ni/authors/index.html>. Refer also to any guidelines provided in this letter.

The Reporting Summary can be found here:
<https://www.nature.com/documents/nr-reporting-summary.pdf>

When submitting the revised version of your manuscript, please pay close attention to our

<https://www.nature.com/nature-portfolio/editorial-policies/image-integrity>>Digital Image Integrity Guidelines. and to the following points below:

Extended Data figures and tables are online-only (appearing in the online PDF and full-text HTML version of the paper), peer-reviewed display items that provide essential background to the Article but are not included in the printed version of the paper due to space constraints or being of interest only to a few specialists. A maximum of ten Extended Data display items (figures and tables) is typically permitted. When re-submitting your manuscript, please ensure that any supplementary figures and tables that are more critical to the manuscript's conclusions are converted to Extended data to increase these data's visibility.

Link Redacted

If you wish to submit a suitably revised manuscript we would hope to receive it within 6 months. If you cannot send it within this time, please let us know. We will be happy to consider your revision so long as nothing similar has been accepted for publication at Nature Immunology or published elsewhere.

Nature Immunology is committed to improving transparency in authorship. As part of our efforts in this direction, we are now requesting that all authors identified as 'corresponding author' on published papers create and link their Open Researcher and Contributor Identifier (ORCID) with their account on the Manuscript Tracking System (MTS), prior to acceptance. ORCID helps the scientific community achieve unambiguous attribution of all scholarly contributions. You can create and link your ORCID from the home page of the MTS by clicking on 'Modify my Springer Nature account'. For more information please visit <http://www.springernature.com/orcid>.

Thank you for the opportunity to review your work.

Sincerely,

Nick Bernard, PhD
Senior Editor
Nature Immunology

Reviewers' Comments:

Reviewer #1 (Remarks to the Author):

This is a relevant paper describing a new contribution of gamma-delta (gd) T cells to the immune response against Plasmodium in mice. The experiments are in general novel and interesting, although data analysis, presentation and description could be significantly improved, as detailed below. The major drawback of the study is the lack of evidence for (substantial) production of IL-4 by gd T cells, which would be essential to support the proposed model linking to Plasmodium antigen-specific CD8+ T cells. This is also critical to rule out that IL-4 is produced by other cells which could act upstream of the gd T cell response.

General issues:

- It seems that no littermate controls were used throughout the study – can the authors comment on this, given the importance of normalizing the microbiome between groups to study immune responses to specific microorganisms?
- A more detailed reasoning should be provided for the models employed; examples: the use of TCRd-GDL mice in Fig 2; or the use of OT-I cells in Fig 3 – namely, what was the advantage compared to the previous PbT-I model?
- The authors should also discuss the physiological relevance and the limitations of their system, especially regarding the

dynamics of the observed response in the spleen in the first 24h (versus the natural course of the disease that includes a 48h liver stage). In fact, what is happening at early time points in the liver with the resident gd T cells?

- The observation that IL-4 signaling in PbT-1 cells results in IL-12R induction in vivo should be strengthened experimentally to support the working model. Ideally, in Fig 7, authors should assess the expression of IL-12R in PbT-1 cells transferred into TCRd KO recipients reconstituted with either WT or IL-4 KO gd T cells (as in Figure 4, panel k).

SPECIFIC COMMENTS ON THE TEXT:

Lines 99-101: The T cell response to intravenously injected RAS is initiated largely in the spleen with priming and expansion occurring in this site, followed by recirculation and resultant rapid accumulation of activated T cells in the liver. In the mentioned referenced, the following is stated: "the spleen is the main site for priming sporozoite specific T cells after intravenous administration of parasites, but they do not formally exclude the liver draining lymph nodes or the liver as important sites of activation for protective immunity". Indeed, since a role for gd T cells in the liver stage has been proposed following attenuated PbA sporozoite infection (Chora et al. *Immunity* 2023), it would be interesting to address the contribution of liver versus spleen gd T cells to the accumulation of PbT-1 cells in the liver/spleen. Given that the PbT-1 accumulation phenotype does not happen upon infection with iRBCs, and gd T cells are essential in the first 24h for later PbT-1 cell accumulation in the spleen, the role of gd T cells in liver stage should be addressed. It would be interesting to assess the accumulation of PbT-1 cells in the liver of WT or TCRd KO splenectomized mice, for example. This becomes even more apparent at Fig 2, where the authors conclude that splenic Vg1+ gd T cells initiate immunity to RAS (given their increased frequencies of CD69, CD25 and CD11a after 24h of RAS infection), without analysing the liver compartment. If all the phenotype is due to gd cells in the spleen in the first 24h, this may question the physiological relevance of the model, given that "efficient priming via this route [spleen] most likely derives from the large load of irradiated sporozoites deposited in the spleen after intravenous administration and the high frequency of T cells found in this organ" [from Lei Shong Lau et al. *PLOS Pathogens* 2014, where PbT-1 transgenic line generation was described]?

Lines 216-219 – there is a gd contribution to PbT-1 cell accumulation (Fig 4h) that is not IL-4 dependent (comparison between groups 5 and 6), which should be discussed; additionally, IL-4 has also been shown to drive the expansion of some gd T cell subsets; how does this data exclude that IL-4 is acting upstream of gd T cell activation? This could be tested (related to Fig 4k) by reconstituting TCRd KO mice with gd T cells from IL4Ra KO.

Line 241 – it is mentioned that IFN-g, unlike IL-4, is not largely produced by gd T cells expressing CXCR5 and/or CCR7, but the data shows that those subsets produce both cytokines at similar frequencies. It is true that CXCR5- and/or CCR7- have increased frequencies of IFN-g production compared to their positive counterparts, but the way it is written is not accurate. Additionally, in lines 366-367 of discussion, it is stated that Vg9+Vd2+ gd T cells produced IL-4, but not IFN-g or TNF following restimulation, which is not at all supported by the data (Fig 4m).

Lines 291-295 – IL-12 blockade is not specific to cDC1; also, it would be important to assess IL-12 production by cDC1 in TCRd KO mice to support the proposed model.

SPECIFIC COMMENTS ON THE FIGURES:

Figure 1, panels e and f – Y scales should be adjusted (potentially in Log scale?) to facilitate reading. Same should be done for Supp. Figure 1, panels g and h. Also in Supp. Figure 1, panel g it is not clear if the statistics presented refers to all CD8 T cell memory subsets – Tcm differences look they might be significant?

Figure 1, panel f – for consistency, the authors should show the absolute number of PbT-1 cells in the spleens of anti-Vg1 and anti-GD antibodies, and not the values normalized to control spleens.

Fig 1e,f and Supp. Fig 1g,h – A more detailed gating strategy for memory T cells should be provided.

Supp Fig 1b is not referenced in the manuscript.

Supp. Figure 2, panel a - for consistency, the frequency of CD69+ CD25+ cells within Vg1- gd T cells should be added to the plots.

Fig 2c – FACS plots are not representative of frequencies in Fig2d; if data in Fig2d are normalized, it should be stated in the plot or caption.

Figure 3, panel a, the PbT-1 proliferation fails to distinguish discrete populations undergoing 1, 2 etc divisions. Why is this? Also, the frequency of proliferating OT-1 cells transferred to WT or TCRd KO mice should be presented and potentially have a separate graph with the mean values for the two different conditions, to assess if the cells transferred into WT recipients have increased proliferation compared to those transferred into TCRd KO mice.

Fig 3a-c – are these CD44+ OT-I cells? If so, is the overall frequency of OT-I or CD44 expression similar between B6 and TCRd KO?

Fig 4d – the fact that PbT-II CD4 T cells are completely dependent on gd T cells but not IL-4 is not discussed, and it is a missing link in the final working model of Fig 7m – an extra arrow with a '?' should be added?

Fig 4f-h – Group 4 and 6 of the table of fig 4f are switched in figs 4g,h; additionally, it would be interesting to see the reconstitution of the gd liver compartment as shown for spleen in fig 4g.

Figure 4, panel h - in the BM Chimera experiments, to claim gd T cells are the source of IL-4 based on such comparisons, the recipient mice should be IL-4 KO mice instead of WT B6 mice.

Figure 4, panel j and k, the legend on the third bar should be "no transfer" as is in the legend?

Fig 4k – it would be good to show absolute numbers, given that the transfer of gd from IL-4 KO mice still supports half the accumulation of PbT-1 cells of transfer with gd cells from B6 mice. Is that half comparable to PbT-1 cell numbers in TCRd KO mice?

Fig S4d – all p-values presented have the same value; is this so or a mistake?

Fig S5b – these data on XCR1 expression are not mentioned in the manuscript?
Fig 6h is referenced in the manuscript, but it is not shown?

Reviewer #2 (Remarks to the Author):

<Reviewer comments to Authors>

Previous studies have suggested that the establishment of malaria immunity induced by sporozoite vaccines is positively correlated with the degree of $\gamma\delta$ T cell expansion (Rer40: Seder et al., 2013 Science). However, how $\gamma\delta$ T cells contribute to the establishment of malaria immunity induced by sporozoite vaccines has remained unclear.

This manuscript showed the importance of $\gamma\delta$ T cells in the expansion of Plasmodium-specific CD8+ T cells in response to RAS immunization and the following liver accumulation to develop into resident memory T cells. Authors tried to prove that $\gamma\delta$ T cell-derived IL-4 was required both to affect IL-12 production in cDC1 and directly affect priming of CD8+ T cells alongside IL-12. Moreover, they showed that $\gamma\delta$ T cell-derived human malaria patients produce IL-4. The experiment is well-conducted, and I was intrigued by this manuscript. However, Zavala groups has already reported that IL-4 is required for protective immunity against liver-stage Plasmodium infection (Ref 16: Carvalho et al., 2002 Nat Med). They also showed that IL-4 signaling in CD8+ T cells is required for the protective immunity (Ref 38: Morrot et al., 2005 JEM). Furthermore, it is known that $\gamma\delta$ T cells can produce IL-4 in both mice and humans (Ref 24: Gerber et al., 1999 J Immunol). At present, it seems that this paper lacks sufficient novelty and impact to be published in Nature Immunology. However, this paper is valuable as it elucidates an aspect of the role of $\gamma\delta$ T cells in RAS vaccination-induced immunity against liver stage malaria, which has been ambiguous until now. I believe that this manuscript will be improved by additional experiments and more detailed analyses as pointed out below.

The authors demonstrate that the absence of an IL-4 signal to cDC1 reduces PbTI (CD8+ T) cell priming efficiency (Fig. 5). This will be attributed to a lack of IL-12 production from cDC1 without IL-4 signaling (Fig. 6), although cDC1 cell numbers remain unaffected (Fig. 5e). Furthermore, Figure 4d shows no impact on PbTII (CD4+ T) cells even in an IL-4-deficient state. While Figure 7m provides a summarized image, the actual results suggest a more complex scenario. For instance, despite reduced cDC1-derived IL-12 levels in the absence of IL-4, PbTII cell priming remains unaffected. This imply that these PbTII cells differentiate into Tfh rather than Th1 cells. The Tfh PbTII cells might provide CD40L stimulation to cDC1.

Regarding with CD4+ T Cells (PbTII Cells):

What polarization of CD4+ T cells (Th1, Th2, Tfh) is responsible for producing CD40L in this situation? How does the presence or absence of IL-4 affect this polarization? IL-4 promotes Th2 and Tfh polarization, while IL-12 drives Th1 polarization. Figure 4d shows that $\gamma\delta$ T cells do not affect PbTII (CD4+ T) cells even in an IL-4-/- state, but was there any change in polarization?

Regarding with cDC1:

Figure 6 presents in vitro culture data and bulk RNA sequencing data for cDC1. While it is surprising that IL-4 and CD40 stimulation increase dendritic cell IL-12 production, the actual in vivo situation is more complex. As shown in Figure S4c, $\gamma\delta$ T cells produce not only IL-4 but predominantly IFN γ . This suggests that IFN γ stimulation should also be considered. For example, during blood-stage infection, $\gamma\delta$ T cells are known to activate cDC1 via IFN γ (Yarob et al., Front Immunol 2024). Regarding CD4+ T cells, it can be inferred that IFN γ -producing $\gamma\delta$ T cells enable cDC1 to produce IL-12. In my opinion, authors should do comparative analysis of IFN- γ -producing and IL-4-producing $\gamma\delta$ T cell fractions (single producers or double producers?).

Moreover, single-cell RNA sequencing of cDC1 during IL-4 inhibition combined with RAS vaccination. If IL-4+ $\gamma\delta$ T cells and IFN- γ + $\gamma\delta$ T cells are distinct populations, cDC1 may split into groups receiving different signals (IFN- γ receptor-positive vs. IL-4 receptor-positive populations). Even if most IL-4+ $\gamma\delta$ T cells also produce IFN- γ , there should still be IFN γ + IL-4- population of $\gamma\delta$ T cells, suggesting multiple cDC1 subsets. Single-cell RNA sequencing is necessary to identify these complex populations. Additionally, it is possible that the cDC1 activating CD4+ T cells differs from the one activating CD8+ T cells.

<the other comments>

All; CD4 T cells should be CD4+ T cells. CD8 T cells should be CD8+ T cells.

Line 78; Not only Ref 20 but also the other precise reference that showed “in human vaccination, $\gamma\delta$ T cells are associated with the success of this response” should be added.

Line 107; the first “ δ -/-” should be “TCR δ -/-” in somewhere.

Lines 122-125; Is this effect limited to irradiated parasites? How about genetically modified attenuated sporozoites (lisp2-, uis4-), or wild type parasites?

Line 133/Figure1 and FigureS1; I wonder why authors showed PbTII and PbTI cells/spleen in main Figure1 and PbTII and PbTI cells/liver in Supplementary Figure. Figure 1 title is “~liver TRM cell generation.”

Line 147/Figure 2; Even in Naïve mice, most splenic $\gamma\delta$ T cells expressed CD11a. CD11a is actually upregulated after RAS vaccination?? I understand that CD25 and CD69 are useful for activation marker for $\gamma\delta$ T cells. But Authors did not show the

data of upregulation of CD11a in $\gamma\delta$ T cells.

Lines 155-171; Why not use the same PbT-I cells and RAS for this experiment?

Line 174-175; Previous paper (Inoue et al., 2012 PNAS) showed that $\gamma\delta$ T cells provide CD40L for signalling CD40 on cDC1, a signal known to be essential in the response to RAS. Authors should mention or just refer the paper in the sentence. Related with this, I'm curious about CD40L expression in $\gamma\delta$ T cells and CD4+ T cells were occurred after RAS vaccination or not. I'm sure that Figure 3e implied that $\gamma\delta$ T cells might not express CD40L. But that is not direct evidence.

Lines 221-228, 367-369; Authors should show IL-4 expression from splenic $\gamma\delta$ T cells after vaccination clearly in mouse model.

Lines 240-241; Around 30% of CXCR5+CCR7+ cells were IFN γ +, similar to IL-4+. Authors should rephrase more accurately

Lines 280-289; Authors should show IL-12 on protein level and also show whether that is reflected on cDC1 in vivo in different conditions (TCRdKO, IL-4 blockade)

Line 347; I could not find the data "In this vaccination setting, IL-4 is produced by V γ 1+ $\gamma\delta$ T cells in the spleen"

Lines 449 and 455; sporozoites and infected RBCs were irradiated with 20K rads. Is this information described in Ref 61? If not, authors should add Ref.

Lines 477-480, Figure 4i; Negative enrichment was used to isolate $\gamma\delta$ T cells. Possibility of contamination with another subset. Is there any contamination with the other cell subsets. E.g. NK cells. Authors should show how match enriched by this method. Is there any problem if you use positive selection?? Did you forget to add Gr1 or is there any meaning without Gr1? (Gr1 was used in T cell enrichment)

Lines 465 and 479; F4/80 is not Mac1/CD11b. It should be separated from there.

Lines 561 and 563; "TCRd -/-" should be "TCR δ -/-".

Line 575; In vivo antibody depletion/blockade. Authors injected antibody via two ways (ip and iv). If the route of injection is important, provide the reasons for it.

Related with this, authors should add the information (ip or iv) in Figures/Figure Legends.

Figure 1 and Figure S1; Authors showed two time-points only. Thus, the time-points should be unified. "day23" in Figure 1 and "day35" in Figure S1. Those are different.

Figure 1 and Figure S1; Authors should clarify what the statistical test was used.

Figure S1; Authors should explain what is "GFP" in Fig S1b.

Figure S1, S2, 3, S5; proportions of cell populations should be added even in representative flow plots. The information is helpful to understand the data.

Figure 2 and S2; Author should explain what is TCR δ -GDL and why you used the mice.

Figure 2 and S2; I'm curious about impact on numbers of whole $\gamma\delta$ T cells and V γ 1+ $\gamma\delta$ T cells after RAS vaccination (not only CD11a+CD25+CD69+ gated cells).

Figure 2c, d; in Fig 2d, %CD11a+CD69+CD25+ $\gamma\delta$ T cells is 0.75% (V γ 1+) in average. This is not match with Fig 2c (9.89% in RAS). Why?

Figure 2f; Why is this the only figure "normalized to control" and not absolute numbers?

Figure 4f, g; Mouse group numbers are mixed up (4 and 6)

Figure 6; I wonder why B16 F1t2L(sc) was used in this experiment. There is no explanation.

Figure 6 a-e; CD40 should be α CD40

Figure 6b; Heat map should start with control (media) Same order as in 6a.

Figure 6d; Should mention both conditions under x axis (CD40 vs CD40+IL-4)

Figure 7a-f; How was (STAT3, 4, 6) signaling impacted?

Figure 7g; No details, Number of cells, CD44+?

Reviewer #3 (Remarks to the Author):

The study by Le et al is very nice and should be published in Nature Immunology. I think the case is made very strongly that IL-4 made by gamma/delta cells reactive to RAS contributes to CD8 responses both directly on T cells and through actions on cDC1. The figures are well-structured and easy to follow, have large enough sample sizes for and cover the various steps in the model. The study both confirms some previous facts, such as the role of gdT cells and IL-4 in the response as well as extending the work to propose a new model for mechanism. The topic of how the immune system responds to plasmodia is very important since there are really no effective vaccines against plasmodia (yet), so that work like this could help in that regard eventually.

The authors use an immunization model with radiation-attenuated Plasmodium sporozoites (RAS) and transfer of antigen-specific transgenic T-cells. They show that in this model, the Vg1+ gd T-cells and, in particular, their IL-4 production are crucial for the expansion of the transferred CD8+ T-cells. Evidence is given using CRISPR'd transferred cells that IL-4 is acts both directly CD8 T-cells and on cDC1 and stimulates DCs to induce IL-12 receptor and IL-12 production, with IL-4 synergizing with IL-12 to expand the T-cells. I particularly like the analysis in Fig. 5 and 6 showing the impact at a molecular level for the CD40 + IL-4 signaling in DCs.

My vote would be to publish this immediately without any more experiments. The case is made for the new model, and additional experiment could strengthen the conclusions (maybe) but I don't think its necessary. For example, a reviewer might ask for more high-tech specific evidence, such as CD8-specific Cre crossed to IL-4Ra floxed allele in mice. But I don't think that's necessary. A reviewer asking for that might just want to prevent this result from getting published any time soon. Or you could ask the authors to delve deeper into how IL-4 plus CD40 induces the changes (is it STAT6 or IRS pathway), asking if it is unique or whether other cytokines can or can't do it, but that's a different issue. Some people might criticize the model saying that it is artificial or that the authors don't only look at the CD8 T-cells as physiologic readout, and not the survival or something. I would challenge that point of view. Every model is artificial and CD8 expansion is likely to be highly relevant to the outcome.

In short, I support the publication of this study without new experiments. It's solid, high quality and important, and provides a new twist in DC biology. It's very interesting to think of this model in the evolutionary context. My colleague Osami Kanagawa always referred to IL-4 as the 'happiness factor', and this is pertinent here. The specificity is in the gdT cell reacting to RAS. That's the danger signal. Then the happiness factor is secreted to enable augmented responses both in the priming APC (inducing the IL-12) and the CD8 T cell directly. My guess is that the IL-4 levels are in the happiness range and not in the TH2-inducing high dose range of GATA-3 autoactivation. So it's interesting that the old innate style gdT cells side reserves that mechanism even now. Just musing.

Minor Comments

Figure 2d: what is the reference population for the % value (Total T-cells? A particular population of T-cells?)

Figure 3: The figure title is a bit misleading, as the figure doesn't really examine the CD40-CD40L signaling. The figure shows a negative result for a measurement (of T cells) in CD40L knockouts. This is an interpretation that CD40 signaling is intact. I would alter the words for more accuracy.

Figure 4: It is too bad that IL-4 is so hard to measure in the mouse Vg1+ cells. In 4 f/g/h, groups 4 and 6 seem to be mixed between the table and graphics. I would check this for accuracy.

Figure 5+6: The cells transferred at day -3 in Figure 5 are only strongly enriched in DC1 (60-80%, according to methods). The cells in Figure 6 were sorted, but even with sorting, there is no 100% purity. Is it possible that e.g. DC2 and not DC1 are the population responding to IL-4 treatment? I doubt it, but maybe the authors could add some text to discuss this. The BATF3 knockout result is the key here.

Regarding Fig 6 b and d, could the authors make a supplementary data item, for example a table. containing data on the most up/downregulated genes.

Figure S5: Do the Pb-T-1 cells also express the IL-4Ra on day 0 when, according to Fig 2, the ydT cells are required

Version 1:

Reviewer comments:

Reviewer #1

(Remarks to the Author)

The authors should be commended for their effort in answering my previous concerns. However, the data provided to address the KEY issue, IL-4 production by gd (in this specific case, Vg1+) T cells are not especially convincing. The percentages and numbers of IL-4-producing gd T cells are extremely low, including the impact of RAS vaccination (Fig. 5a-f). And the assessment of the relevance of gd T cells as IL-4 sources comes from the indirect comparison of different BM

chimera groups (5 and 6), which differ only by 2-fold in numbers (Fig. 5m). Therefore, the data come across as statistically significant, but biologically not particularly impressive. This diminishes the enthusiasm for the overall manuscript when considered for publication at the level of Nature Immunology, especially since its novelty is highly dependent on the gd/IL-4-mediated mechanism.

Reviewer #2

(Remarks to the Author)

The authors have made a sincere and substantial effort to address all my comments. The revised manuscript is significantly improved, both in experimental depth and in clarity of presentation. The additional data regarding $\gamma\delta$ T cell activation, CD4⁺ T cell phenotype (including the mixed Th1/Th2 subset), and IL-4/IFN γ co-production have strengthened the mechanistic model. Overall, most of my previous concerns have been appropriately resolved.

I now consider the manuscript suitable for publication in Nature Immunology after minor revision to correct a few remaining minor issues related to Figure presentation and the Legends.

Authors' response:

> Lines 122-125; Is this effect limited to irradiated parasites? How about genetically modified attenuated sporozoites (lisp2⁻, uis4⁻), or wild type parasites?

> To address this, we generated Pb1 Δ mei2 genetically attenuated parasites (GAP) (now described in methods lines 587-604) and immunised mice with this strain. Examination of the memory T cell response in the liver showed that the response to GAP was also gd T cell dependent. These data are shown in (Figure 1h-k) and are described in the results lines 124-133.

2nd Reviewer's comment:

The additional experiment using Pb1 Δ mei2 parasites is highly valuable.

However, since multiple experimental systems are presented in the manuscript, it is not immediately clear where Pb1 Δ mei2 was used.

Please indicate this information explicitly in the Figure 1h-k legend.

Authors' response:

> Figure 2c,d; in Fig 2d, %CD11a+CD69+CD25+ $\gamma\delta$ T cells is 0.75% (Vg1+) in average. This is not match with Fig2 c (9.89% in RAS). Why?

> We apologise to the reviewer as we should have applied the same gating strategy to the two plots. We have now provided a more detailed analysis of $\gamma\delta$ T cell activation and phenotype in the first three days of infection as provided in (Figure 2c-g and Supplementary Figure 2a-k). We feel that this new analysis answers the reviewer's question.

2nd Reviewer's comment:

• In Figure 2c, right representative flow plot in RAS, %CD69+CD25+ is 3.04. But I can't find "3.04%" in Figure 2d. Please explain it.

< The other minor points >

• In the newly added text, I noticed multiple instances where formatting requires superscript (e.g., "CD8⁺ T cells"). For example, the title of Fig. S1 and the first part of its legend contain "CD8+ T cells" that should be formatted with a superscripted "+". Although this is not a substantive scientific issue, this manuscript is intended for a high-quality journal and such typographical details should be corrected throughout the main text, all figure panels, and all figure legends. Please carefully check and fix all occurrences of superscripts (e.g., CD4⁺, CD8⁺) to ensure consistent, journal-appropriate formatting.

• What fluorochromes were used for CD14 and CD19 in Figure 6a?

In addition, please indicate in the figure that the lower flow plots represent the population gated on V δ 2⁺ cells, as this is shown explicitly in other figures.

• In Figure 5j-k, the label "IL-12" should be corrected to "Il12p40^{-/-}" to accurately reflect the experimental condition.

• Reference 56 should be correct. Please carefully check References and ensure that the citation corresponds correctly to the content referenced in the text.

• Authors should change $\alpha\gamma\delta$ to α -pan $\gamma\delta$ TCR in Figure 5 Legend to unify as the other parts.

• In Figure 6l, 7r, color of IFN γ is difficult to see. It's better to change to slightly darker or the other color.

• Figure 6f should add the label "IL-12a" to understand just by at a glance that the graph show comparison of IL-12a expression levels.

· In Figures, authors should unify the labels of y axis for showing cell number data, “# of” or “#” only.

Decision Letter:

Our ref: NI-A39470A

27th Oct 2025

Dear Dr. Beattie,

Thank you for submitting your revised manuscript "γδT cell-derived IL-4 initiates CD8 T cell immunity" (NI-A39470A). It has now been seen by the original referees and their comments are below.

The reviewers find that the paper has improved in revision, but as you can see reviewer 1 is not particularly impressed by the effect sizes re the IL-4 production by your γδ T cells. We think the comments of reviewer 1 are accurate but also that you have done as much as you can here, so overall we have decided that as long as you tone down related conclusions and discuss these limitations we are happy in principle to publish it in Nature Immunology. You will also need some textual revisions to address reviewer 2 comments.

However, first we will perform detailed checks on your paper and will send you a checklist detailing our editorial and formatting requirements in about a week. Please do not upload the final materials and make any revisions until you receive this additional information from us.

If you had not uploaded a Word file for the current version of the manuscript, we will need one before beginning the editing process; please email that to immunology@us.nature.com at your earliest convenience.

Thank you again for your interest in Nature Immunology Please do not hesitate to contact me if you have any questions.

Sincerely,

Nick Bernard, PhD
Senior Editor
Nature Immunology

Reviewer #1 (Remarks to the Author):

The authors should be commended for their effort in answering my previous concerns. However, the data provided to address the KEY issue, IL-4 production by gd (in this specific case, Vg1+) T cells are not especially convincing. The percentages and numbers of IL-4-producing gd T cells are extremely low, including the impact of RAS vaccination (Fig. 5a-f). And the assessment of the relevance of gd T cells as IL-4 sources comes from the indirect comparison of different BM chimera groups (5 and 6), which differ only by 2-fold in numbers (Fig. 5m). Therefore, the data come across as statistically significant, but biologically not particularly impressive. This diminishes the enthusiasm for the overall manuscript when considered for publication at the level of Nature Immunology, especially since its novelty is highly dependent on the gd/ IL-4-mediated mechanism.

Reviewer #2 (Remarks to the Author):

The authors have made a sincere and substantial effort to address all my comments. The revised manuscript is significantly improved, both in experimental depth and in clarity of presentation. The additional data regarding γδ T cell activation, CD4⁺ T cell phenotype (including the mixed Th1/Th2 subset), and IL-4/IFNγ co-production have strengthened the mechanistic model. Overall, most of my previous concerns have been appropriately resolved.

I now consider the manuscript suitable for publication in Nature Immunology after minor revision to correct a few remaining minor issues related to Figure presentation and the Legends.

Authors' response:

> Lines 122-125; Is this effect limited to irradiated parasites? How about genetically modified attenuated sporozoites (lisp2⁻, uis4⁻), or wild type parasites?

> To address this, we generated Pb1Δmei2 genetically attenuated parasites (GAP) (now described in methods lines 587-604) and immunised mice with this strain. Examination of the memory T cell response in the liver showed that the response to GAP was also gd T cell dependent. These data are shown in (Figure 1h-k) and are described in the results lines 124-133.

2nd Reviewer's comment:

The additional experiment using Pb1Δmei2 parasites is highly valuable.

However, since multiple experimental systems are presented in the manuscript, it is not immediately clear where Pb1Δmei2 was used.

Please indicate this information explicitly in the Figure 1h–k legend.

Authors' response:

> Figure 2c ,d; in Fig 2d, %CD11a+CD69+CD25+ γδ T cells is 0.75% (Vg1+) in average. This is not match with Fig2 c (9.89% in RAS). Why?

> We apologise to the reviewer as we should have applied the same gating strategy to the two plots. We have now provided a more detailed analysis of γδ T cell activation and phenotype in the first three days of infection as provided in (Figure 2c-g and Supplementary Figure 2a-k). We feel that this new analysis answers the reviewer's question.

2nd Reviewer's comment:

· In Figure2c, right representative flow plot in RAS, %CD69+CD25+ is 3.04. But I can't find "3.04%" in Figure 2d. Please explain it.

< The other minor points >

· In the newly added text, I noticed multiple instances where formatting requires superscript (e.g., "CD8⁺ T cells"). For example, the title of Fig. S1 and the first part of its legend contain "CD8+ T cells" that should be formatted with a superscripted "+". Although this is not a substantive scientific issue, this manuscript is intended for a high-quality journal and such typographical details should be corrected throughout the main text, all figure panels, and all figure legends. Please carefully check and fix all occurrences of superscripts (e.g., CD4⁺, CD8⁺) to ensure consistent, journal-appropriate formatting.

· What fluorochromes were used for CD14 and CD19 in Figure 6a?

In addition, please indicate in the figure that the lower flow plots represent the population gated on Vδ2⁺ cells, as this is shown explicitly in other figures.

· In Figure 5j–k, the label "IL-12" should be corrected to "Il12p40^{-/-}" to accurately reflect the experimental condition.

· Reference 56 should be correct. Please carefully check References and ensure that the citation corresponds correctly to the content referenced in the text.

· Authors should change α-γδ to α-pan γδ TCR in Figure 5 Legend to unify as the other parts.

· In Figure 6l, 7r, color of IFNγ is difficult to see. It's better to change to slightly darker or the other color.

· Figure 6f should add the label "IL-12a" to understand just by at a glance that the graph show comparison of IL-12a expression levels.

· In Figures, authors should unify the labels of y axis for showing cell number data, "# of" or "#" only.

Point by Point Response for ‘ $\gamma\delta$ T cell-derived IL-4 initiates CD8 T cell immunity’ (NI-
A39470)

We thank the Editors and Reviewers for their constructive comments. We have added
substantial new data that addresses the Reviewers’ concerns which is included in Figures 1 -
7 and Supplementary Figures 1-6. We have built upon our original submission and have
added compelling new data to show direct IL-4 production by $\gamma\delta$ T cells in the first days of
vaccination. We have shown that $\gamma\delta$ T cells do not need to express IL-4Ra to support CD8 T
cell accumulation and we have expanded our analysis beyond the spleen to include the liver
and liver draining lymph node. We examined the CD4 T cell response in more detail and
showed that in response to RAS vaccination, a proportion of *Plasmodium*-specific CD4 T
cells have a mixed Th1/Th2 phenotype with expression of both Tbet and GATA3, and
production of IFN γ and IL-4. Additionally, we showed that although IFN γ can enhance the
CD8 T cell response by acting with IL-4 to drive more IL-12 production by DC, $\gamma\delta$ T cells
are not the essential source of this IFN γ . We believe that these detailed mechanistic analyses
provide a new understanding of the events that lead to CD8 T cell priming in the context of
RAS vaccination. Our responses to all Reviewers’ concerns are detailed below and provided
in the text of the new manuscript in blue. We hope that our study will now be acceptable for
publication in *Nature Immunology*.

Reviewers' Comments:

Reviewer #1 (Remarks to the Author):

This is a relevant paper describing a new contribution of gamma-delta (gd) T cells to the
immune response against Plasmodium in mice. The experiments are in general novel
and interesting, although data analysis, presentation and description could be
significantly improved, as detailed below. The major drawback of the study is the lack of
evidence for (substantial) production of IL-4 by gd T cells, which would be essential to
support the proposed model linking to Plasmodium antigen-specific CD8+ T cells. This
is also critical to rule out that IL-4 is produced by other cells which could act upstream
of the gd T cell response.

General issues:

- It seems that no littermate controls were used throughout the study – can the authors
comment on this, given the importance of normalizing the microbiome between groups
to study immune responses to specific microorganisms?

**We agree with the reviewer that this is a potential confounder but believe that it is not
an issue in our study for the following reasons:**

The mice used in this study (including the WT B6 mice) are bred in the same SPF facility and
are housed in the same rooms before and throughout the experimental time frames and as

such, the microbiome is likely to be very similar between mouse strains. The B6 mice are
littermates.

All key findings were validated via multiple experimental approaches. For example, $\text{TCR}\delta^{-/}$
mice show the same defect in PbT-I expansion as B6 mice that have $\gamma\delta$ T cell function
blocked via administration of $\alpha\text{TCR}\delta$ antibody (GL3). The B6 mice in these experiments
were littermates. This demonstrates that differences in microbiome are not mediating the
effects of $\gamma\delta$ T cells.

Additionally, many experiments are conducted with BM chimeras or cell transfer into hosts
of the same genetic background ($\text{TCR}\delta^{-/}$, $\text{Batf3}^{-/}$).

- A more detailed reasoning should be provided for the models employed; examples:
the use of TCRd-GDL mice in Fig 2; or the use of OT-I cells in Fig 3 – namely, what was
the advantage compared to the previous PbT-I model?

**The basis for using these models is:**

$\text{TCR}\delta$ -GLD mice were used in **Figure 2** because expression of GFP by $\gamma\delta$ T cells in this
model can be used to gate on the cells in flow-cytometry experiments even if activation
causes down-regulation of the TCR. We apologise for not explaining this in the original
manuscript.

To alleviate any concerns about differences in this model, we have now conducted a full $\gamma\delta$ T
cells analysis in B6 mice by examining activation on days 1, 2 and 3 across the spleen, liver
and liver draining lymph node (dLN), now shown in (**Figure 2c-m** and **Supplementary**
**Figure 2a-k**). The original analysis in TCRd-GLD mice is now included in (**Supplementary**
**Figure 2l and m**). These data are discussed in the text on lines 172-176.

OT-I cells were used in **Figure 3** because adoptive transfer leads to spontaneous up-
regulation of CD69 on PbT-I T cells for reasons yet to be determined. We therefore used OT-I
T cells, which do not exhibit spontaneous CD69 upregulation upon transfer, allowing for
sensitive detection of antigen presentation to these cells within the first 24 hours of RAS
vaccination. We have provided an explanation of this in the text (lines 188-191). OT-I T cell
accumulation at day 6 was defective in $\text{TCR}\delta^{-/}$ mice mirroring the response of PbT-I T cells
(**Supplementary Figure 3a-c**). This (in addition to the endogenous response data in **Figure**
**1**) has the secondary effect of demonstrating that the defect in the response in $\text{TCR}\delta^{-/}$ mice is
not restricted to one TCR-specificity.

- The authors should also discuss the physiological relevance and the limitations of
their system, especially regarding the dynamics of the observed response in the spleen
in the first 24h (versus the natural course of the disease that includes a 48h liver stage).
In fact, what is happening at early time points in the liver with the resident gd T cells?

As discussed above, we have added data to (**Figure 2 (c-m)** and **Supplementary Figure 2a-**
**k** showing the activation phenotype of $\gamma\delta$ T cells across the spleen, liver and liver draining

LN in the first three days after RAS injection. These data show a small but significant
increase in the activation of $V\gamma 1^+$ $\gamma\delta$ T cells in the liver and liver draining LN after RAS
injection.

We also clarified our model with the following line in the discussion (lines 492 to 494): ‘As
we used a model of intravenous vaccination with RAS, activation of $\gamma\delta$ T cells was observed
in the spleen, the liver and the liver dLN, consistent with the antigen distribution in this
model.’

We would like to highlight that we have used IV injection of RAS as a model to study the
response. We are not trying to extrapolate our data to the kinetics of natural malaria infection.

- The observation that IL-4 signaling in PbT-1 cells results in IL-12R induction in vivo
should be strengthened experimentally to support the working model. Ideally, in Fig 7,
authors should assess the expression of IL-12R in PbT-1 cells transferred into TCRd KO
recipients reconstituted with either WT or IL-4 KO gd T cells (as in Figure 4, panel k).

We did not conduct an experiment in the way suggested by the reviewer but have provided
IL-12R and IL-4Ra expression on in vivo derived PbT-I T cells 3 days after RAS vaccination
in B6, *Il4*^{-/-} and *Tcr δ* ^{-/-} recipients in (**Figure 7j, k and l**). We feel that this demonstrates that
the up-regulation of these surface molecules on CD8 T cells relies both on IL-4 and on the
presence of $\gamma\delta$ T cells in this model.

SPECIFIC COMMENTS ON THE TEXT:

Lines 99-101: The T cell response to intravenously injected RAS is initiated largely in the
spleen with priming and expansion occurring in this site, followed by recirculation and
resultant rapid accumulation of activated T cells in the liver¹⁸. In the mentioned
referenced, the following is stated: “the spleen is the main site for priming sporozoite
specific T cells after intravenous administration of parasites, but they do not formally
exclude the liver draining lymph nodes or the liver as important sites of activation for
protective immunity”. Indeed, since a role for gd T cells in the liver stage has been
proposed following attenuated PbA sporozoite infection (Chora et al. Immunity 2023), it
would be interesting to address the contribution of liver versus spleen gd T cells to the
accumulation of PbT-I cells in the liver/spleen. Given that the PbT-1 accumulation
phenotype does not happen upon infection with iRBCs, and gd T cells are essential in
the first 24h for later PbT-I cell accumulation in the spleen, the role of gd T cells in liver
stage should be addressed. It would be interesting to assess the accumulation of PbT-I
cells in the liver of WT or TCRd KO splenectomized mice, for example. This becomes
even more apparent at Fig 2, where the authors conclude that splenic $V\gamma 1^+$ gd T cells
initiate immunity to RAS (given their increased frequencies of CD69, CD25 and CD11a
after 24h of RAS infection), without analysing the liver compartment. If all the
phenotype is due to gd cells in the spleen in the first 24h, this may question the

physiological relevance of the model, given that “efficient priming via this route [spleen]
most likely derives from the large load of irradiated sporozoites deposited in the spleen
after intravenous administration and the high frequency of T cells found in this organ”
[from Lei Shong Lau et al. PLOS Pathogens 2014, where PbT-I transgenic line generation
was described]?

Apologies to the reviewer, but we are not trying to imply that the CD8 T cell response to
intravenous RAS vaccination is only initiated in the spleen. We have now included data to
show that the CD8 T cell response generated in the liver draining LN is similarly $\gamma\delta$ T cell
dependent (**Figure 1b and Supplementary Figure 3b**).

As discussed above, we have also included data showing $\gamma\delta$ T cell activation in the first 3
137 days after RAS vaccination (**Figure 2 c-m and Supplementary Figure 2a-k**).

Thus, while we used the model of intravenous RAS inoculation for simplicity, and to
generate the largest possible magnitude of response for dissection of mechanism, it seems,
wherever the response is initiated, it depends on $\gamma\delta$ T cells. Reviewer proposed experiments
using splenectomised mice therefore would not have yielded valuable information on this
point because CD8 T cell priming in liver dLN’s would still show $\gamma\delta$ T cell dependence. The
splenectomy experiments were therefore not conducted.

Lines 216-219 – there is a gd contribution to PbT-I cell accumulation (Fig 4h) that is not
IL-4 dependent (comparison between groups 5 and 6), which should be discussed;

The reviewer is correct. We have added text on this point in lines 309-312.

We added:

‘There was a small, but significant increase in PbT-I accumulation when $\delta^{-/-}\rightarrow B6$ (Group 4)
chimeras were compared with $IL4^{-/-}+\delta^{-/-}\rightarrow B6$ (group 6) chimeras, suggesting an additional
small contribution of $\gamma\delta$ T cells that was not IL-4 dependent, but the nature of this
contribution is yet to be determined.’

additionally, IL-4 has also been shown to drive the expansion of some gd T cell subsets;
how does this data exclude that IL-4 is acting upstream of gd T cell activation? This
could be tested (related to Fig 4k) by reconstituting TCRd KO mice with gd T cells from
IL4Ra KO.

We have now provided these data in (**Figure 5n-p**) with description in the text (lines 324-
326). These data show that $IL-4R^{-/-}$ $\gamma\delta$ T cells rescue the accumulation of PbT-I T cells in
$TCR\delta^{-/-}$ mice to the same level achieved by WT $\gamma\delta$ T cells. IL-4 therefore does not act
upstream of $\gamma\delta$ T cells to initiate immunity in this system.

Line 241 – it is mentioned that IFN-g, unlike IL-4, is not largely produced by gd T cells
expressing CXCR5 and/or CCR7, but the data shows that those subsets produce both
cytokines at similar frequencies. It is true that CXCR5- and/or CCR7- have increased
frequencies of IFN-g production compared to their positive counterparts, but the way it

is written is not accurate. Additionally, in lines 366-367 of discussion, it is stated that
Vg9+Vd2+ gd T cells produced IL-4, but not IFNg or TNF following restimulation, which
is not at all supported by the data (Fig 4m).

We apologise for these oversights and have corrected the interpretation of the results (lines
341-343) and changed the wording of the discussion of these data (lines 483-486).

Lines 291-295 – IL-12 blockade is not specific to cDC1; also, it would be important to
assess IL-12 production by cDC1 in TCRd KO mice to support the proposed model.

Because RAS is a weak antigenic stimulus, we have not successfully detected cDC1
activation in our experiments, even though all of our functional analysis shows that these
cells are crucial to the response. Our attempts to perform the experiment suggested by the
reviewer to detect IL-12 production from cDC1 *in vivo* were unsuccessful.

We have however addressed this concern by performing DC reconstitution experiments
whereby cDC1 from *Il12p40^{-/-}* mice were used to reconstitute the cDC1 pool in *Batf3^{-/-}*
recipients. In these experiments, PbT-I T cells numbers were lower when cDC1 were IL-12
deficient, with a similar reduction in the response to that seen in mice that received IL-12
blocking antibody. These data are now shown in (**Figure 6i-k**) with the results described in
(lines 400-404). We believe that this addresses the reviewers concern and validates that cDC1
are the important source of bioactive IL-12 in this system.

SPECIFIC COMMENTS ON THE FIGURES:

Figure 1, panels e and f – Y scales should be adjusted (potentially in Log scale?) to
facilitate reading. Same should be done for Supp. Figure 1, panels g and h. Also in Supp.

The scales have been adjusted as much as possible for readability. We do not think that log
scales would be suitable for these data.

Figure 1, panel g it is not clear if the statistics presented refers to all CD8 T cell memory
subsets – Tcm differences look they might be significant?

We apologise and have now provided the statistics for the comparisons in different colours.

Figure 1, panel f – for consistency, the authors should show the absolute number of
PbT-1 cells in the spleens of anti-Vg1 and anti-GD antibodies, and not the values
normalized to control spleens.

We are now showing absolute numbers in (**Figure 2o**).

Fig 1e,f and Supp. Fig 1g,h – A more detailed gating strategy for memory T cells should
be provided.

The gating strategy has been added.

Supp Fig 1b is not referenced in the manuscript.

We apologise for the oversight, A reference to this figure has been added.

Supp. Figure 2, panel a - for consistency, the frequency of CD69+ CD25+ cells within
Vg1- gd T cells should be added to the plots.

We have provided substantial new data in this figure (**Figure 2c-m** and **Supplementary**
**Figure 2a-m**), including the gating requested.

Fig 2c – FACS plots are not representative of frequencies in Fig2d; if data in Fig2d are
normalized, it should be stated in the plot or caption.

There was an error in the way that the gating was labelled for these plots which is now
corrected, with new plots added in (**Figure 2c**). The frequencies have also been added to all
FACS plots.

Figure 3, panel a, the PbT-1 proliferation fails to distinguish discrete populations
undergoing 1, 2 etc divisions. Why is this? Also, the frequency of proliferating OT-1 cells
transferred to WT or TCRd KO mice should be presented and potentially have a
separate graph with the mean values for the two different conditions, to assess if the
cells transferred into WT recipients have increased proliferation compared to those
transferred into TCRd KO mice. Fig 3a-c – are these CD44+ OT-I cells? If so, is the
overall frequency of OT-I or CD44 expression similar between B6 and TCRd KO?

The response to RAS is variable (as reflected by the error bars of our data). We apologise that
we did not select a FACS plot for the day 2 time point that was more representative of the
mean of the response across the replicate mice in our original submission. We have now
added plots that show the absolute number of OT-I cells (**Figure 3e**), the proportion of
CD69+ OT-I cells (**Figure 3b**) and the proportion of proliferating OT-I T cells (**Figure 3c**), in
addition to other data including the number of CD44+ OT-I cells (**Figure 3g**). These data
show no defect in the early response of OT-I (up to day 3) across the spleen, liver or liver
dLN.

Fig 4d – the fact that PbT-II CD4 T cells are completely dependent on gd T cells but not
IL-4 is not discussed, and it is a missing link in the final working model of Fig 7m – an
extra arrow with a '?' should be added?

We agree and have added an arrow to the final working model in Figure 7r as suggested by
the reviewer.

In the original version of our manuscript, we had the following statement in the discussion:
'Here, we have shown that the initiation of the CD4 response to liver stage parasites is also
dependent on a factor provided by $\gamma\delta$ T cells but have not yet identified the nature of this
factor.'

Fig 4f-h – Group 4 and 6 of the table of fig 4f are switched in figs 4g,h; additionally, it
would be interesting to see the reconstitution of the gd liver compartment as shown for
spleen in fig 4g.

We apologise for the formatting error in the table, this has been fixed. We have now added
the data for $\gamma\delta$ T cell reconstitution in the liver which can be seen in (**Supplementary Figure**
**5i**) and detailed in the results (referenced in line 304).

Figure 4, panel h - in the BM Chimera experiments, to claim gd T cells are the source of
IL-4 based on such comparisons, the recipient mice should be IL-4 KO mice instead of
WT B6 mice.

We could not find a rationale for doing the experiment in this way, and the reviewer does not
provide one. In figure 4h of the original submission (now **Figure 5m**), there is a control
chimera group that is B6 \rightarrow IL-4KO recipient. The PbT-I accumulation in this group was not
different from the chimeras that were B6 into B6 recipients. This shows that the important IL-
4 producing cells are BM-derived. As such, we see no rationale for the experiment proposed.

Figure 4, panel j and k, the legend on the third bar should be “no transfer” as is in the
legend?

This has been modified.

Fig 4k – it would be good to show absolute numbers, given that the transfer of gd from
IL-4 KO mice still supports half the accumulation of PbT-I cells of transfer with gd cells
from B6 mice. Is that half comparable to PbT-I cell numbers in TCRd KO mice?

These are technically very challenging experiments that rely on cell enrichments and transfers
from large numbers of donor mice and then dissection of live sporozoites from mosquito
salivary glands. When the level of infection of the salivary glands varies, this causes
variability in our T cell response data. In most cases we manage to control for this, but for
these transfer experiments, the two experiments showed a large variability in the magnitude
of the response. We have now performed additional experiments and have included $\gamma\delta$ T cell
transfers from *Il4ra*^{-/-} mice as requested by the reviewer with all data shown as absolute
numbers of PbT-I T cells/spleen. These data are now shown in (**Figure 5n-p**). The
reconstitution of the PbT-I cell response by *Il4*^{-/-} $\gamma\delta$ T cells is negligible when compared to
the response observed in the ‘no transfer’ group.

Fig S4d – all p-values presented have the same value; is this so or a mistake?

We thank the reviewer for noticing this error and have now corrected it, and the other
statistical analyses associated with these data in the new version of (**Figure 5 and**
**Supplementary Figure 6**).

Fig S5b – these data on XCR1 expression are not mentioned in the manuscript?

A reference to these data as a gating strategy now (Supplementary Figure 6d) is provided
on line 370.

Fig 6h is referenced in the manuscript, but it is not shown?

This was a reference to the model that was shown in Figure 6h (now Figure 6l).

Reviewer #2 (Remarks to the Author):

<Reviewer comments to Authors>

Previous studies have suggested that the establishment of malaria immunity induced
by sporozoite vaccines is positively correlated with the degree of $\gamma\delta$ T cell expansion
(Rer40: Seder et al., 2013 Science). However, how $\gamma\delta$ T cells contribute to the
establishment of malaria immunity induced by sporozoite vaccines has remained
unclear.

This manuscript showed the importance of $\gamma\delta$ T cells in the expansion of Plasmodium-
specific CD8+ T cells in response to RAS immunization and the following liver
accumulation to develop into resident memory T cells. Authors tried to prove that $\gamma\delta$ T
cell-derived IL-4 was required both to affect IL-12 production in cDC1 and directly
affect priming of CD8+ T cells alongside IL-12. Moreover, they showed that $\gamma\delta$ T cell-
derived human malaria patients produce IL-4. The experiment is well-conducted, and I
was intrigued by this manuscript. However, Zavala groups has already reported that IL-4
is required for protective immunity against liver-stage Plasmodium infection (Ref 16:
Carvalho et al., 2002 Nat Med). They also showed that IL-4 signaling in CD8+ T cells is
required for the protective immunity (Ref 38: Morrot et al., 2005 JEM). Furthermore, it is
known that $\gamma\delta$ T cells can produce IL-4 in both mice and humans (Ref 24: Gerber et al.,
1999 J Immunol). At present, it seems that this paper lacks sufficient novelty and
impact to be published in Nature Immunology. However, this paper is valuable as it
elucidates an aspect of the role of $\gamma\delta$ T cells in RAS vaccination-induced immunity
against liver stage malaria, which has been ambiguous until now. I believe that this
manuscript will be improved by additional experiments and more detailed analyses as
pointed out below.

There are two aspects to novelty in our report.

1. Novelty related to the responses to malaria. While some components of the model have
been elucidated prior to this report (all acknowledged in the introduction and/or discussion),
how these components fit together was unclear and the key role for IL-4 in IL-12 induction

and their combined effect on CD8 T cell responsiveness, completely unknown. Our report
links these and new components (IL-12) together to provide **novel mechanistic**
**understanding of how CD8 T cell responses can be induced to the weak antigenic**
**stimuli provided by RAS through initial detection by $\gamma\delta$ T cells.**

**2. Novelty in the bigger picture.** While Zavala's group may have elucidated some steps in
this response, the Immunology community has essentially ignored IL-4 as a key cytokine
involved in CD8 T cell immunity. IL-4 has recently been linked to better outcomes in CART
cell therapy (2 Nature papers), however, with no mention of the Zavala data raised by these
reports. **Our data and the elucidated model provide novel understanding of how IL-4**
**can contribute to CD8 T cell immunity**, which the immunological community can now
extrapolate to other relevant situations. Secondly, **it provides novel mechanistic**
**understanding of how innate like cells (in this case $\gamma\delta$ T cells) can act as catalysts for**
**initiation of adaptive immunity.** *These two broader novelties are key to the importance of*
*this report.*

The authors demonstrate that the absence of an IL-4 signal to cDC1 reduces PbTII
(CD8+ T) cell priming efficiency (Fig. 5). This will be attributed to a lack of IL-12
production from cDC1 without IL-4 signaling (Fig. 6), although cDC1 cell numbers
remain unaffected (Fig. 5e). Furthermore, Figure 4d shows no impact on PbTII (CD4+ T)
cells even in an IL-4-deficient state.

While Figure 7m provides a summarized image, the actual results suggest a more
complex scenario. For instance, despite reduced cDC1-derived IL-12 levels in the
absence of IL-4, PbTII cell priming remains unaffected. This imply that these PbTII cells
differentiate into Tfh rather than Th1 cells. The Tfh PbTII cells might provide CD40L
stimulation to cDC1.

Regarding with CD4+ T Cells (PbTII Cells):

What polarization of CD4+ T cells (Th1, Th2, Tfh) is responsible for producing CD40L in
this situation? How does the presence or absence of IL-4 affect this polarization? IL-4
promotes Th2 and Tfh polarization, while IL-12 drives Th1 polarization. Figure 4d shows
that $\gamma\delta$ T cells do not affect PbTII (CD4+ T) cells even in an IL-4^{-/-} state, but was there
any change in polarization?

We thank the reviewer for thinking about our data in this considered way and agree with
them that this response represents a very complex scenario, that was not completely
represented by the model shown in the original version of the manuscript. In response to the
reviewers' comments, we have assessed the phenotype of the CD4 T cells that are generated
by RAS vaccination in B6, $\delta^{-/-}$ and $Il4^{-/-}$ mice. These data are now shown in (**Figure 4p-s and**
**Supplementary Figure 4i and j**) and discussed in lines 270-284 of the results.

We didn't find any evidence that the frequency of CD4 PbT-II T cells that differentiated into
Tfh was different between B6, $Il4^{-/-}$ or $Tcrd^{-/-}$ hosts. We did however observe that RAS

vaccination induced a population of ‘mixed phenotype’ CD4 T cells that expressed both Tbet
and Gata3 and produced IFN γ and IL-4. These mixed phenotype cells were absent in *Il4*^{-/-} and
TCR δ ^{-/-} mice, suggesting that IL-4 from $\gamma\delta$ T cells induced IL-4 production by CD4 T cells in
this model.

We were unsuccessful in our attempts to identify any difference in the expression of CD40L
between the T cells in these experiments and as such have not included these data. We have
found it difficult to find evidence in the published literature that this is a reliable technique
that can be used *ex vivo* on activated CD4 T cells, likely due to the transient nature of CD40L
expression.

Regarding with cDC1:

Figure 6 presents *in vitro* culture data and bulk RNA sequencing data for cDC1. While it
is surprising that IL-4 and CD40 stimulation increase dendritic cell IL-12 production, the
actual *in vivo* situation is more complex. As shown in Figure S4c, $\gamma\delta$ T cells produce not
only IL-4 but predominantly IFN γ . This suggests that IFN γ stimulation should also be
considered. For example, during blood-stage infection, $\gamma\delta$ T cells are known to activate
cDC1 via IFN γ (Yarob et al., Front Immunol 2024). Regarding CD4⁺ T cells, it can be
inferred that IFN γ -producing $\gamma\delta$ T cells enable cDC1 to produce IL-12. In my opinion,
authors should do comparative analysis of IFN- γ -producing and IL-4-producing $\gamma\delta$ T cell
fractions (single producers or double producers?).

We appreciate the reviewers’ points and have now added these data in (**Supplementary**
**Figure 4a-f**). The reviewer was correct that a large proportion of the V γ 1⁺ $\gamma\delta$ T cells do
produce IFN γ in response to RAS. We do not see segregation between IFN γ producers or IL-
4 producers with a small proportion of the cells producing both cytokines (as has previously
been reported in the literature). These data are now discussed in lines 228-232 of the results
section.

Given these data, we examined the role of IFN γ by blocking IFN γ after RAS vaccination and
examining PbT-I accumulation in the spleen 6 days later, or via transfer of PbT-I and PbT-II
T cells into *Ifng*^{-/-} hosts. These data showed that while IFN γ does contribute to the
accumulation of PbT-I T cells in response to RAS vaccination, the effect is not as profound
as the effect of IL-4. We have also provided additional data derived from mixed bone marrow
chimeras to show that $\gamma\delta$ T cells are not an essential source of IFN γ in this model (**Figure 4d-**
**h**). Please also consider the answer to the below question.

Moreover, single-cell RNA sequencing of cDC1 during IL-4 inhibition combined with
RAS vaccination. If IL-4+ $\gamma\delta$ T cells and IFN- γ + $\gamma\delta$ T cells are distinct populations, cDC1
may split into groups receiving different signals (IFN- γ receptor-positive vs. IL-4
receptor-positive populations). Even if most IL-4+ $\gamma\delta$ T cells also produce IFN- γ , there

should still be IFN γ + IL-4- population of $\gamma\delta$ T cells, suggesting multiple cDC1 subsets.
Single-cell RNA sequencing is necessary to identify these complex populations.
Additionally, it is possible that the cDC1 activating CD4+ T cells differs from the one
activating CD8+ T cells.

We thank the reviewer for making this suggestion as we believe they have strengthened our
analysis and conclusions. We did not find any experimental evidence for IFN γ and IL-4 being
derived from different $\gamma\delta$ T cell subsets in response to RAS vaccination (see above). As such,
we integrated new IFN γ samples into our RNA sequencing analysis (**Figure 6**), to consider
the combined effects of IFN γ and IL-4 in our datasets.

These analyses showed that IFN γ had an enhancing effect on CD40 and IL-4 signals in cDC1
with important genes such as Il12a being further increased in transcript by the addition of
IFN γ . These data are now shown in Figure 6 and described in the results on lines 382-395.

All; CD4 T cells should be CD4+ T cells. CD8 T cells should be CD8+ T cells.

This has been modified throughout.

Line 78; Not only Ref 20 but also the other precise reference that showed “in human
vaccination, $\gamma\delta$ T cells are associated with the success of this response” should be
added.

Additional references #21-#24 have been added.

Line 107; the first “ δ -/-” should be “TCR δ -/-“ in somewhere.

Addressed on line 108 of the results section.

Lines 122-125; Is this effect limited to irradiated parasites? How about genetically
modified attenuated sporozoites (lisp2-, uis4-), or wild type parasites?

To address this, we generated Pb \$\Delta\$ mei2 genetically attenuated parasites (GAP) (now
described in methods lines 587-604) and immunised mice with this strain. Examination of the
memory T cell response in the liver showed that the response to GAP was also \$\gamma\delta\$ T cell
dependent. These data are shown in (**Figure 1h-k**) and are described in the results lines 124-
133.

Line 133/Figure1 and FigureS1; I wonder why authors showed PbTII and PbTI
cells/spleen in main Figure1 and PbTII and PbTI cells/liver in Supplementary Figure.
Figure 1 title is “~liver TRM cell generation.”

Our aim was to highlight the data that conveyed the impact of \$\gamma\delta\$ T cells on the early
accumulation of T cells in the spleen, and the secondary effect on liver Trm cell generation.

We have renamed the figure to: ‘Figure 1 - $\gamma\delta$ T cells are essential for the CD8 T cell
response to RAS’ which we feel more accurately reflects the data shown.

Line 147/Figure 2; Even in Naïve mice, most splenic $\gamma\delta$ T cells expressed CD11a. CD11a
is actually upregulated after RAS vaccination?? I understand that CD25 and CD69 are
useful for activation marker for $\gamma\delta$ T cells. But Authors did not show the data of
upregulation of CD11a in $\gamma\delta$ T cells.

We have removed all reference to CD11a in our $\gamma\delta$ T cell activation analysis and gate only on
CD25 and CD69 to identify activated $\gamma\delta$ T cells in (Figure 2c-m and Supplementary
Figure 2a-m).

Lines 155-171; Why not use the same PbT-I cells and RAS for this experiment?
Please see our response to reviewer 1 (lines 68-76 above). We have also provided an
explanation of this in the text of the manuscript (lines 188-191).

Line 174-175; Previous paper (Inoue et al., 2012 PNAS) showed that $\gamma\delta$ T cells provide
CD40L for signalling CD40 on cDC1, a signal known to be essential in the response to
RAS. Authors should mention or just refer the paper in the sentence. Related with this,
I’m curious about CD40L expression in $\gamma\delta$ T cells and CD4+ T cells were occurred after
RAS vaccination or not. I’m sure that Figure 3e implied that $\gamma\delta$ T cells might not express
CD40L. But that is not direct evidence.

We have now included the appropriate reference to Inoue et al 2012 PNAS. The reviewer is
correct that we have not provided direct evidence that $\gamma\delta$ T cells do not provide CD40L
signals in this system. The data do however show that $\gamma\delta$ T cells are not the essential source of
this signal when CD4 T cells are able to provide it. We have altered the wording of the text in
the results to reflect this (Line 219) and modified the title of Figure 3 to ‘Figure 3 - Antigen
presentation is intact in the absence of $\gamma\delta$ T cells, which are not an essential source of
CD40L.’

Lines 221-228, 367-369; Authors should show IL-4 expression from splenic $\gamma\delta$ T cells
after vaccination clearly in mouse model.

We have now provided these data in (Figure 5b-i and Supplementary Figure 5a-h) as
described in lines 287-300.

Lines 240-241; Around 30% of CXCR5+CCR7+ cells were IFN γ +, similar to IL-4+.
Authors should rephrase more accurately

We apologise and have reworded this description (lines 341-343).

Lines 280-289; Authors should show IL-12 on protein level and also show whether that
is reflected on cDC1 in vivo in different conditions (TCRdKO, IL-4 blockade)

Please see our answer to reviewer 1 on this issue (Lines 179-189 above).

Line 347; I could not find the data “In this vaccination setting, IL-4 is produced by V γ 1+
$\gamma\delta$ T cells in the spleen”

We apologise and have now added these data (Figure 5b-i and Supplementary Figure 5a-h)
as described in lines 287-300.

Lines 449 and 455; sporozoites and infected RBCs were irradiated with 20K rads. Is this
information described in Ref 61? If not, authors should add Ref.

References now provided.

Lines 477-480, Figure 4i; Negative enrichment was used to isolate $\gamma\delta$ T cells. Possibility
of contamination with another subset. Is there any contamination with the other cell
subsets. E.g. NK cells. Authors should show how match enriched by this method. Is
there any problem if you use positive selection?? Did you forget to add Gr1 or is there
any meaning without Gr1? (Gr1 was used in T cell enrichment)

We elected to use negative enrichment as also used by others in the field (eg PMID
38802512) to avoid any confounding factors that could be induced by positive selection with
the antibody against the delta TCR (GL3). We did not add GR1 antibody as a population of
\$\gamma\delta\$ T cells are known to express GR1 (PMID 38816652) and we did not want to lose this
population from our experiments. The enrichment method did not result in a pure population
of \$\gamma\delta\$ T cells but in all cases where this method was used, the level of enrichment was
equivalent between the WT control transfer and the genetic knockout (IL-4, IL-4Ra etc...)
and transfer of the \$\gamma\delta\$ T cells was into TCR \$\delta^{-/-}\$ mice which lack \$\gamma\delta\$ T cells but are replete for
all other cell types (including the cells that may have been contaminating our enriched \$\gamma\delta\$ T
cell preparations).

Lines 465 and 479; F4/80 is not Mac1/CD11b. It should be separated from there.

Rectified.

Lines 561 and 563; “TCRd -/-” should be “TCR $\delta^{-/-}$ ”.

Rectified.

Line 575; In vivo antibody depletion/blockade. Authors injected antibody via two ways
(ip and iv). If the route of injection is important, provide the reasons for it.

Related with this, authors should add the information (ip or iv) in Figures/Figure
Legends.

We have added the route of injection to the figure legends and the methods section with
references. In all cases, we used the dose and route of injection identified from published
papers.

Figure1 and FigureS1; Authors showed two time-points only. Thus, the time-points
should be unified. “day23” in Figure1 and “day35” in FigureS1. Those are different.
We are unclear on what the reviewer is specifying here. In all cases, we have stated the day
post-RAS vaccination for our analysis in the figure legends. In the case of later time point
analysis (greater than day 21) we consider these to be ‘memory’ time points when the peak of
the primary response has waned, and the memory populations have stabilised. As we have
specified the days at which the analysis was performed, we do not feel that repeating the
experiments to have unified time points is justified in this situation.

Figure1 and FigureS1; Authors should clarify what the statistical test was used.
In all figures, the statistical tests used are listed in the figure legends.

Figure S1; Authors should explain what is “GFP” in Fig S1b.
Now defined in the Methods section.

Figure S1, S2, 3, S5; proportions of cell populations should be added even in
representative flow plots. The information is helpful to understand the data.
The proportions have been added.

Figure 2 and S2; Author should explain what is TCR δ -GDL and why you used the mice.
Please see our response to reviewer 1 on this issue (Reviewer 1, lines 58-66 above).

Figure 2 and S2; I’m curious about impact on numbers of whole $\gamma\delta$ T cells and V γ 1+ $\gamma\delta$ T
cells after RAS vaccination (not only CD11a+CD25+CD69+ gated cells).
We have added these data in our analysis of $\gamma\delta$ T cell activation in (Figure 2h-m) and
described in lines 166-172.

Figure 2c ,d; in Fig 2d, %CD11a+CD69+CD25+ $\gamma\delta$ T cells is 0.75% (V γ 1+) in average. This
is not match with Fig2 c (9.89% in RAS). Why?

We apologise to the reviewer as we should have applied the same gating strategy to the two
plots. We have now provided a more detailed analysis of $\gamma\delta$ T cell activation and phenotype

in the first three days of infection as provided in (**Figure 2c-g and Supplementary Figure**
**2a-k**). We feel that this new analysis answers the reviewer's question.

Figure 2f; Why is this the only figure "normalized to control" and not absolute numbers?
We are now showing these data as absolute numbers in (**Figure 2o**).

Figure 4f, g; Mouse group numbers are mixed up (4 and 6)

Sincere apologies. This has been rectified.

Figure 6; I wonder why B16 Flt2L(sc) was used in this experiment. There is no
explanation.
B16.Flt3

B16.Flt3L subcutaneous injection is a published method for the *in vivo* expansion of murine
dendritic cells (PMID23329497). This reference and explanation has been added to the
methods Lines 631-632).

Figure6 a-e; CD40 should be αCD40
We have modified the figure.

Figure 6b; Heat map should start with control (media) Same order as in 6a.

The heat map has been modified to include new data. The order is now media first.

Figure 6d; Should mention both conditions under x axis (CD40 vs CD40+IL-4)
We have provided new volcano plots to encompass the additional condition included in the
new analysis. These plots are labelled as requested.

Figure 7a-f; How was (STAT3, 4, 6) signaling impacted?
We did not examine these molecules in our initial analysis of these data and as we did not
need to repeat these experiments to satisfy any concerns raised by multiple reviewer's we
have not examined these pathways. We feel that this is somewhat outside of the scope of the
current manuscript and hope that the reviewer would allow us to investigate these pathways
in our follow up studies.

Figure 7g; No details, Number of cells, CD44+?
This is fixed in the new (**Figure 7g**).

Reviewer #3 (Remarks to the Author):

The study by Le et al is very nice and should be published in Nature Immunology. I think
the case is made very strongly that IL-4 made by gamma/delta cells reactive to RAS
contributes to CD8 responses both directly on T cells and through actions on cDC1. The
figures are well-structured and easy to follow, have large enough sample sizes for and
cover the various steps in the model. The study both confirms some previous facts,
such as the role of gdT cells and IL-4 in the response as well as extending the work to
propose a new model for mechanism. The topic of how the immune system responds to
plasmodia is very important since there are really no effective vaccines against
plasmodia (yet), so that work like this could help in that regard eventually.

The authors use an immunization model with radiation-attenuated Plasmodium
sporozoites (RAS) and transfer of antigen-specific transgenic T-cells. They show that in
this model, the Vg1+ gd T-cells and, in particular, their IL-4 production are crucial for
the expansion of the transferred CD8+ T-cells. Evidence is given using CRISPR'd
transferred cells that IL-4 acts both directly on CD8 T-cells and on cDC1 and stimulates
DCs to induce IL-12 receptor and IL-12 production, with IL-4 synergizing with IL-12 to
expand the T-cells. I particularly like the analysis in Fig. 5 and 6 showing the impact at a
molecular level for the CD40 + IL-4 signaling in DCs.

My vote would be to publish this immediately without any more experiments. The case
is made for the new model, and additional experiment could strengthen the
conclusions (maybe) but I don't think it's necessary. For example, a reviewer might ask
for more high-tech specific evidence, such as CD8-specific Cre crossed to IL-4Ra
floxed allele in mice. But I don't think that's necessary. A reviewer asking for that might
just want to prevent this result from getting published any time soon. Or you could ask
the authors to delve deeper into how IL-4 plus CD40 induces the changes (is it STAT6 or
IRS pathway), asking if it is unique or whether other cytokines can or can't do it, but
that's a different issue. Some people might criticize the model saying that it is artificial
or that the authors don't only look at the CD8 T-cells as physiologic readout, and not
the survival or something. I would challenge that point of view. Every model is artificial
and CD8 expansion is likely to be highly relevant to the outcome.

In short, I support the publication of this study without new experiments. It's solid, high
quality and important, and provides a new twist in DC biology. It's very interesting to
think of this model in the evolutionary context. My colleague Osami Kanagawa always
referred to IL-4 as the 'happiness factor', and this is pertinent here. The specificity is in
the gdT cell reacting to RAS. That's the danger signal. Then the happiness factor is
secreted to enable augmented responses both in the priming APC (inducing the IL-12)
and the CD8 T cell directly. My guess is that the IL-4 levels are in the happiness range
and not in the TH2-inducing high dose range of GATA-3 autoactivation. So it's

interesting that the old innate style gdT cells side reserves that mechanism even now.
Just musing.

We wish to thank the reviewer for their very positive comments and understood them to mean
that no extra experiments were required. We have addressed the concerns and added data as
requested below.

Minor Comments

Figure 2d: what is the reference population for the % value (Total T-cells? A particular
population of T-cells?)

We apologise for not specifying this. The reference population was all splenic $\gamma\delta$ T cells. In
the new analysis now provided in (Figure 2c-m and Supplementary Figure 2a-m), we have
clearly specified the reference population(s).

Figure 3: The figure title is a bit misleading, as the figure doesn't really examine the
CD40-CD40L signaling. The figure shows a negative result for a measurement (of T
cells) in CD40L knockouts. This is an interpretation that CD40 signaling is intact. I
would alter the words for more accuracy.

We agree and have changed the wording to 'Figure 3 - Antigen presentation is intact in the
absence of $\gamma\delta$ T cells, which are not an essential source of CD40L.' We hope that this
addresses the reviewer's concern.

Figure 4: It is too bad that IL-4 is so hard to measure in the mouse Vy1+ cells. In 4 f/g/h,
groups 4 and 6 seem to be mixed between the table and graphics. I would check this for
accuracy.

Thank you for appreciating the difficulty in conducting these studies. However, we were able
to achieve the measurement of IL-4 production from Vy1+ $\gamma\delta$ T cells using a reporter mouse
system. These data are now shown in (Figure 5b-i and Supplementary Figure 5a-h).

We have also corrected the table and apologise for the error in the labelling.

Figure 5+6: The cells transferred at day -3 in Figure 5 are only strongly enriched in DC1
(60-80%, according to methods). The cells in Figure 6 were sorted, but even with
sorting, there is no 100% purity. Is it possible that e.g. DC2 and not DC1 are the
population responding to IL-4 treatment? I doubt it, but maybe the authors could add
some text to discuss this. The BATF3 knockout result is the key here.

We have added text to the discussion on this point (lines 536-543).

Regarding Fig 6 b and d, could the authors make a supplementary data item, for
example a table. containing data on the most up/downregulated genes.

We have added this table and now also incorporate the changes induced by IL-4 in the
presence of IFN γ (**Supplementary Table 3**) and refer to it on line 387.

Figure S5: Do the Pb-T-1 cells also express the IL-4Ra on day 0 when, according to Fig 2,
the ydT cells are required

Yes, naive PbT-I T cells do express IL-4Ra on day 0. These data are now provided in (**Figure**
**7i**).